# Inflammatory signals from fatty bone marrow support *DNMT3A* driven clonal hematopoiesis

N. Zioni[1], A. Akhiad Bercovich[2], N. Chapal-Ilani[1], Tal Bacharach[1], N. Rappoport[2,3], A. Solomon[4], R. Avraham[4], E. Kopitman[5], Z. Porat [5], M. Sacma[6], G. Hartmut[6], M. Scheller [7], C. Muller-Tidow [8,9,10], D. Lipka [10], E. Shlush[11], M. Minden[12,13,14,15,16], N. Kaushansky[1,18] & Liran I. Shlush [1,17,18] ✉

Both fatty bone marrow (FBM) and somatic mutations in hematopoietic stem cells (HSCs), also termed clonal hematopoiesis (CH) accumulate with human aging. However it remains unclear whether FBM can modify the evolution of CH. To address this question, we herein present the interaction between CH and FBM in two preclinical male mouse models: after sub-lethal irradiation or after castration. An adipogenesis inhibitor (PPARγ inhibitor) is used in both models as a control. A significant increase in self-renewal can be detected in both human and rodent *DNMT3A*^Mut-HSCs when exposed to FBM. *DNMT3A*^Mut-HSCs derived from older mice interacting with FBM have even higher self-renewal in comparison to *DNMT3A*^Mut-HSCs derived from younger mice. Single cell RNA-sequencing on rodent HSCs after exposing them to FBM reveal a 6-10 fold increase in *DNMT3A*^Mut-HSCs and an activated inflammatory signaling. Cytokine analysis of BM fluid and BM derived adipocytes grown in vitro demonstrates an increased IL-6 levels under FBM conditions. Anti-IL-6 neutralizing antibodies significantly reduce the selective advantage of *DNMT3A*^Mut-HSCs exposed to FBM. Overall, paracrine FBM inflammatory signals promote *DNMT3A*-driven clonal hematopoiesis, which can be inhibited by blocking the IL-6 pathway.

Age-related accumulation of adipocytes in the human bone marrow (BM) is ubiquitous. At birth, the BM contains functionally active hematopoietic tissue, known as red marrow. With aging, there is a shift from red marrow to adipocyte-enriched yellow marrow that begins in the distal parts of the bones and expands proximally[1–3]. The initial step of adipogenesis is an enhanced lineage commitment of mesenchymal stem cells (MSCs) into preadipocytes followed by the expansion of preadipocytes into mature adipocytes, mainly in the cavities of trabecular bones[3]. BM adipocytes are different from adipocytes in other parts of the body[4]. Gene expression analysis of BM adipocytes suggested that they have distinct immune regulatory properties and high expression of pro-inflammatory cytokines (IL-1B, IL-15, and stem cell factor (SCF)[5].

Furthermore, BM adipocytes secrete IL-6, IL-8, and TNF-α[6]. As normal hematopoiesis and HSPCs are profoundly dependent on interactions with the microenvironment to maintain self-renewal capacity and normal differentiation[7], it has been speculated that the inflammatory signals originating from fatty bone marrow (FBM) could alter hematopoiesis. Stirewalt et al. have shown that BM stromal cells are determinants of the fate of hematopoietic progenitors and have an important role in the pathogenesis of MDS in which TNF-α induces significant changes in gene expression, particularly in apoptosis-related genes and cytokines/chemokines such as IL-6 and IL-8[8].

A broad variety of positive and negative effects on HSPCs and hematopoiesis have been observed when BM adipocytes are co-

cultured with hematopoietic cells[5]. BM adipocytes are less supportive of hematopoiesis than undifferentiated stromal or pre-adipocyte counterparts, owing to decreased production of growth factors such as GM-CSF and G-CSF[9] and secretion of signaling molecules with potential to impede hematopoietic proliferation such as neuropillin-1, lipocalin, adiponectin, and TNF-α. TNF-α and adiponectin demonstrate the complex effect of adipocytes through their ability to suppress progenitor activity while favorably affecting the most primitive HSCs. This imply that adipocytes restrict hematopoietic progenitor growth while maintaining the HSCs pool. The ability of BM adipocytes to support primitive hematopoietic cells is one of its beneficial effects[10,11]. On the other hand, HSCs from obese mice with FBM have reduced repopulation potential[12]. Limited data is available on the role of FBM in leukemia evolution. Adipocytes can modify cancer evolution and fitness by stimulating fatty acid oxidation and mitochondrial OXPHOS due to high-energy fatty acid transfer[13]. Leukemia stem cells use free fatty acids to generate energy, however, these studies were performed in the gonadal fat[14]. With age, BM adiposity correlates with increased density of mature myeloid cells and CD34+ HSPCs that contributing to age-related risk of myeloid malignancies[15]. Adipocytes might provide a protective niche for leukemia cells during chemotherapy by increasing the expression of the anti-apoptotic genes *BCL2* and *PIM2*[13]. On the other hand, AML cells can reduce FBM, resulting in imbalanced regulation of HSCs and in myelo-erythroid maturation[16].

While the interactions between FBM and normal or leukemic hematopoiesis were partially studied in the past, it remains unclear whether signals from FBM could shape the evolution of the early stages of leukemia and clonal hematopoiesis (CH). CH is defined by the expansion of HSPCs carrying leukemia-related mutations. HSPCs carrying leukemia-related mutations and still capable of differentiation are termed preL-HSPCs[17]. PreL-HSPCs are the evolutionary unit of CH[18], and as such, understanding the selective pressures which shape their fitness is crucial for the understanding of the early stages of leukemia evolution. As both FBM accumulation and clonal hematopoiesis are age-related and occur in the same geographic location, we hypothesized that the accumulation of FBM may provide selective advantage to specific preL-HSPCs. In mice, FBM accumulates in the tibia with age, however, strain has a considerable influence on the number of adipocytes and their age-related dynamics[19]. As in this study, we aimed at investigating both human and mice preL-HSPCs, we chose immunodeficient mice models; however, such mice have a shorter life span and tend to develop malignant tumors at a median age of 52 weeks thus aging them would be biased[20]. Therefore, we searched for other options to increase FBM in immunodeficient mice. We used different external stresses to induce FBM accumulation in long bones of NOD-SCID-Gamma (NSG) mice so we could study the interaction of FBM with both human and mice preL-HSPCs. As external stress can influence other cell types and other tissues, we used a crucial control, namely, a PPARγ inhibitor (PPARγi) which selectively inhibits the most important transcription factor in adipogenesis and reduces FBM formation after irradiation[21].

In this study, we provide evidence that inflammatory cytokines secreted by FBM can activate inflammatory signaling in preL-HSPCs. The activation of the IL-6 pathway by FBM can increase the self-renewal of preL-HSPCs carrying *DNMT3A* mutations.

## Results

### Establishment of FBM models in NSG mice

Murine BM does not recapitulate the dramatic age-related increase in FBM which can be observed in humans. Accordingly, in order to be able to test the effect of FBM on primary human HSPCs, we aimed at inducing FBM in NSG mice by external stress. Previous reports documented the accumulation of FBM few days to weeks after total body irradiation[22,23]. Nevertheless, total body irradiation causes cytokine storm and dramatic remodeling of all components of the BM

microenvironment, including osteoblasts, megakaryocytes, and vasculature[24–26]. To control all these off-targets effects of irradiation and other external stresses, we have used a control group of mice that were irradiated and treated with a PPARγi, bisphenol ADiGlycidyl ether (BADGE). Previous studies have shown that PPARγi treatment inhibits adipogenic differentiation in vitro[27].

To better characterize our sublethal irradiation model (225 rad), we irradiated NSG mice and noticed an enhanced adipocyte presence in the BM a week following irradiation. High FBM was maintained even two months after irradiation (Supplementary Fig 1b). These changes did not appear 24–48 h following irradiation (which is the usual time frame in which human HSPCs are injected to NSG mice) (Fig. 1a–c). Indeed, PPARγi treatment seven days prior to and post irradiation resulted in reduced FBM accumulation (Fig. 1d). Quantifying lipid levels in the BM by LipidTOX™ Deep Red Neutral Lipid Stain demonstrated significantly reduced FBM (30-folds) in the irradiated mice treated with PPARγi compared to irradiation without PPARγi treatment (Fig. 1e and Supplementary Fig. 1a). These results were validated by staining irradiated BM with the adipocyte marker fatty acid binding protein 4 (FABP4) (Fig. 1f and Supplementary Fig 2a–c) and with perilipin, which is located on the surface layer of intracellular lipid droplets[28] (Fig. 1g). To better characterize the adipocytes in our FBM model, we have performed single-nuclei RNA sequencing on BM-derived cells, as was previously described[29]. BM cells from both irradiated NSG mice and BADGE control mice were analyzed. Gene expression in our cells was compared to a recently reported atlas of mice BM cells[30]. While the majority of our cells originated from the hematopoietic lineage, we identified 25 adipocytes that most of them (68%) were retrieved from the irradiated mice and the rest from the BADGE control. Our adipocytes ($n = 25$) clustered together with previously reported BM-derived adipocytes ($n = 9$) in the same Metacells, and with other adipocyte Metacells (Supplementary Fig. 3a) Metacells#81) and expressed many known adipocyte markers (*Lepr, Adipoq, Cxcl12, Cxcl14, Kng1, Lpl*, and more) (Supplementary Fig. 3b). The low number of adipocytes in our experiments reduces the ability to identify heterogeneity in adipocyte populations in our data, however, it suggests that they share a high degree of similarity with normal adipocytes. The integration of our data with the data by Baccin et al. can be observed at: https://tanaylab.weizmann.ac.il/MCV/FBM/Single_nuclei_for_adipocytes/.

Altogether, these results provide evidence that low-dose irradiation of NSG mice causes more than just a hypocellular marrow, but also an active accumulation of BM adipocytes. To expand our capabilities to study FBM interactions with preL-HSPCs and control for off-targets effects of irradiation, we devised other FBM models. We first focused on a castration model as it recapitulates the age-related decline in testosterone among males[31]. We demonstrated that a month after castration (CAS), male mice developed FBM (Supplementary Fig. 1c). These findings were validated by staining bones from castrated mice with Perilipin (Supplementary Fig. 1d), which revealed an increase in adipocytes following castration and a substantial decrease after PPARγi treatment (Supplementary Fig. 1d). A similar effect could be achieved by treating NSG mice with the PPARγ activator (rosiglitazone maleate)[32] for three weeks (Supplementary Fig. 1e). Interestingly, analyzes of tibia bones derived from 1-year-old NSG-SGM3 mice (which express human IL-3 (hIL-3), hSCF and hGM-CSF) demonstrated high FBM levels compared to 1-year-old NSG or NSG-hSCF mice (Supplementary Fig. 1f). While the 1-year-old NSG-SGM3 model had high FBM levels, this model was linked to a reduction in the self-renewal capacity of normal HSCs[33] and therefore we did not used it in our future analysis. The fact that four different conditions can all contribute to FBM accumulation in mice, might explain why FBM accumulation with aging is both common and multifactorial. While all of our models increase FBM, they most probably have other molecular and cellular consequences in the BM and other tissues. To mitigate these of target effects, we chose to study the interaction between human preL-HSPCs and FBM in two different

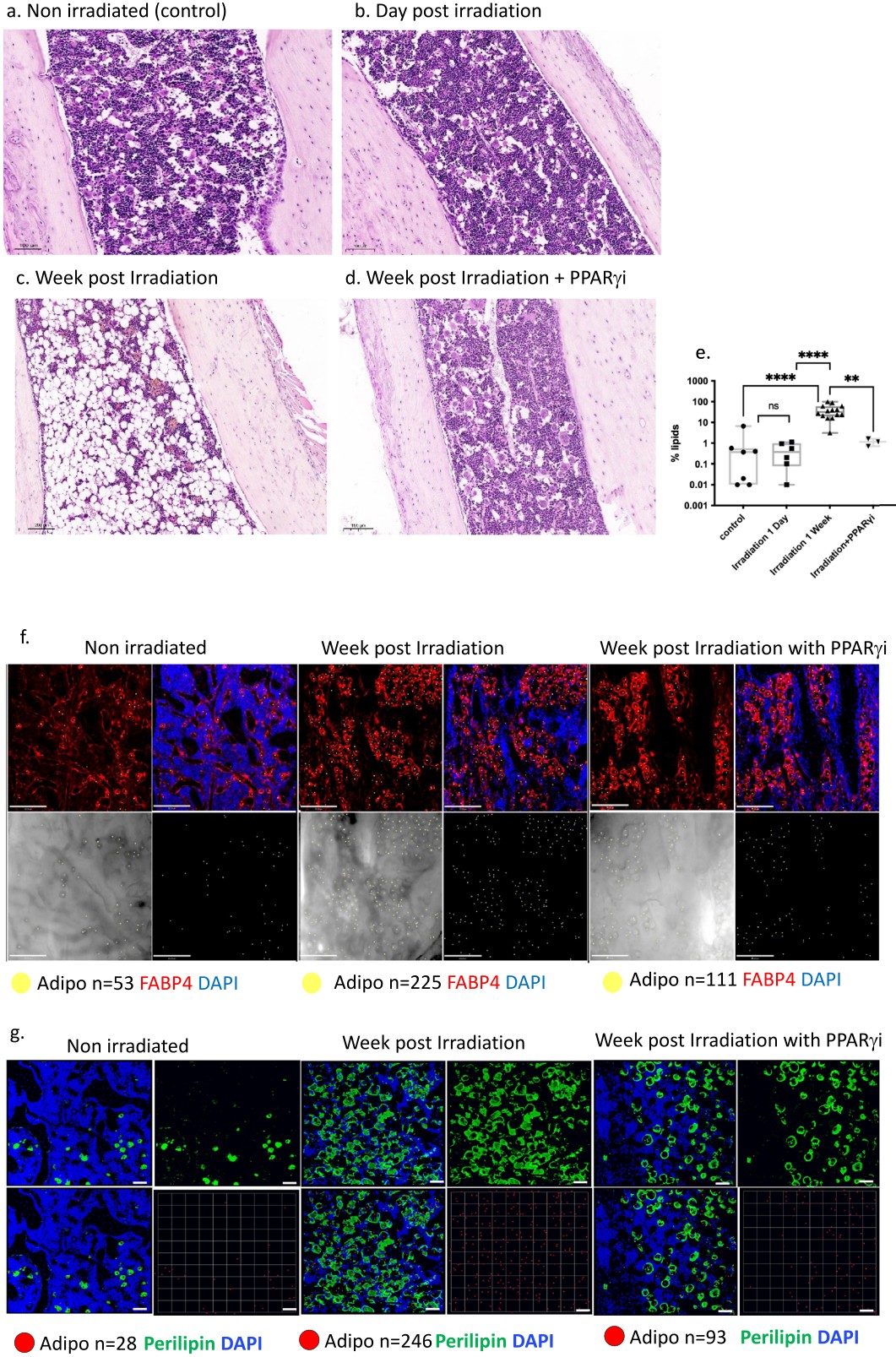

**e.**

**f.**

Non irradiated — Week post Irradiation — Week post Irradiation with PPARγi

🟡 Adipo n=53 FABP4 DAPI   🟡 Adipo n=225 FABP4 DAPI   🟡 Adipo n=111 FABP4 DAPI

**g.**

Non irradiated — Week post Irradiation — Week post Irradiation with PPARγi

🔴 Adipo n=28 Perilipin DAPI   🔴 Adipo n=246 Perilipin DAPI   🔴 Adipo n=93 Perilipin DAPI

models: post-irradiation and post-castration FBM models. We selected the irradiation model as it was the most robust (reproducible) and created the largest effect (accumulation of FBM). We decided to work with the castration model because it best reflects the aging process in males. We predicted that calibrating the BADGE control and the rosiglitazone maleate (both dealing with PPARγ) would be challenging. In addition, rosiglitazone maleate produced fewer adipocytes; despite

these drawbacks, this model should be improved and tested in the future, perhaps with a different control.

**FBM provides a selective advantage to human HSPCs carrying the *DNMT3A* R882H mutation**

Following the establishment of the different FBM models in NSG mice, we studied the interaction between FBM and primary preL-

**Fig. 1 | Models of BM adipogenesis in vivo in NSG mice.** H&E staining of NSG mice tibia/femur. NSG mice were irradiated with 225 rad (X ray). **a** Non-irradiated control normal bone marrow (NBM) (*n* = 5). scale bar: 100 μM. **b** One day following irradiation (*n* = 5) scale bar: 100 μM or **c** Seven days (*n* = 5) following irradiation mice were sacrificed. Shown are the H&E staining of tibial of one experiment out of five independent experiments. scale bar: 200 μM. **d** H&E staining of NSG (*n* = 5) mice that were administrated intraperitoneally for seven days with PPARγi (30 mg/kg) before and seven days after the irradiation. scale bar: 100 μM. **e** Adipocytes quantification by LipidTOX™ Deep Red Neutral Lipid Stain in ImageStream X Mark II Luminex imaging Flow Cytometer. Data presented as box and whiskers min to max. **\*\*P* < 0.005, \*\*\*\*P* < 0.0005. Each dot represents a mouse (control *n* = 7, irradiation 1 day *n* = 6, irradiation 1 week *n* = 14, irradiation+ PPARγi *n* = 3). Control vs. irradiation 1 week *P* < 0.0001, irradiation 1 day vs. irradiation 1 week *P* < 0.0001, irradiation 1 week vs. irradiation+ PPARγi *P* = 0.0015. All comparisons were performed using a two-tailed, nonpaired, nonparametric Mann–Whitney test compare ranks. **f** Representative stacked whole-mount immunofluorescence staining of epiphyseal-metaphyseal BM femur derived from NBM, FBM and FBM & PPARγi treated NSG mice. Adipocytes are depicted by FABP4 expression and by distinctive unilocular morphology in the DIC (differential interference contrast) channel. DAPI in blue. Adipocytes are additionally marked by yellow dots. Scale Bar: 200 μm; *n* = 4 775 μm × 775 μm × 50 μm stacked images from 2 mice and 2 bones (femur, tibia) each group. **g** Representative 3D whole-mount immunofluorescence staining of epiphyseal-metaphysealBMlong bones derived from control, NBM, FBM, and FBM & PPARγi-treated NSG mice. Adipocytes are depicted by Perilipin expression and by distinct iveunilocular morphology in the DIC (differential interference contrast) c channel. DAPI in blue. Adipocytes are additionally marked by red dots. 1 unit scale bar: 70.99 μm; *n* = 12–14708 μm × 708 μm × 30 50 μm stacked images from three mice and two bones (femur, tibia) each group. Source data are provided as a Source Data file.

HSPCs carrying the *DNMT3A* R882H mutation. We chose to focus on *DNMT3A* R882 mutations as they are the most common mutations in CH[34]. To this end, we selected an AML sample carrying both the *DNMT3A* mutation (VAF of 50%) and the *NPM1*c mutation (VAF of 50%). However, when injected into FBM NSG mice, the engrafting cells following amplicon sequencing were only carrying the *DNMT3A* mutation, suggesting that only the preL-HSPCs with *DNMT3A* mutation could engraft in NSG mice at this specific sample (sample #160005)[35] (Fig. 2a and Supplementary Fig 4a). Engraftment of sample #160005 cells was higher under FBM conditions compared to normal BM (NBM) mice and the PPARγi-treated control in which less adipocytes accumulated (Fig. 2b). The graft had a multilineage differentiation capacity as shown in Supplementary Fig. 4b. Furthermore, we sequenced NBM samples, however, due to the limited engraftment, we were unable to obtain any human cells after sorting and no human reads were available after sequencing. We repeated this experiment on the castration (CAS) FBM model, and again a significantly higher engraftment of sample #160005 cells was detected under FBM conditions. The administration of PPARγi to FBM-CAS mice resulted in a significant decrease of sample #160005 cells engraftment (Fig. 2c). To further validate these results, we used HSPCs collected for auto-transplantation (Sample# 141464). This sample had a high variant allele frequency (VAF) of the *DNMT3A* R882H mutation, and had *DNMT3A* R882H mutation following transplantation to FBM as determined by amplicon sequencing (Fig. 2d). Again, a significantly higher engraftment was observed in both the irradiation and castration FBM models (Fig. 2e, f).

To study the interaction between FBM and normal hematopoiesis, we used normal CD34+ cells (no *DMNT3A* mutation) derived from different sources. We transplanted wild-type (WT) CD34+ cells from pooled cord blood samples and from aged-matched healthy donors without clonal hematopoiesis. In contrary to the *DNMT3A*-R882 mutated cells, the CD34+ cells from both sources had no increased engraftment when exposed to FBM (Fig. 2g, h). These results may suggest a role for an adipocyte-rich environment in enhancing engraftment of human preL-HSPCs, but not for normal cord blood-derived HSPCs. To validate our results and better understand the mechanisms behind the increased engraftment of human preL-HSPCs under FBM conditions, we turned to a rodent model of mutant *DNMT3A* (*DNMT3A*^Mut)[36].

## FBM support *DNMT3A*^Mut HSPCs derived from a genetic rodent model

For the next set of experiments, we crossed the human *DNMT3A* R882H knock-in mice model[36], with mice carrying a Cre recombinase allele under the VAV promotor which is expressed only in the hematopoietic system to create hematopoietic specific *DNMT3A* mutant (*DNMT3A*^Mut) mice. C57BlxVAV-cre mice were used as a control group (denoted *DNMT3A*^WT).

The injection of two months BM-derived *DNMT3A*^Mut cells (CD45.2) intrafemorally (IF) to NSG (CD45.1) mice with FBM resulted in significantly elevated engraftment of *DNMT3A*^Mut cells in comparison to control mice with NBM or mice who were irradiated and treated with the PPARγ inhibitor (PPARγi) (Fig. 3a). This increased engraftment under FBM conditions could not be observed when *DNMT3A*^WT cells were injected (Fig. 3a). To test whether the increased engraftment of *DNMT3A*^Mut preL-HSPCs under FBM conditions was due to increased self-renewal, we repeated our experiment and injected the cells exposed to FBM and NBM into secondary recipients that had FBM. Engraftment analysis demonstrated that the highest increase in self-renewal was present in *DNMT3A*^Mut cells exposed to FBM (Fig. 3b).

Next, we hypothesized that as *DNMT3A* R882 mutations are known to cause hypomethylation over time[37] and can cause human leukemia after a long latency period[38], the effects of FBM could be even more pronounced if older preL-HSPCs were to be studied. Indeed, a significant increase in engraftment was detected following transplantation of old *DNMT3A*^Mut cells derived from 1-year-old mice injected into NBM, FBM and PPARγi controls compare to two months *DNMT3A*^Mut-derived BM (Supplementary Fig 4c). *DNMT3A*^Mut cells derived from 1-year-old mice injected into FBM had the most significant growth advantage in comparison to NBM and PPARγi controls. When *DNMT3A*^WT cells were injected, this effect of FBM could not be observed (Fig. 3c and Supplementary Fig. 4c). Interestingly, the administration of PPARγi to FBM mice transplanted with 1-year-old *DNMT3A*^WT cells resulted in a significant increase of engraftment (Fig. 3c and Supplementary Fig. 4c). Similar results on the effect of PPARγ inhibition on HSCs have been reported in the past[39]. To test whether the increased engraftment of 1-year-old cells was due to increased self-renewal, secondary engraftment was performed. In agreement with our results derived from 2-month-old mice, preL-HSPCs carrying the *DNMT3A*^Mut and exposed to FBM had significantly higher secondary engraftment compared to controls (Fig. 3d, e). These results suggest that FBM provides higher selective advantage to older *DNMT3A*^Mut cells through increased self-renewal.

We repeated this experiment on the castration FBM model (FBM-CAS), and again a significantly higher engraftment of *DNMT3A*^Mut cells derived from 1-year-old mice injected into FBM-CAS in comparison to NBM and PPARγi control was detected. FBM-CAS had no effect on *DNMT3A*^WT cells (Fig. 3f). To extend our results beyond *DNMT3A* R882 mutations we tested the effects of FBM on *DNMT3A* haploinsufficient mice (*DNMT3A*^haplo) as frameshift mutations causing such genotype are common among humans[40]. Significantly increased engraftment of *DNMT3A*^haplo was noted when cells from 2-month-old mice were exposed to FBM both in primary and secondary recipients (Supplementary Fig 5a, b). Similar results were obtained when 1-year-old cells were transplanted (Supplementary Fig 5c–e). No significant differences were observed between the engraftment of 1-year-old *DNMT3A*^haplo and *DNMT3A*^Mut mice when transplanted to FBM

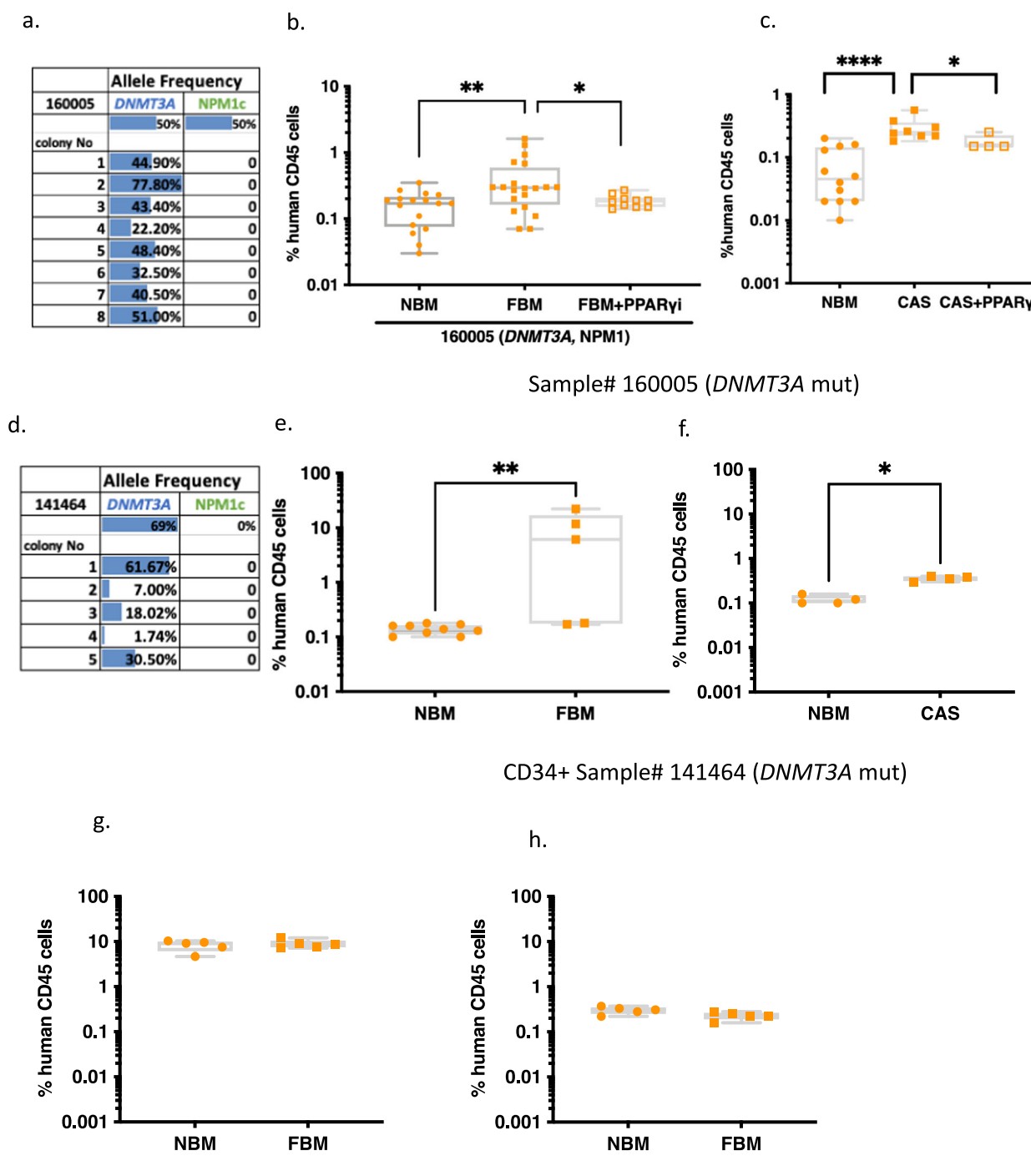

Sample# 160005 (*DNMT3A* mut)

CD34+ Sample# 141464 (*DNMT3A* mut)

CD34+ Cord Blood                 CD34+ (*DNMT3A* wt)

(Supplementary Fig. 5f). This result suggests that different *DNMT3A* mutations (and not just R882) have a selective advantage under FBM conditions. However, it remains unclear whether the same effect holds for other pre-Leukemic mutations (pLMs).

**FBM does not support HSPCs carrying *SRSF2* P95H mutations derived from a genetic rodent model**

To address this question, we used a *SRSF2* P95H knock-in model[41]. BM cells derived from 2-month-old mice were injected IF into FBM or NBM NSG mice. In contrast to our findings with *DNMT3A*^Mut cells, no significantly higher engraftment of *SRSF2*^Mut cells under FBM conditions was observed (Supplementary Fig. 6a). Furthermore, no differences in engraftment were observed in NBM or FBM mice following

transplantation of 1-year-old *SRSF2*^Mut BM-derived cells in both primary and secondary transplantation (Supplementary Fig. 6b). Altogether, our results so far indicate that FBM provides a selective advantage to preL-HSPCs carrying *DNMT3A*^Mut. While these results are supported by both human and mice data the molecular mechanism behind this observation remains elusive.

**HSPCs carrying *DNMT3A*^Mut exposed to FBM maintain their stem cell pool**

To further study why HSPCs carrying *DNMT3A*^Mut have increased self-renewal upon exposure to FBM, we injected 1-year-old *DNMT3A*^Mut and *DNMT3A*^WT HSPCs to either FBM or NBM mice. Three days following injection (see "Methods" for rational) (10 days after low-dose

**Fig. 2 | Increased engraftment of human *DNMT3A*-mutated preleukemic cells under FBM conditions. a** Variant allele frequency (VAF) analysis of *DNMT3A* and *NPM1c* mutation of the engrafted preleukemic cells by amplicon sequencing in FBM NSG mice. NSG mice were irradiated with 225 rad, and after a week injected intrafemur (IF) with CD3-depleted $1 \times 10^6$ AML primary human cells (sample #160005) (50% *hDNMT3A*^R882H, 50% *NPM1c*). Eight weeks later, mice were sacrificed and BM was flashed from tibias and femurs and sequenced. All engrafted cells were preleukemic namely carrying only the *DNMT3A*^R882H mutation without *NMP1c* mutation. NBM vs. FBM $P = 0.0038$, FBM vs. FBM+ PPARγi $P = 0.0377$.
**b** Engraftment of human preleukemic cells in normal bone marrow (NBM) ($n = 17$), fatty bone marrow (FBM) ($n = 20$) and in irradiated NSG mice ($n = 10$) treated with PPARγi (FBM+ PPARγi). Eight weeks following $1 \times 10^6$ AML primary human cells transplantation, BM was flashed from tibia/femur and expression of human CD45+ was measured by FACs. NBM vs. FBM $P = 0.0038$, FBM vs. FBM+ PPARγi $P = 0.037$.
**c** Engraftment of preleukemic cells in NBM ($n = 12$), in castrated (CAS) NSG mice ($n = 8$) and in CAS+ PPARγi ($n = 4$). One month following castration, $1 \times 10^6$ AML primary human cells were transplanted. Eight weeks following $1 \times 10^6$ AML primary human cells transplantation, BM was flashed from tibia/femur and expression of human CD45+ was measured by FACs. NBM vs. FBM $P < 0.00001$, FBM vs. FBM+

PPARγi $P = 0.044$. **d–f** The same analysis was carried out for another human sample as described in (**a–c**). (Sample #141464 CD34+ enriched) (VAF 69% of the *DNMT3A*^R882H in the primary sample). **e** Engraftment of human pre- leukemic cells in NBM ($n = 9$ mice), FBM ($n = 5$ mice). NBM vs. FBM $P = 0.004$. **f** Engraftment of preleukemic cells in NBM ($n = 4$ mice) and in castrated (CAS) NSG mice ($n = 4$). NBM vs. CAS $P = 0.0286$. **g** Level of chimerism in NBM ($n = 5$ mice) or FBM NSG mice ($n = 5$) transplanted with $50 \times 10^5$ CD34+ WT cord blood cells. **h** Transplantation of CD34+ from mobilized peripheral blood autotransplant bags to FBM and NBM NSG mice. In total, $1.2 \times 10^5$ cells CD34+ from mobilized peripheral blood autotransplant bag (#141519) that have been sequenced and identified w/o age related clonal hematopoiesis (ARCH) mutation, were transplanted intrafemur to NBM ($n = 5$ mice) and FBM NSG mice ($n = 5$). Eight weeks following cells transplantation, mice were sacrificed, BM was flashed from tibia/femur and expression of human CD45+ was measured by FACs. Human engraftment was assessed according to presence of ≥0.1% human CD45+ cells. Data in this figure presented as box and whiskers min to max, *$P < 0.05$, **$P < 0.005$, ***$P < 0.0005$, ****$P < 0.00005$. Each dot represents a mouse. All comparisons were performed using a two-tailed, non-paired, nonparametric Mann–Whitney test compare ranks. Source data are provided as a Source Data file.

irradiation), we isolated single LIN-KIT+ (LK) cells and performed single-cell RNA-seq (scRNA-seq) using the MARS-Seq technology[42]. After filtering outlier cells, we identified 1999 LK cells from the different conditions (Supplementary Table 1). First, we aimed to explore whether exposure to FBM influenced the abundancy of different HSPCs. To this aim, we used the MetaCell2 algorithm to assign single cells to Metacell which have unique gene expression programs[43,44]. Each Metacell can then be assigned to one of the major HSPCs lineages based on the expression of canonical (cell type related) genes. The main genes expressed along the hematopoietic hierarchy were chosen based on previous reports[45] (Supplementary Figs. 7–10) (Supplementary Table 1). The scRNA-seq data can be explored in the following link: https://tanaylab.weizmann.ac.il/MCV/FBM/.

Interestingly, all our experiments involving injection of cells to either NBM or FBM mice, showed a marked reduction in HSCs. The only exception to this rule were the *DNMT3A*^Mut cells exposed to FBM who maintained their HSC pool and clustered together with naive cells (LSK cells extracted directly from BM with no injection) (Fig. 4a, b). *DNMT3A*^Mut cells exposed to FBM had tenfold more HSCs compared to *DNMT3A*^WT cells exposed to FBM ($P = 2.3e-9$), and a significant increase in their HSC pool compared to *DNMT3A*^Mut cells exposed to NBM ($P = 0.03$) (Fig. 4c and Supplementary Data 2). The maintenance of HSCs among *DNMT3A*^Mut cells exposed to FBM was followed by expansion of myeloid progenitors as opposed to the enrichment of lymphoid progenitors in the naive LSK cells (Fig. 4a, b). LSK cells from other conditions after transplantation lose some of their HSCs which differentiate mainly into the myeloid lineage (Fig. 4a, b and Supplementary Fig. 14b). We have repeated the scRNA-seq experiment with more cells and with the 10X genomics platform from *DNMT3A*^Mut cells exposed to FBM or NBM and validated the significantly increased HSCs population in the *DNMT3A*^Mut cells exposed to FBM (this time sixfold more HSCs than under NBM conditions) (Supplementary Fig. 15a, b). Altogether, the scRNA-seq data like the secondary engraftment experiments also suggest that *DNMT3A*^Mut cells exposed to FBM undergo self-renewal, while under NBM conditions they undergo (mostly myeloid) differentiation.

## HSPCs carrying *DNMT3A*^Mut exposed to FBM activate inflammatory pathways
To better understand why *DNMT3A*^Mut HSCs exposed to FBM undergo increased self-renewal and not differentiation, we performed differential expression analysis from the scRNA-seq data. Unsupervised clustering of the single cells using the UMAP algorithm[46] suggested that *DNMT3A*^Mut cells exposed to FBM were almost exclusively clustered together in a single cluster (Supplementary Fig. 11a). To better characterize the *DNMT3A*^Mut FBM cluster, we calculated differential

gene expression between the different clusters in the UMAP (Supplementary data 1) and used all cells in each cluster. Next, we performed ranked GSEA analysis on the differentially expressed genes of the cluster containing the *DNMT3A*^Mut FBM cells. We identified that genes related to the IFN-α, IFN-γ, TNF-α, and the IL-6 signaling pathways were upregulated (Supplementary Fig. 11b) (Supplementary Table 2). To better understand which of these pathways are truly upregulated, we created a score for each single cell which was based on the expression of the genes in the significant gene sets (IFN-α, IFN-γ, TNF-α, and IL-6). *DNMT3A*^Mut cells exposed to FBM had the highest IFN-α score which was significantly higher than all other cells (Supplementary Fig. 12a) (Supplementary Data 3 and 4). However also *DNMT3A*^WT cells exposed to FBM had significantly increased IFN-α score compared to other groups. Based on previous reports[47], we suspected that the activation of IFN-α response in the scRNA-seq data could be due to a stress response and not specific to the *DNMT3A*^Mut cells exposed to FBM. Furthermore, we noticed that the IFN-α gene set is not specific and has high overlap with the IFN-γ gene set (73 gene out of 98 in the IFN-α gene set are also present in the IFN-γ gene set). Accordingly, we reanalyzed the IFN-α score this time without the IFN-γ overlapping genes (Supplementary Fig. 12b). *DNMT3A*^Mut exposed to FBM still had a significantly higher IFN-α score, however, the IFN-α score of *DNMT3A*^WT cells exposed to FBM was similar to all other conditions (Supplementary Fig. 11b). When we studied the IFN-γ score derived from genes exclusive to its gene set, a similar pattern was observed. The highest IFN-γ score was recorded in *DNMT3A*^Mut cells exposed to FBM followed by *DNMT3A*^WT exposed to FBM (Supplementary Fig. 12c). TNF-α gene set score analysis demonstrated the same TNF-α score when comparing *DNMT3A*^Mut cells exposed to FBM to most other conditions (Supplementary Fig. 12d) (Supplementary Data 3 and 4). The score for the IL-6 gene is significantly higher in *DNMT3A*^Mut cells injected to FBM in comparison to all other conditions (Fig. 4c) (Supplementary Data 3 and 4). Altogether, the analysis of the scRNA-seq suggested that *DNMT3A*^Mut and to a lesser extent *DNMT3A*^WT exposed to FBM activate the IFN-γ, and IL-6 pathways. The IFN-α and TNF-α signaling were not stable after exclusion of IFN-γ related genes, and therefore we cannot firmly conclude that they were activated. We have repeated the scRNA-seq experiment with more cells and with the 10X genomics platform from *DNMT3A*^Mut cells exposed to FBM or NBM and validated that exposure to FBM activated the IFN-γ and IL-6 pathways in *DNMT3A*^Mut derived from 1-year-old mice (Supplementary Fig. 15c–g).

## IL-6 is secreted by FBM
As we discovered that *DNMT3A*^Mut cells exposed to FBM demonstrated activation of several inflammatory pathways, we next hypothesized

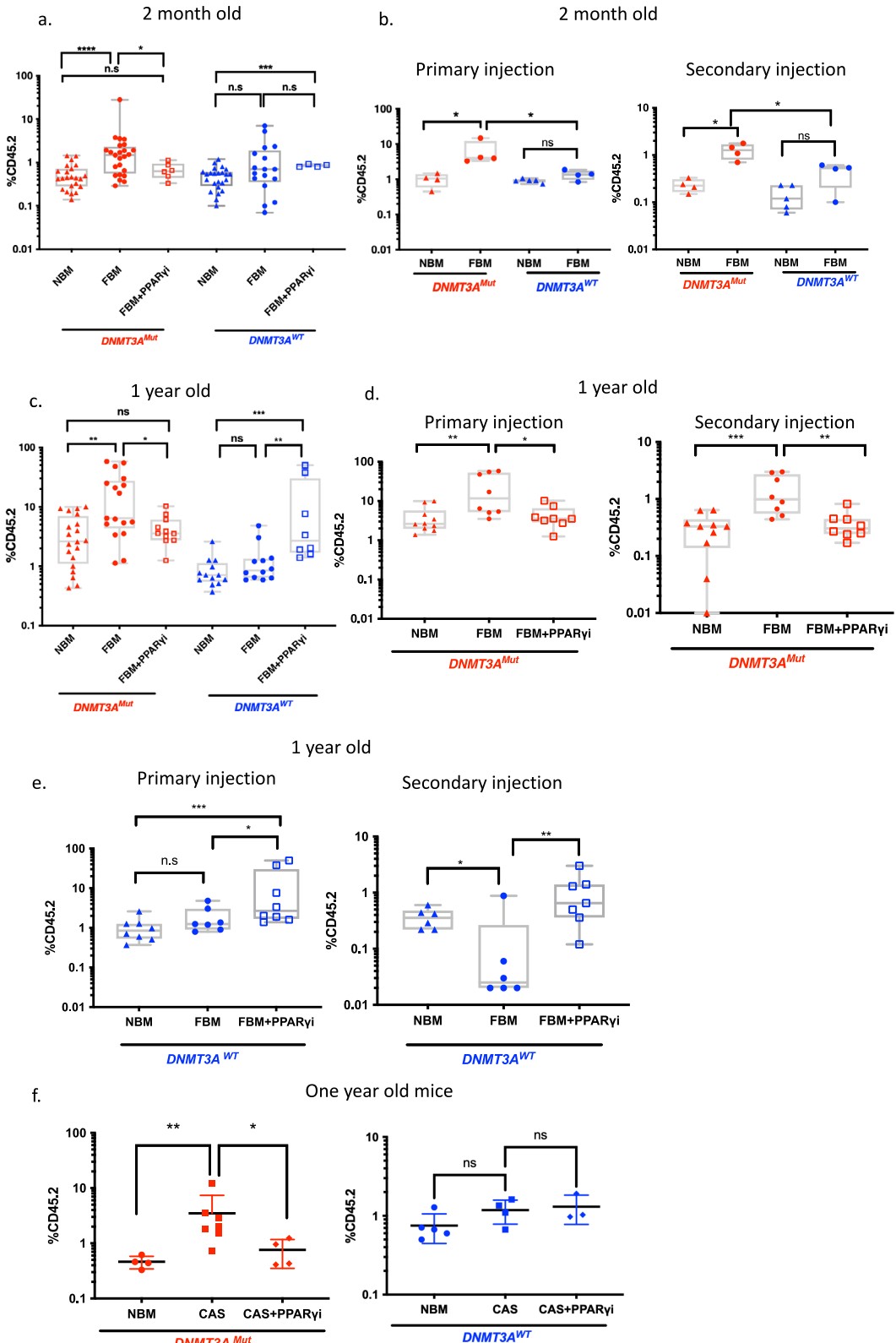

that adipocytes might secrete pro-inflammatory cytokines which will induce such a response. We used a multiplex cytokines assay (FirePlex, Abcam) to measure cytokine levels in the BM fluid from mice with and without FBM. We initially analyzed cytokine secretion in 1-year-old NBM donor mice, comparing Mut to WT (no cells injected). No significant differences were detected in cytokines secretion between donor Mut and WT mice with NBM (Supplementary Fig. 13a). We also

examined cytokines secretion in NSG mice with either NBM or FBM prior to donor injection. NSG FBM mice had higher levels of IL-6, IL-2, TNF-α, MIP1a, MIP1b, and IL-1β secretion (Fig. 4d). NSG FBM-CAS also characterized in increase of IL-6 secretion compared to NBM (Supplementary Fig. 13b). Importantly, following administration of PPARγi to FBM NSG mice, prior to donor injection, the levels of IL-6 and MIP1b were significantly decreased, suggesting correlation to FBM condition.

**Fig. 3 | Engraftment of _DNMT3A_<sup>Mut</sup>-derived BM cells in NSG mice. a** FACs analysis of young-2-month-old $6 \times 10^6$ _DNMT3A_<sup>Mut</sup> (CD45.2) (red) or _DNMT3A_<sup>WT</sup> (CD45.2) (blue) BM-derived cells transplanted to normal bone marrow (NBM) NSG mice (_DNMT3A_<sup>Mut</sup> to $n = 23$ NSG mice, _DNMT3A_<sup>WT</sup> to $n = 24$ NSG mice), and to fatty bone marrow (FBM) (_DNMT3A_<sup>Mut</sup> to $n = 24$ NSG mice, _DNMT3A_<sup>WT</sup> to $n = 17$ NSG mice) and to irradiated NSG mice (CD45.1) treated with PPARγi (FBM+ PPARγi) (_DNMT3A_<sup>Mut</sup> to $n = 6$ NSG mice, _DNMT3A_<sup>WT</sup> to $n = 4$ NSG mice). Eight weeks following transplantation, BM was flashed from tibia/femur and expression of mCD45.2 was measured by FACs. Engraftment was assessed according to presence of ≥0.1% mCD45.2 cells. _DNMT3A_<sup>Mut</sup>: NBM vs. FBM $P < 0.00001$, FBM vs. FBM+ PPARγi $P = 0.0452$. _DNMT3A_<sup>WT</sup>: NBM vs. FBM+ PPARγi $P = 0.0067$. **b** Self-renewal of _DNMT3A_<sup>Mut</sup> derived BM cells in FBM NSG mice. Primary transplantation of _DNMT3A_<sup>Mut</sup> or _DNMT3A_<sup>WT</sup> was performed as detailed in (**a**). For the primary transplantation, BM-derived cells were transplanted to NBM mice (_DNMT3A_<sup>Mut</sup> to $n = 4$ NSG _DNMT3A_<sup>WT</sup> to $n = 5$ NSG mice) and to FBM mice ($n = 4$ for _DNMT3A_<sup>Mut</sup>, $n = 4$ for _DNMT3A_<sup>WT</sup>). After eight weeks, cells were harvested and a secondary transplantation was performed to FBM NSG mice. Primary: _DNMT3A_<sup>Mut</sup>: NBM vs. FBM $P = 0.02$, _DNMT3A_<sup>Mut</sup> FBM vs _DNMT3A_<sup>WT</sup> FBM $P = 0.02$. Secondary: _DNMT3A_<sup>Mut</sup>: NBM vs. FBM $P = 0.0286$, _DNMT3A_<sup>Mut</sup> FBM vs _DNMT3A_<sup>WT</sup> FBM $P = 0.02$. **c** FACs analysis of 1-year-old $6 \times 10^6$ _DNMT3A_<sup>Mut</sup> (CD45.2) (red) or _DNMT3A_<sup>WT</sup> (CD45.2) (blue) BM-derived cells transplanted to NBM mice (_DNMT3A_<sup>Mut</sup> to $n = 20$ NSG mice, _DNMT3A_<sup>WT</sup> to $n = 13$ NSG mice), to FBM mice (_DNMT3A_<sup>Mut</sup> to $n = 18$ NSG mice, _DNMT3A_<sup>WT</sup> to $n = 12$ NSG mice) and to irradiated NSG mice (CD45.1) treated with PPARγi (FBM+ PPARγi) (_DNMT3A_<sup>Mut</sup> to $n = 10$ NSG mice, _DNMT3A_<sup>WT</sup> to $n = 8$ NSG mice) performed as detailed in (**a**). _DNMT3A_<sup>Mut</sup> : NBM vs. FBM $P = 0.0023$, FBM vs. FBM+ PPARγi $P = 0.0369$. _DNMT3A_<sup>WT</sup>: NBM vs. FBM+ PPARγi $P = 0.0001$, FBM vs. FBM+ PPARγi $P = 0.0015$. **d, e** Primary and secondary transplantation of middle-aged _DNMT3A_<sup>Mut</sup> (**d**) in primary: _DNMT3A_<sup>Mut</sup> derived BM was transplanted to NBM mice ($n = 10$) and to FBM mice ($n = 8$) and to irradiated NSG mice treated with PPARγi (FBM+ PPARγi) ($n = 8$). For the secondary transplantation: _DNMT3A_<sup>Mut</sup>-derived BM cells from NBM, FBM, and FBM+ PPARγi were transplanted to FBM NSG mice ($n = 10$, $n = 8$, $n = 8$ respectively) _DNMT3A_<sup>WT</sup>. For primary transplantation: NBM vs. FBM $P = 0.0041$. FBM vs. FBM+ PPARγi $P = 0.0281$. For secondary transplantation: NBM vs. FBM $P = 0.0005$. FBM vs. FBM+ PPARγi $P = 0.0022$. **e** In primary: _DNMT3A_<sup>WT</sup> derived BM cells were transplanted to NBM ($n = 8$) and to FBM mice ($n = 7$) and to irradiated NSG mice treated with PPARγi (FBM+ PPARγi) ($n = 8$). In secondary: _DNMT3A_<sup>WT</sup>-derived BM cells from NBM, FBM, and FBM+ PPARγi were transplanted to FBM NSG mice ($n = 6$, $n = 6$, $n = 7$, respectively). For primary: NBM vs. FBM+ PPARγi $P = 0.0005$, FBM vs. FBM+ PPARγi $P = 0.0022$. For secondary: NBM vs. FBM $P = 0.026$, FBM vs. FBM+ PPARγi $P = 0.0064$. **f** Left: primary transplantation of _DNMT3A_<sup>Mut</sup>-derived BM to NBM mice ($n = 4$), to FBM-CAS mice ($n = 7$), and to FBM– CAS mice treated with PPARγi (CAS FBM+ PPARγi) ($n = 4$). Right: primary transplantation of _DNMT3A_<sup>WT</sup> derived BM to NBM mice ($n = 5$), to FBM– CAS mice ($n = 4$) and to FBM- CAS mice treated with PPARγi (FBM- CAS +PPARγi) ($n = 3$). _DNMT3A_<sup>Mut</sup>: NBM vs. CAS $P = 0.0061$, _DNMT3A_<sup>WT</sup>: CAS vs. CAS + PPARγi $P = 0.0242$. For all results, data presented as box and whiskers min to max *$P < 0.05$, **$P < 0.005$, ***$P < 0.0005$, ****$P < 0.00005$. Each dot represents a mouse. Data in this figure presented as box and whiskers min to max. All comparisons were performed using a two-tailed, nonpaired, nonparametric Mann–Whitney test compare ranks. n.s not significant. Source data are provided as a Source Data file.

Next, we tested cytokine expression in our single-nuclei RNA-seq data integrated model. The adipocytes derived from irradiated NSG mice expressed IL-6, IL-18, and IL-1b. Specifically, IL-6 was not expressed in all adipocyte Metacells (from ref. 30) and the same was true for other cytokines as well. These suggest that some of the heterogeneity between BM adipocytes is driven by cytokine expression (Supplementary Fig. 3c). To better characterize such heterogeneity in our adipocyte, many more cells are needed, which is not the scope of the current study. To validate that the cytokine secretion we observed, was specific to the FBM, we measured cytokine levels in the mice serum in parallel. Significant differences were noted only in MCP1 between FBM and NBM (Fig. 4e). To learn whether cytokine levels under FBM conditions remain increased following cells transplantation, we first transplanted two months _DNMT3A_<sup>Mut</sup> and _DNMT3A_<sup>WT</sup> BM cells to FBM and NBM mice. A significant increase in IL-6 secretion was noted regardless of the genotype of injected cells (Supplementary Fig. 13c, d). Nonetheless, transplantation of 1-year-old _DNMT3A_<sup>Mut</sup> and _DNMT3A_<sup>WT</sup> BM cells to FBM also resulted in increase in IL-6 secretion compare to NBM, regardless of the genotype of injected cells and decreased following administration of PPARγi (Fig. 4f, g). These results suggest that the increased IL-6 score (IL-6 signaling activation) we observed in _DNMT3A_<sup>Mut</sup> cells, might be the result of IL-6 secretion from FBM regardless of which cells are transplanted. Furthermore, following transplantation of 1-year-old _DNMT3A_<sup>WT</sup> or _DNMT3A_<sup>Mut</sup> BM-derived cells to FBM NSG mice, a significant increase in the secretion of TNF-α was observed (Fig. 4f, g). Altogether, we concluded that IL-6 levels in BM fluid were increased under all conditions and were independent of cells injected. TNF-α showed a similar but less consistent pattern. Based on these results and the results of the scRNA-Seq, we chose to further explore the role of IL-6 in the self-renewal of _DNMT3A_<sup>Mut</sup> cells under FBM conditions. Of note, our results are in agreement with previous reports on FBM cytokine secretions in humans[6].

## IL-6 provides selective advantage to preL-HSPCs carrying _DNMT3A_<sup>Mut</sup> in a methylcellulose colony-forming assay

The interaction between adipocytes and _DNMT3A_<sup>Mut</sup> BM-derived cells via IL-6 was validated in vitro by serial replating of preL-HSPCs carrying the _DNMT3A_<sup>Mut</sup> in methylcellulose. _DNMT3A_<sup>Mut</sup> or _DNMT3A_<sup>WT</sup> BM-derived cells were used for the colony-forming cell (CFC) assay with and without IL-6. _DNMT3A_<sup>Mut</sup> cells in the presence of IL-6 demonstrated increased self-renewal over the three control groups (_DNMT3A_<sup>WT</sup> cells with IL-6, _DNMT3A_<sup>WT</sup> without IL-6, and _DNMT3A_<sup>Mut</sup> without IL-6). The control groups did not survive the third replete compared to the _DNMT3A_<sup>Mut</sup> BM-derived cells with IL-6 who survived two more cycles of replating than the controls (Fig. 5a, b). Furthermore, we set up experiments in which we cultured human/mouse bone marrow-derived mesenchymal stem cells (MSCs) and adipocytes in vitro (Supplementary Fig. 13e, f). Adipocytes were quantified using oil red staining. After 10 days in culture (without HSPCs), we collected the media and found significantly elevated IL-6 levels in the adipocyte cultures compared to MSCs cultures in both mice (Fig. 5c) and human (Fig. 5e). Next, 1-year-old mice _DNMT3A_<sup>Mut</sup> BM-derived Lin⁻ cells were co-cultured with adipocytes/MSCs. After 10 days, we used a colony-forming unit (CFU) assay in which cells were seeded in methylcellulose with adipocytes or MSCs-derived media. _DNMT3A_<sup>Mut</sup> HSPCs co-cultured with adipocytes, produced significantly more colonies than co-culturing with MSCs (Fig. 5d). This experiment was replicated with _DNMT3A_ mutated CD34+ cells derived from four human AML samples: sample #141164 (Fig. 5f), sample #141467 (Supplementary Fig. 13g), #150279 (Supplementary Fig. 13h) and sample #141464 (Supplementary Fig. 13i). Again, co-culturing human _DNMT3A_ mutant CD34+ cells with adipocytes yielded significantly more colonies than co-culturing with MSCs (Fig. 5f, g and Supplementary Fig. 13g–i). We have genotype 13 of the colonies co-cultured with adipocytes and all of them were positive for _DNMT3A_ and negative for _NPM1c_, suggesting that they originate from preL-HSPCs (Fig. 5g). Altogether, these findings suggest that adipocytes secreting IL-6 (and possibly other factors) provide a selective advantage to both human and mouse HSPCs carrying _DNMT3A_ mutations in vitro.

## In vivo treatment with neutralizing IL-6 antibodies (Ab) results in decreased self-renewal of _DNMT3A_<sup>Mut</sup> cells under FBM conditions

To study the effects of IL-6 on _DNMT3A_ preL-HSPCs in vivo, we transplanted 1-year-old _DNMT3A_<sup>Mut</sup> BM-derived cells into NBM and FBM NSG mice that had been treated with neutralizing IL-6 antibodies (Ab) two days before and seven days after transplantation. The administration of neutralizing IL-6 Ab resulted in a significant decrease in engraftment of _DNMT3A_<sup>Mut</sup> BM-derived cells (Fig. 6a). For the secondary transplantations, cells were harvested from primary mice and injected to NBM NSG mice with no further treatment. Secondary

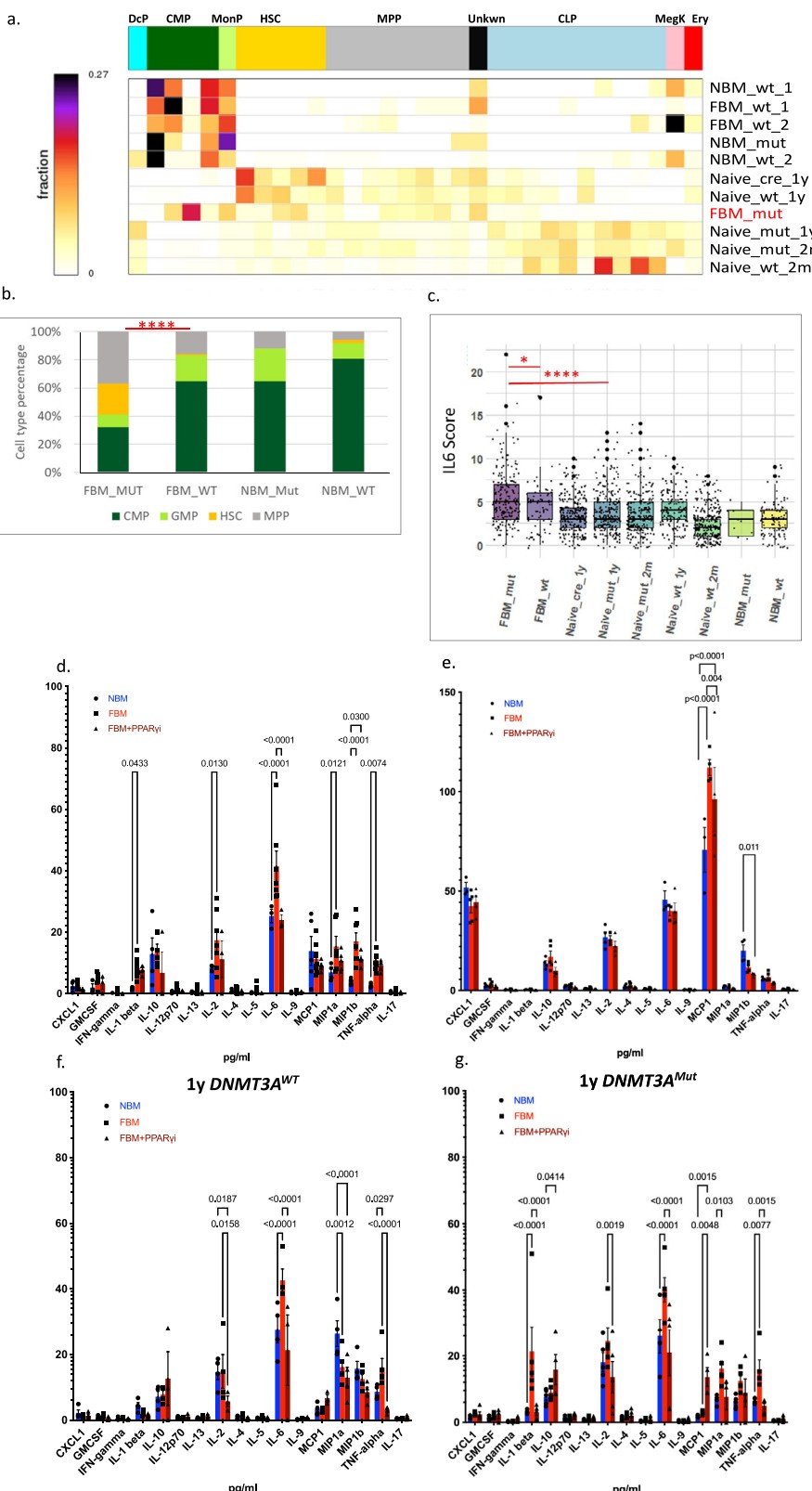

transplantation experiments provided evidence that anti-IL-6 neutralizing Abs can reduce the self-renewal advantage harbored by $DNMT3A^{Mut}$ preL-HSPCs under FBM conditions (Fig. 6b).

## Discussion

Our study provides a number of key insights into the contribution of FBM to the evolution of leukemia and CH. Our efforts established new models for the study of FBM interactions with both human and mice CH. We recognized that to date, there is no gold standard model that is specific on increasing the number of adipocytes. As we used external stress to increase FBM, it was crucial to use the PPARγi group to control for off-targets effects of the sublethal irradiation and castration on other cell types in the BM (stroma cells, endothelial cells etc.). We demonstrated that the administration of PPARγi after sublethal

**Fig. 4 | *DNMT3A*^Mut cells exposed to FBM maintain an HSC pool characterized by an inflammatory phenotype. a** Cells from *DNMT3A*^Mut and *DNMT3A*^WT were injected to mice (*n* = 3) with fatty bone marrow (FBM) and normal bone marrow (NBM). Three days after injection lin- KIT+ (LK) cells were isolated from mice bone marrow (BM) and underwent single-cell RNA-Seq analysis. MetaCell algorithm was used to assign different single cells to Metacells with unique gene programs and cell types[44]. Gold, hematopoietic stem cells (HSCs); dark green, common myeloid progenitors (CMP); light blue, common lymphoid progenitors, (CLP); cyan, dendritic progenitors (DcP); gray, multipotent progenitors (MPP); dark olivegreen, monocyte progenitors (MonoP); pink, megakaryocyte progenitors (MegK); red, erythroid progenitors (Ery); grey4 (Unknown). Conditions: normal bone marrow (NBM); wild-type (wt); fatty bone marrow (FBM); naive cells−are cells extracted directly from BM of respective mice without transplantation. cre is the cre control. **b** While all injected cells show a marked reduction in HSCs after transplantation *DNMT3A*^Mut cells exposed to FBM maintain their HSC pool which is significantly higher than all other condition (which were transplanted). Exact fisher test was used ****P < 0.0005 to compare proportions of HSCs between the groups. FBM_MUT vs. FBM_WT *P* = 2.3 e10^−9. **c** Ranked GSEA analysis on differentially expressed genes between *DNMT3A*^Mut cells exposed to FBM cluster and other clusters exposed significant enrichment of inflammatory pathways one of them was the IL-6, *JAK* and *STAT3* response gene set. An expression score for every single cell was calculated based on the expression of each of the genes in the IL-6 gene set. All comparisons were performed using a two-tailed, nonpaired, nonparametric Wilcoxon

rank-sum test with 95% confidence interval with FDR multiple hypothesis. *P < 0.05, ****P < 0.0005. FBM_mut vs. FBM_wt *P* = 0.01281, FBM_mut vs. Naive_mut_1y *P* = 2.18E-09. **d** Multiplex cytokines assay (FirePlex-96 Key Cytokines (Mouse) Immunoassay Panel (ab235656)) of 17 common cytokines analyzed by FACS-based multiplex method of serum from NBM, FBM and following PPARγi administration to NSG mice, without any cell's transplantation. Data were analyzed by two-way ANOVA test−Sidaks multiple comparison test. The figure displays the *P* values. NBM *n* = 4, 3, 3, 4, 4, 4, 4, 3, 4, 4, 3, 4, 6, 4, 5, 4, 6. FBM *n* = 9, 7, 6, 9, 9, 6, 6, 7, 7, 8, 8, 7, 9, 9, 6, 7, 9, 8. FBM+PPARγi *n* = 5, 3, 3, 5, 3, 3, 3, 3, 3, 3, 3, 5, 5, 5, 5, 3. **e** Multiplex cytokines assay (FirePlex-96 Key Cytokines (Mouse) Immunoassay Panel (ab235656)) of 17 common cytokines analyzed by FACS-based multiplex method of BM from NBM, FBM and following PPARγi administration to NSG mice, without any cell's transplantation. Data were analyzed by two-way ANOVA test−Sidaks multiple comparison test. The figure displays the *P* values. NBM *n* = 3, 4, 4, 4, 4, 4, 4, 4, 4, 4, 3, 4, 3, 4, 4, 4, 4. FBM *n* = 5, 5, 5, 5, 5, 5, 3, 5, 5, 3, 4, 4, 4, 5, 5, 4. FBM+PPARγi *n* = 4 for all. **f, g** FACS-based multiplex method of NBM, FBM, and following PPARγi administration to NSG mice transplanted with 1-year-old *DNMT3A*^Mut or *DNMT3A*^WT BM-derived cells. Each bar represents 4 to 5 mice. Data were analyzed by two-way ANOVA test−Sidaks multiple comparison test. The figure displays the *P* values. **f** NBM *n* = 4 for all. FBM *n* = 4, 4, 4, 4, 4, 4, 4, 4, 5, 4, 5, 5, 5, 5, 4. FBM+PPARγi *n* = 4, 3, 3, 4, 3, 3, 3, 3, 3, 3, 3, 3, 4, 4, 4, 3. **g** NBM *n* = 5 for all, except for IL-6 *n* = 4. FBM *n* = 5 for all. FBM+PPARγi *n* = 5, 3, 3, 5, 4, 3, 5, 5, 5, 5, 5, 4, 3, 3, 5, 5. Source data are provided as a Source Data file.

irradiation resulted in a significant reduction of adipocytes (Fig. 1). We provide evidence for the contribution of andropenia to FBM accumulation (Supplementary Fig. 1c). In addition, we provide evidence that the interaction of FBM with human preL-HSPCs carrying *DNMT3A* mutations provide them with selective advantage (Fig. 2) and can increase the self-renewal capacity of mice preL-HSPCs carrying *DNMT3A* mutations (Figs. 3 and 4a). Accordingly, FBM might contribute to the expansion of preL-HSPCs over time and to the evolution of CH. We have demonstrated in the past that large CH clones have increased risk for evolving to AML[48], thus the increasing in clone size under FBM conditions can promote AML evolution. Furthermore, paracrine inflammatory signals from the FBM microenvironment (in the form of IL-6), can activate the IL-6 pathway in preL-HSPCs (Fig. 4). We provide evidence that the addition of IL-6 to the CFU assay increase the clonogenic capacity of preL-HSPCs. Furthermore, *DNMT3A* mutated mice and human HSPCs have a selective advantage if co-cultured in vitro with bone marrow-derived adipocytes which also secret IL-6 (Fig. 5). The activation of the IL-6 pathway contributes to mice preL-HSPCs expansion and can be prevented by the administration of IL-6 neutralizing Abs (Fig. 6).

The contribution of the environment to the evolution of CH has been reported in the past. Our study correlates aging and micro-environmental changes. Smoking and chemotherapy have been suggested to shape the fitness of *ASXL1, TP53* and *PPM1D* mutations[49]. Male gender was associated with splicing mutations and *ASXL1*[50,51]. Individuals with aplastic anemia (who carry oligo-clonal T-cell attacking their BM) presented with increased rate of *BCOR* and *BCORL1* mutations[52]. Chronic infection with HIV was associated with several pLMs[53] and hyperglycemia contributed to the evolution of *TET2* preL-HSPCs[54]. *DNMT3A* mutations had a selective advantage if exposed to chronic infection with mycobacterium avium[55]. While interactions between such pathological states and pLMs can explain some of the cases of CH evolution, it remains unclear why CH is so common among healthy elderly individuals[56,57]. Our discovery that the most common pLMs (*DNMT3A*) gain selective advantage when exposed to FBM can partially explain the increased prevalence of *DNMT3A* mutations with aging.

The accumulation of FBM with age is ubiquitous, however, large variability exists in its extent[58]. Epidemiological studies suggest that several factors can explain this high variability including: the age-related decline in renal function[59], increased body mass index (BMI)[60], andropenia and menopause[61]. Interestingly, FBM continues to increase steadily in males, while in females it increases dramatically following

menopause[58]. This phenomenon might be related to the rapid estrogen deficiency during menopause as oppose to the gradual decrease of testosterone in males. Previous studies demonstrated enrichment of *DNMT3A* mutations among females[51]. The sharp increase in FBM during menopause could suggest that it is the dynamics of FBM accumulation that shape CH rather than the actual FBM mass. Interestingly, a recent report suggested that *DNMT3A* mutations were significantly associated with premature menopause[62].

While the interaction between FBM and *DNMT3A* mutations can explain a large portion of CH evolution, it remains unclear whether FBM can provide selective advantage to other pLMs. In this study, we tested a specific *SRSF2* mutated mice model and could not observe increased self-renewal after exposure to FBM (Supplementary Fig. 6). While these results are valid, they are limited to the model we used. One attractive pLM candidate to be influenced by FBM are the *TET2* mutations. Recent studies suggest that *TET2* knockout mice which developed a myeloid malignancy had higher levels of IL-6 when exposed to microbiota[63]. Altogether, the contribution of FBM to the fitness of other common pLMs needs to be further explored. However, we believe that for future studies more FBM models should be established. We propose that an optimal humanized NSG mice that will include human IL-6 and other humanized cytokines together with a conditional activation of PPARγ in MSCs.

The interaction between IL-6 and preL-HSPCs in our study (both in vitro Fig. 5 and in vivo Fig. 6) might have clinical relevance as anti-IL-6 treatments are available and can be used to control *DNMT3A*-driven CH. However, the role of higher IL-6 levels in the improved engraftment of human preL-HSPCs remains unclear, as murine IL-6 does not cross-react with the human IL-6 receptor[64]. Future human in vitro studies and humanized mice expressing IL-6 are needed. Despite these limitations, we still observed increased engraftment of human preL-HSPCs under FBM conditions (Fig. 2). This could be explained by the interaction of human preL-HSPCs with other cytokines. Altogether, the FBM most probably provide several inflammatory signals which can stimulate human preL-HSPCs, and future studies are needed to provide evidence for this hypothesis.

Based on the results presented here, it is becoming clear that FBM is more than just hypocellular marrow, and that it can shape CH evolution and contribute to other adverse metabolic effects. More research could be directed toward the prevention of FBM accumulation and its interaction with other mutations and with human HSCs.

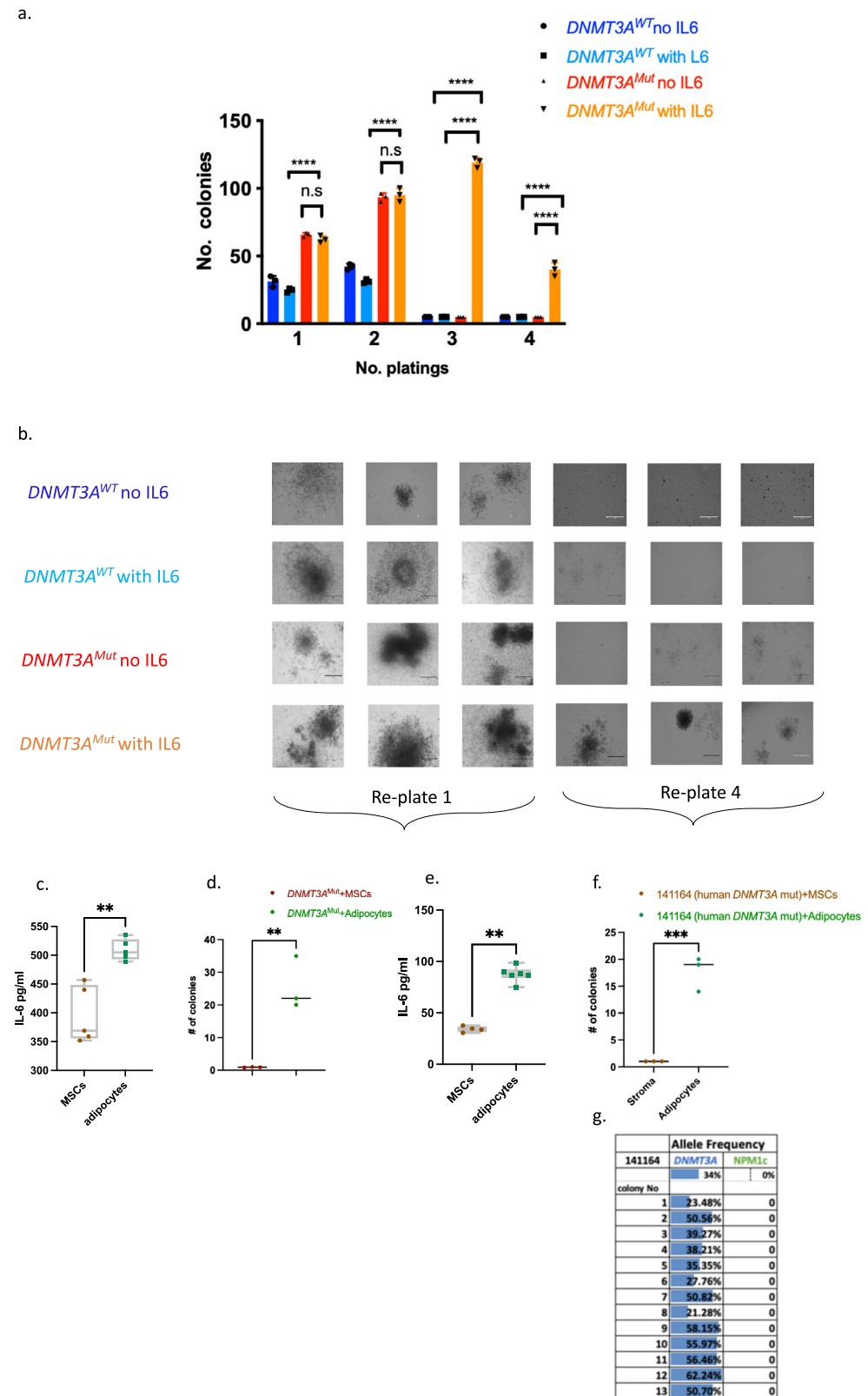

With the ultimate goal of correcting the very first steps in leukemia evolution and aging.

## Methods

### Mice

In all, 8–12 weeks male immune-deficient NSG (NOD/SCID/IL-2Rgc-null) mice: NSG (Stock No: 005557) (The Jackson Laboratory, Bar-Harbor, ME, USA). NOD.Cg-*Prkdc^scid^ Il2rg^tm1Wjl^* Tg(PGK1-KITLG*220) 441Daw/SzJ mice (stock No: 017830) (The Jackson Laboratory, Bar-Harbor, ME, USA). NOD.Cg-*Prkdc^scid^ Il2rg^tm1Wjl^* Tg(CMV-IL-3,CSF2,KITLG)1Eav/MloySzJ mice (Stock No.: 013062) NSG-SGM3(The Jackson Laboratory, Bar-Harbor, ME, USA) *DNMT3A^R882H^* KI mice, constitutively express the human *DNMT3A* mutation. *SRSF2^P95H^* floxed mice possess *loxP* sites flanking the endogenous

**Fig. 5 | Selective advantage to *DNMT3A*<sup>Mut</sup> BM-derived cells under methylcellulose colony assay. a** Number of colonies in methylcellulose (MethoCult M3334) colony-forming-unit assays of *DNMT3A*<sup>Mut</sup> and *DNMT3A*<sup>WT</sup> BM-derived cells. Data was analyzed by two-way ANOVA test−Sidaks multiple comparison test. Data are presented as mean values ± s.d. *n* = 3 biologically independent experiments. ****$P < 0.00001$. **b** Representative photograph of the methylcellulose plating from (**a**). Scale bar: 100 μM. **c–f** All comparisons were performed using a two-tailed, nonpaired, nonparametric Mann−Whitney test compare ranks. Data presented as box and whiskers min to max. **c** Mouse-derived MSCs/adipocytes were cultured for 10 days. Then, media was collected and IL-6 levels were analyzed by ELISA. **$P < 0.005$. MSCs *n* = 5, adipocytes *n* = 5, MSCs vs. adipocytes $P = 0.0039$. **d** One-year *DNMT3A*<sup>Mut</sup> BM-derived Lin⁻ cells were co-cultured with adipocytes/MSCs. After 10 days, colony-forming unit (CFU) assay was performed, in which cells were

cultured in methylcellulose with adipocytes or MSCs-derived media. **$P < 0.005$. *DNMT3A*<sup>Mut</sup>+MSCs *n* = 3, *DNMT3A*<sup>Mut</sup>+adipocytes *n* = 3, *DNMT3A*<sup>Mut</sup>+MSCs vs. *DNMT3A*<sup>Mut</sup>+adipocytes $P = 0.0031$. **e** Human-derived MSCs/adipocytes were cultured for 10 days. Then media was collected and IL-6 levels were analyzed by ELISA. **$P < 0.005$. MSCs *n* = 4, adipocytes *n* = 6, MSCs vs. adipocytes $P = 0.0048$. **f** Human AML CD34+ cells with a *DNMT3A* mutation (sample #141164) were co-cultured with adipocytes/MSCs. After 10 days, colony-forming unit (CFU) assay was performed, in which cells were cultured in methylcellulose with adipocytes or MSCs-derived media. ***$P < 0.0005$. 141164 (human *DNMT3A* Mut)+MSCs *n* = 3, 141164 (human *DNMT3A* Mut)+adipocytes *n* = 3. 141164 (human *DNMT3A* Mut)+MSCs vs. 141164 (human *DNMT3A* Mut)+adipocytes $P = 0.0004$. **g** Variant allele frequency (VAF) analysis of *DNMT3A*. Source data are provided as a Source Data file.

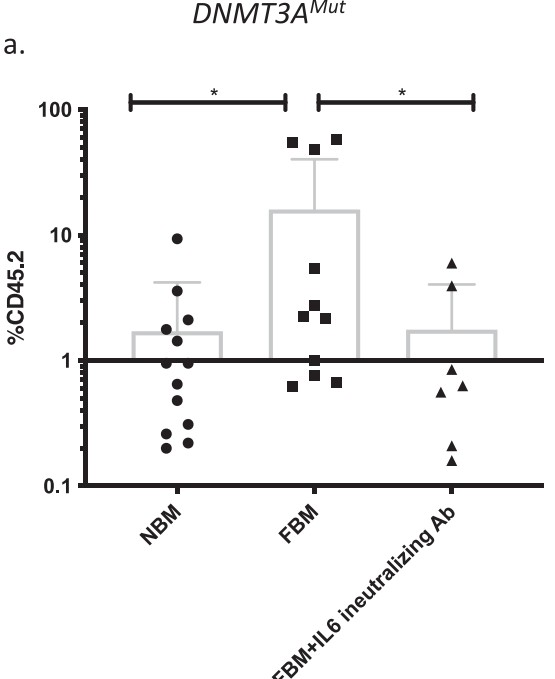
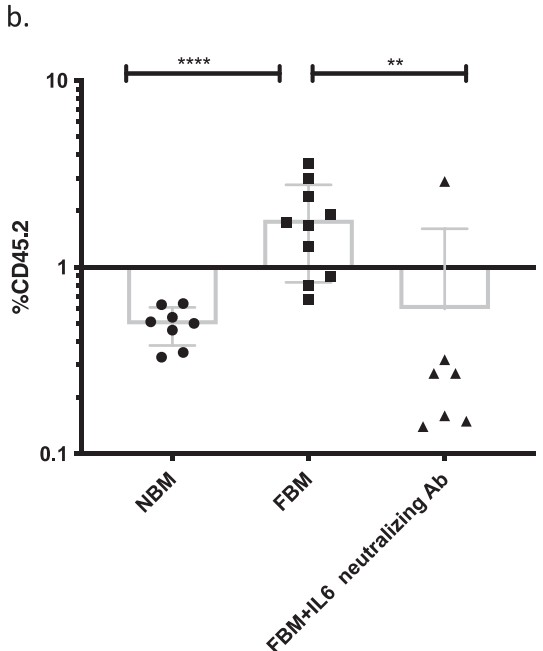

**Fig. 6 | In vivo treatment with neutralizing IL-6 antibodies. a** Significant engraftment decrease of *DNMT3A*<sup>Mut</sup> BM-derived cells in NSG mice following administration of neutralizing IL-6 Ab. In all, $6 \times 10^6$ *DNMT3A*<sup>Mut</sup> BM-derived cells were transplanted to normal bone marrow (NBM) (*n* = 13), and fatty bone marrow (FBM) (*n* = 11) and FBM NSG mice (*n* = 7) treated with neutralizing IL-6 Ab (Clone: MP5-20F3, Ultra-LEAF™ Purified anti-mouse IL-6 Antibody, BioLegend 504512). Neutralizing IL-6 Ab (50 μg/mouse) was administrated to NSG mice intraperitoneal one day before cell transplantation and 10 days after transplantation. NBM vs. FBM $P = 0.0145$, FBM vs. FBM+ neutralizing IL-6 Ab $P = 0.0427$. **b** Secondary

transplantation of cells from (**a**). To FBM NSG mice (*n* = 8, *n* = 10, *n* = 7, respectively). *DNMT3A*<sup>Mut</sup> BM-derived cells do not perform any self-renewal following treatment with neutralizing IL-6 Ab. NBM vs. FBM $P < 0.00001$, FBM vs. FBM+ neutralizing IL-6 Ab $P = 0.0064$. Data are presented as mean values ± SD. *$P < 0.05$, **$P < 0.005$, ***$P < 0.0005$, ****$P < 0.00005$. Each dot represents a mouse. Data in this figure presented as box and whiskers min to max. All comparisons were performed using a two-tailed, nonpaired, nonparametric Mann−Whitney test compare ranks. Source data are provided as a Source Data file.

coding region of the serine/arginine-rich splicing factor 2 (*SRSF2*) gene (Stock No: 028376) (The Jackson Laboratory, Bar-Harbor, ME, USA). *DNMT3A*<sup>R882H</sup> or *SRSF2*<sup>P95H</sup> were crossed with VAV Cre (Stock No: 008610) (The Jackson Laboratory, Bar-Harbor, ME, USA). All experiments were performed in accordance with institutional guidelines approved by the Weizmann Institute of Science Animal Care Committee (02630418-4).

### Patient samples
All human samples were collected, Ficoll separated and viably frozen with informed consent according to procedures approved by Rambam Health Care Campus, Haifa, Israel IRB # 0280-09-RMB and the university health network (UHN) IRB # 01-0573. All samples were sequenced with our in house clonal hematopoiesis panel[65]. Information on all samples can be found in Supplementary Table 3.

### CD3 depletion
CD3 cells were isolated from thawed human samples (peripheral blood AML sample, mobilized peripheral blood mononuclear cells (PBMCs) and cord blood) using magnetic beads according to the manufacturer's protocol (EasySep™ Human CD3 Positive Selection Kit II, StemCell Technologies, Vancouver, Canada).

### Xenotransplantation assays
PBMCs from AML patients were CD3 depleted as described above. Mobilized PBMCs or cord blood were enriched for CD34⁺ cells using magnetic beads according to the manufacturer's protocol (CD34 MicroBead Kit, Miltenyi Biotec, Bergisch Gladbach, Germany). CD3 depletion and CD34 enrichment were validated by flow cytometry unless specified otherwise, $1–2.5 \times 10^6$ CD3-depleted mononuclear cells were injected intra-femoral (right femur) into 8–12-week-old male

mice. According to approval by the Weizmann Institute of Science Animal Care Committee (02630418-4): the side effects from leukemia are reduced by the early euthanasia. Usually, mice recovered immediately following intrafemur injection. In case animals not recovering within 24 h, they will be euthanized. Endpoint and criteria for early withdrawal: lower limb paralysis from leukemia involvement of pelvis bones.

### Flow cytometry

Primary human samples and cells extracted from mice bone marrow were stained with the antibodies shown in Supplementary Table 4.

Primary mouse samples and cells extracted from mice bone marrow were stained with the antibodies shown in Supplementary Table 4.

All flow cytometry analyses were performed on FirePlex (Beckman Coulter, Brea, CA, USA), using CytExpert software v 2.4.0.28 (Beckman Coulter, Brea, CA, USA).

### BM transplantation

Freshly dissected femora and tibiae were isolated from 2-months-old or 1-year mice $DNMT3A^{Mut}$, $DNMT3A^{haplo}$, $DNMT3A^{WT}$, $SRSF2^{Mut}$ or control $SRSF2^{WT}$ mice CD45.2. BM was flushed with a 1cc (23G) into IMDM (Iscove's Modified Dulbecco's Medium). The BM was spun at $0.3 \times g$ by centrifugation, and RBCs were lysed in ammonium chloride-potassium bicarbonate lysis buffer for 1 min. After centrifugation, cells were resuspended in PBS, passed through a cell strainer, and counted. Then $6 \times 10^6$ cells were injected intrafemur into NSG (CD45.1) mice that were irradiated (FBM) 7 days before with low-dose irradiation (225 rad) or to non-irradiated (CONTROL) NSG mice. Eight weeks following cells transfer, mice were sacrificed. Right femur and the other bones (left femur and tibias) were cut and BM cells were flushed with IMDM (Iscove's Modified Dulbecco's Medium) and analyzed by FACS. Engraftment was defined by the presence of mCD45.2. Engraftment was assessed according to the presence of ≥0.1% mCD45.2 cells. Ab's that were used: APC anti-mouse CD45.2 (Biolegend, clone 104), PE anti-mouse CD45.1 (Biolegend clone A20).

### BADGE administration

NSG mice were treated intraperitoneally with the PPARγi (which is a critical transcription factor in adipogenesis) inhibitor, bisphenol ADi-Glycidyl Ether (BADGE) (30 mg/kg) (Sigma, cat 15138) for 7 days, irradiated and treated for more 7 days following cells transplantation.

### IL-6 neutralization in vivo

NSG mice were administrated intraperitoneally with IL-6 (50 µg/mouse) neutralizing Ab (BioLegend 504512) for 2 days. Then mice were irradiated with low dose (225 rad), and treated intraperitoneally with a IL-6 neutralizing Ab for 7 days, followed by $DNMT3A^{Mut}$ or $DNMT3A^{WT}$ BM-derived cells transplantation.

### Histological analysis and adipocytes quantification

Mice were sacrificed and autopsied, and dissected tissue samples were fixed for 24 h in 4% paraformaldehyde, dehydrated, and embedded in paraffin. Paraffin blocks were sectioned at 4 mm and stained with H&E. Images were scanned by Pannoramic SCAN II (3DHISTECH, Hungary). BM adipocytes were quantified by intracellular staining of the FBM lipid with LipidTox (a fluorescent dye that stains neutral lipids; Life Technologies) and analyzed using ImageStream X Mark II, Luminex.

### Image stream analysis

To quantify the number of adipocytes in mice, we needed to sacrifice three mice per sample. All bones (femur and tibia) were cut, and BM cells were flushed with IMDM (Iscove's Modified Dulbecco's Medium). Furthermore, the bones were crushed in order to obtain the adipocytes attached to the bones. The BM was filtered through 30-µm mesh,

centrifuged and 1 ml PBS×1 and 1 ml 8% PFA were added. The sample was vortex, and PBS×1 was added. Next, the sample was centrifuged and fixed with 200 μL designated fixative (00-5223-56 and 00-5123-43, eBioscience) and incubated for 30 min at 4 °C (in the dark). Next, the cells were washed with 1 ml designated permeabilization buffer (00-8333-56, eBioscience), centrifuged, and stained with 1:100 AB (PE anti-mouse CD45 Antibody BLG-103106) overnight spinning in the fridge. The next day, the sample was centrifuged at max speed 30 s and washed with PBS×1 twice, and stained with 1 ml DAPI (dilute 1:1000 in PBS) 7 min in the ice. To quantify the number adipocytes, we used HCS LipidTOX™ Deep Red neutral lipid stain (H34477, Thermofisher). The LipidTOX™ neutral lipid stain has an extremely high affinity for neutral lipid droplets and can be detected by fluorescence microscopy or an HCS reader. The sample was centrifuged, then 40 µl PBS×1 were added, and 1:50 LipidTOX was added to each sample[66]. The samples were quantified by ImageStream X Mark II, Luminex, and analyzed by ImageStream software.

### Whole-mount immunofluorescence staining

The procedure was performed as described previously in ref. [67]. Following antibodies were used: Goat anti-mouse/rat FABP4/A-FABP Antibody, R&D Systems, Cat#AF1443-SP; RRID: AB_2102444 and Alexa Fluor® 647 AffiniPure Donkey Anti-Goat IgG (H+L), Jackson ImmunoResearch, Cat#705-605-003; RRID: AB_2340436, RabbitPerilipin-1 (D418) Antibody, CellSignaling Technology Cat# 3470, RRID:AB_2167268, AB_2340436, and Alexa Fluor®488 Donkey Anti-RabbitCat#711-545-152,RRID:AB_2313584, Jackson ImmunoResearch.

For evaluation, fluorescently labeled bone tissues were placed onto a four-well-μ-Slide and covered in antifade or PBS to prevent tissue desiccation. The preparations were examined using a Leica TCS SP8 confocal microscope and analyzed with the image analysis software Volocity (v6.2, Perkin Elmer) and ImageJ. In addition, bone matrix and adipocytes were detected using the DIC (TLD) mode of the microscope.

Z-stacked confocal images are generated from whole mounts of bisected mouse bones in which the structural and cellular integrity is highly preserved, and they show epiphyseal/metaphyseal BM regions (see Supplementary Fig 2a, zoom-ins). Additional markers for adipocytes, vasculature, and bone are not necessary[67,68]. FABP4 is also expressed by endothelial cells; however, adipocytes have higher expression. The DIC (differential interference contrast) channel was additionally used for the detection of the typical unilocular morphology of the adipocytes (see Image S2a) and other structures. Indeed, in BM sections big sinusoidal vessels are often collapsing, however here we used protective whole mounts and the "empty" spaces here are trabecular bone structures (without surrounding FABP4 cells, see Image S2c), which are very frequently present in epiphyseal/metaphyseal BM. The sinusoidal vessels, with specific morphology, are clearly visible by lower FABP4 expression, surrounded by FABP4 low endothelial cells, and in the DIC channel bone structures and sinusoidal vessels are reflected differently (see Supplementary Fig. 2b, c).

### Colony-forming unit assay

$DNMT3A^{Mut}$ or $DNMT3A^{WT}$ mice were sacrificed, all bones (femur and tibia) were cut, and BM cells were flushed with IMDM (Iscove's Modified Dulbecco's Medium), the cells where count and seeded at a density of $2 \times 10^4$ cells per replicate into cytokine-supplemented methylcellulose medium (MethoCult M3334, Stemcell Technologies). After 10–14 days, the colonies propagated and scored. The remaining cells were resuspended and counted, and a portion was taken for replating ($2 \times 10^4$ cells per replicate) with human (GenScript Z03034-50) or mouse (GenScript Z02767-10) IL-6 or w/o.

### Human MSCs

Primary bone marrow-derived human MSCs were generously received from RAMBAM hospital. The cells were thawed and suspended with a

suitable medium that contains MEMα, 1% P/S, 1% ʟ-glu, 10% FBS (MSCs media).

## Human adipocytes differentiation in vitro

The primary human MSCs were seeded on 96-well plates with MSCs medium at $10^4$ cells/well. After 2 days, when reached confluence, the medium was changed with MesenCult Adipogenic Diff Kit, Human (stem cell, cat#05412) for 10–14 days until the MSCs differentiated into mature adipocytes.

## Mouse MSCs production

MSCs were generated from the tibia and femur of two-month-old C57BL/6 mice. The bones were flushed, and cells were seeded in MSC media in a six-well plate. Every other day, the medium was replaced.

## Mouse adipocytes differentiation in vitro

The primary mouse MSCs were seeded on 96-well plates at $10^4$ cells per well with MSCs cell media. After two days, the cells reached full confluence, and the medium was replaced with MesenCult Adipogenic Diff Kit, mouse (stem cell, cat#05507) for 10–14 days, until the MSCs differentiated into mature adipocytes.

## Oil Red Staining

Cells were washed twice with PBSX1, fixed with 4% PFA and washed twice with distilled water. Oil red was added for an hour.

## Amplicon sequencing

We used an amplicon-based approach to sequence *DNMT3A* and *NPM1c* from human samples after and before engraftment.

*DNMT3A* Fw primer for amplicon sequencing:
CTACACGACGCTCTTCCGATCTttgtttgtttgtttaactttgtg
*DNMT3A* Rev for amplicon sequencing:
CAGACGTGTGCTCTTCCGATCTcactatactgacgtctccaacat
*NPM1c* Fw for amplicon sequencing:
CTACACGACGCTCTTCCGATCTgttgaactatgcaaagagacatt
*NPM1c* Rev for amplicon sequencing primer:
CAGACGTGTGCTCTTCCGATCTagaaatgaaataagacggaaaat.

## Enrichment for mouse adipocytes for single-nuclei sequencing

All bones (femur and tibia) were cut in both ends, put in a 0.5-mL microcentrifuge tube with the bottom cut off, and put into a 1.5-mL microcentrifuge tube. A brief centrifuge (from 0 to 11,200×*g*, 9 s, room temperature (RT)) was used to separate the fresh bone marrow. Red blood cells (RBC) lysing buffer (Sigma) was used to lyse RBCs. Floating cells were extracted from the top layer and repeatedly washed with PBS following centrifugation (1000×*g*, 5 min, RT).

## Single RNA-seq

Cells from 2-month-old and 1-year-old *DNMT3A*^Mut or *DNMT3A*^WT were injected to FBM and normal mice. Three days after injection, CD45.2 LSK cells were isolated. We also isolated cells from the same donor mice before they were injected and termed them naive cells. From all the above conditions LSK cells were isolated and single-cell sorted. For sorting, we used the following antibodies: CD45.2-APC, anti-mouse (BLG), Ly-6G/Ly-6C (Gr-1) FITC anti-mouse (BLG), CD11b FITC anti-mouse/human (BLG), CD45R/B220 FITC anti-mouse/human (BLG), CD4 FITC anti-mouse (BLG), TER-119 FITC anti-mouse (BLG), CD8a FITC anti-mouse (BLG), Sca-1-PE-Vio770 anti-mouse (Miltenyi), CD117 (c-kit)-BV605 anti-mouse (BLG). FACs analysis for LSK sorting is described in Supplementary Fig. 14a. Cells were sorted to 384 low-binding plates and sequenced according to MARS-seq protocol[42]. A total of 3923 cells were available for analysis after mapping. To filter empty cells and doublets we selected single cells with more than 450 UMIs per cells, and excluded all cells in the 3% upper percentile of UMI counts. After filtering cells, a total of 2198

cells were available for further analysis. We then used the Metacell library for noise reduction, clustering and cell type annotation[43,44]. Removal of lateral effects (cell-cycle, stress) and batch effects was performed using gene module analysis to filter genes used for Metacells grouping. The Metacell analysis partitioned 1999 cells to 32 Metacells, while filtering 199 cells as outliers. We annotated Metacells cell types based on known genes defining cell populations[45]. The following genes were used: HSCs (*Procr*); MLP (*Dntt*); CMP (*Mpo*), MegK (Pf4); ERY (*Hba-a2*); MonP (Irf8), DC (Cd74), MPP (Fgd5 and no other conditions).

We have used another method for single-cell clustering reduction of dimension and clustering and differential expression analysis based on the UMAP algorithm. After filtration of cells in the same way we did for the Metacell analysis no batch effects could be noticed in the ERCC counts between the different conditions. We have used the clustering of the UMAP data to perform differential expression (DE) analysis on the different clusters (Supplementary Data 1).

We have repeated the scRNA-seq experiment for validation of our results. One-year-old *DNMT3A*^Mut cells were injected to FBM and NBM mice. Then, 3 days after injection CD45.2 LSK cells were isolated. Cells were sorted to low-binding Eppendorf tubes and library was prepared by the 10X genomics V3 3' kit. Libraries were sequenced by Novaseq. For the *DNMT3A*^Mut FBM condition we analyzed 1228 cells with median coverage of 60,000 reads per cell and 9440 UMIs per cells. For the *DNMT3A*^Mut NBM condition, we analyzed 1405 cells with median coverage of 195,000 reads per cell and 9469 UMIs per cells. We have performed the same bioinformatics analysis as in the first experiment.

To gain more insights into the characteristics of adipocytes under our FBM irradiation model. we have used single-nuclei RNA sequencing (snRNA-seq) due to the challenging fragility of lipid-filled adipocytes. We have adopted a protocol developed by others[29]. The protocol combines an adipocyte enrichment protocol (described above) followed by nuclei "cleanup" step by sorting. Nuclei were sorted into low-binding Eppendorf tubes, and the library was prepared by the 10X genomics V3 3' kit. Libraries were sequenced by Novaseq. We have used the Metacell platform to analyze the cells. After filtering empty cells and doublets, we analyzed 4058 cells from the irradiation FBM model and 3428 cells from the BADGE control group. Removal of lateral effects (cell-cycle, stress) was performed using gene module analysis to filter genes used for Metacells grouping, and cells with high expression of mitochondrial genes were excluded. We have combined our dataset together with that of Baccin et al. and have annotated cells using key gene markers described in ref. 30. The Metacell analysis partitioned cells to 181 Metacells. Next, we compared the nuclei we sequenced to adipocytes from ref. 30. The majority of the cells in our cohort were monocyte/neutrophils and late erythroid cells, which had similar gene expression compared to the same populations in Baccin et al. We identified one Metacell and 25 cells in our cohort which clustered together with nine adipocytes from ref. 30, these cells had similar gene expression.

## GSEA analysis

The DE genes of cluster 1 (Supplementary Data 1) were ranked based on fold change and analyzed using the GSEA software version 4.1.0[69]. Significant genes set had FDR *q*-val < 0.2. The Hallmark genes sets were used for the analysis.

## Gene sets scores (IL-6, TNF-α, IFN-α, and IFN-γ)

To generate scores for the different gene sets across cells, we down-sampled the original umi matrix to 750 umis, and calculated the score per cell as the sum of all genes in the respective gene sets (Supplementary Data 3 and 4). These scores per cell were used to generate the plots per experimental condition in Fig. 4c and Supplementary Fig. 12.

## Statistical analysis

In all figures and tables comparison between medians was performed using the Mann–Whitney $U$ test with FDR correction for multiple hypothesis testing. Calculations were performed using the compare_means function of the R library ggpubr. The comparison between cytokine levels was analyzed by a two-way ANOVA test with Sidaks multiple comparison correction.

## Reporting summary

Further information on research design is available in the Nature Portfolio Reporting Summary linked to this article.

## Data availability

The sequencing data and processed data generated in this study have been deposited in the Array Express database (https://www.ebi.ac.uk/biostudies/arrayexpress) under accession codes: E-MTAB-12613, E-MTAB-12614, E-MTAB-12620. All relevant data are also available from the corresponding author upon reasonable request. Source data are provided with this paper.

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

## Acknowledgements

This research was supported by the EU horizon 2020 grant project MAMLE ID: 714731, LLS and rising tide foundation Grant ID: RTF6005-19, ISF-NSFC 2427/18, ISF-IPMP-Israel Precision Medicine Program 3165/19, ISF 1123/21, BIRAX 713023, the Ernest and Bonnie Beutler Research Program of Excellence in Genomic Medicine, awarded to LIS. LIS is an incumbent of the Ruth and Louis Leland career development chair. N.K. is an incumbent of the Applebaum Foundation Research Fellow Chair. This research was also supported by the Sagol Institute for Longevity Research, the Barry and Eleanore Reznik Family Cancer Research Fund, Steven B. Rubenstein Research Fund for Leukemia and Other Blood Disorders, the Rising Tide Foundation, and the Applebaum Foundation.

## Author contributions

N.Z. designed and developed the study, performed mice experiments, cells culture, targeted sequencing, and single-cell RNA-seq, and wrote the manuscript. A.A.B., N.C.I., and N.R. performed single-cell RNA analysis. S.A. and T.B. performed single-cell RNA-seq experiments. E.K. helped with FACs experiments. Z.P. helped with imageStream experiments. G.H. and M.S. performed whole-mount immune fluorescent staining of BM. M.S. and T.C.M. developed DNMT3Amut mice model, D.L. helped with the analysis of methylation data (not presented). E.S. developed the FBM castration model. M.M. contributed clinical samples. N.Z. and N.K. Performed xenotransplantation experiments, cytokine experiments, L.I.S. and N.K. designed and supervised the study and wrote the manuscript.

## Competing interests

L.I.S. is a consultant to Metasight Isreal LTD; and to Sequentify Israel LTD. The remaining authors declare no other competing interests.

## Additional information

[1]Department of Molecular Cell Biology, Weizmann Institute of Science, Rehovot, Israel. [2]Department of Computer Science and Applied Mathematics, Weizmann Institute of Science, Rehovot, Israel. [3]Blavatnik School of Computer Science, Tel Aviv University, Tel Aviv, Israel. [4]Department of Biological Regulation, Weizmann Institute of Science, Rehovot, Israel. [5]Life Sciences Core Facilities, Weizmann Institute of Science, Rehovot, Israel. [6]Institute of Molecular Medicine Ulm University, Ulm, Germany. [7]Department of Medicine, Hematology, Oncology and Rheumatology, University Hospital Heidelberg, Heidelberg, Germany. [8]Department of Internal Medicine V, Heidelberg University Hospital, Heidelberg, Germany. [9]European Molecular Biology Laboratory (EMBL), Heidelberg, Germany. [10]German Cancer Consortium (DKTK) and German Cancer Research Center (DKFZ), Partner Site Heidelberg, Heidelberg, Germany. [11]IVF Unit, Galilee Medical Center, Nahariya, Israel. [12]Princess Margaret Cancer Centre, University Health Network (UHN), Toronto, ON, Canada. [13]Department of Medical Biophysics, University of Toronto, Toronto, ON, Canada. [14]Department of Medicine, University of Toronto, Toronto, ON, Canada. [15]Division of Medical Oncology and Hematology, University Health Network, Toronto, ON, Canada. [16]Division of Hematology, University Health Network, Toronto, ON, Canada. [17]Hematology and Bone Marrow Transplantation Institute Rambam Healthcare campus Haifa, Haifa, Israel. [18]Theses authors contributed equally: N. Kaushansky, Liran I. Shlush. ✉e-mail: liran.shlush@weizmann.ac.il

