## [Peer Review File · Nature Communications]

Inflammatory signals from fatty bone marrow supports DNMT3A driven clonal hematopoiesisREVIEWERS' COMMENTS:

Reviewer #1 (Remarks to the Author):

In this paper, Zioni N et al tested different mouse models to recapitulate the high percentage of adipocytes that are found in humans during aging. Most of their work is performed in one of the models (sublethal irradiation with and without an adipogenesis inhibitor as a control). They tested the hypothesis that the hematopoietic microenvironment (specifically adipocytes) plays a role in the growth advantage of cells harboring DNMT3A mutations. The authors used several models including human pre-leukemic stem cells, and mouse models that harbored different DNMT3A mutations. They also assessed the effect of intrinsic aging, transplanting cells of different ages. To my knowledge the paper is original and relevant to the field. Although there is an increasing number of papers that have been exploring the role of adipocytes in recent years, they have been focus on normal and leukemic hematopoiesis, but not in the interaction with pre-leukemic mutations. The assessment of different methods to study the role of adipocytes is very valuable to the field, as there is not current gold standard. The authors recognized that to date, there is no model that is specific on increasing the number of adipocytes, as other stromal and hematopoietic populations may be affected with castration and irradiation. The use of the PPAR gamma inhibitor was key to minimize such effect. The finding that adipocytes increased the engraftment and confer a growth advantage for DNMT3A cells is novel and will have an important impact in the literature. The role of IL-6 on this mechanism is a less strong finding. The methods used to quantify adipocytes needs to be revised. Overall, I consider that the paper is relevant and will be an important contribution to the field.

I consider that the results support most of their conclusions, however, I suggest revising the following points:

1.- Introduction:

Add stem cell factor (SCF) to the list of cytokines produced by the adipocytes (Zhou B et al. 2017, reference 7), and the role of adipocytes to support myeloid-erythroid differentiation (Boyd A et al. reference 14) that is mentioned, but not as evidence of adipocytes as positive regulators or hematopoiesis. I consider that the authors focus on the role of adipocytes as promoters of inflammation (line 68), and do not stress the fact that they also play a key role on the support normal hematopoietic stem and progenitor cells (HSPC). It should also be discussed if there is a possibility that in the mouse the adipocytes are more reactive than in humans, as their number is higher and it is increased with physiological aging, in contrast to the mice. The paper of Aguilar Navarro (89) does not have a reference number, and it mentions it was done in leukemic samples, but the paper is on human normal cells in aging and do not include AML samples.

2.- The assessment of adipocytes needs to be more detailed and include more information:

a) The assessment of adipocytes was done at one week of irradiation, however there is no data on the content of adipocytes at end point of the experiment (8 weeks). On my experience (data not published) the increased-on adipocytes and sinusoids is transitory, so I consider it will be very helpful to know the content at end point. If it is transitory, the model is still valid and important, but need to be taken into consideration.

b) Figure 1e represents the amount of lipidTOX stain but is not fully described in methods, the information is not enough to repeat the experiment. I suggest adding a reference of this method if it was used in another publication. It is not clear about how LipidTOX was quantified. If it was in situ, they need to include representative images. They have included in methods, the Image Stream analysis to quantify adipocytes. The method quantifies adipocytes on flushed samples, but the adipocytes will unlikely be at this fraction. To obtained stromal cells, the bones need to be crushed. So, it is important that the authors show the method in detailed. This is relevant because the use of FABP4 as a marker for adipocytes is not specific. FABP4 also stains some sinusoids (which are evident in the right panel of non irradiated mice on figure 1f). It is very important to validate the model with at least two markers. Mouse sinusoids are larger and more open that their human counterpart, that usually look collapsed in bone marrow biopsies, so it is hard to assessed them by histology only in both species for different reasons. The use of LpidTOX is adequate, however it is not clear how it was used. I strongly suggest adding another adipocyte maker (for example perilipin) or quantitate sinusoids (VEGFR3) to rule out their participation. Figure 1f need to be more detailed in figure legend and specify what are each panel. I was not able to assess the bottom panels.

3.- I suggest revising the panel a on figure 4. Label the track on top of the heat map as

hematopoietic subpopulations, and add a reference of the heat map, what those colors represent? The supplemental figure on the single cell data is hard to read (Figure 5S). I advice adding a stain for MPO in situ by IHC to complement Figure 4.

4.- The evidence of IL-6 is not direct, as it is a cytokine produced by many cell types. IL-6 is a key cytokine that supports the growth of progenitor cells and is included in methocult assays to grow human cells and is an essential cytokine to expand HSC, so it can not be consider to have a more harmful role (line 426). To design a specific experiment to rule out their specific participation, it will need an approach as the one taken by Zhou, which conditionally deleted SCF from adipocytes, or co-cultured DNMT3A mut and wt cells with adipocytes or perform an ISH for IL-6 mRNA. I suggest avoiding any strong conclusions on the role of IL-6 secreted by adipocytes only. The experiments with the antibody are very informative, however it does not rule out the source of IL-6. The authors may benefit to include reference of increased levels of TNF alpha and IL-6 on myelodysplastic syndromes, specially by stromal cells, as MDS cells also harbored similar pre-leukemic mutations.

5.- I did not understand the context for the term durability (328).

Reviewer #2 (Remarks to the Author):

Zioni et al. present a study on the interesting topic of interactions between the BM microenvironment and genetically defined subclones of pre-leukemic disease. They convincingly show that paracrine signals in BM can preferentially favour the outgrowth of DNMT3A mutant hematopoietic cells, which potentially represents a pre-leukemic state. However, it is unclear what cells or signals are responsible for expanding DNMT3A mutant populations. The authors suggest that fatty BM provides a selective advantage to DNMT3A mutant HSPCs, but the support for this relationship is weak at best. Molecularly, the authors identify the IL-6 pathway as part of the mechanism that promotes DNMT3A mutant HSPC expansion, but later rationalize that IL-6 is unlikely to explain their results seen when using human pre-leukemic HSPCs. Because of the models used, this more realistically seems to be a paper about response to total body irradiation rather than BM adipocytes or IL-6. Furthermore, and unfortunately, this has less relevance to the natural evolution of leukemic disease, as exposure to high dose radiation or chemotherapy is not often encountered before progression to overt leukemia occurs.

Major conceptual issues:

1. Although the authors dedicate the majority of their Introduction to BM adipocyte biology and develop four different models of "fatty BM", there is no analysis of BM adipocytes shown beyond Figure 1. In fact, two of the "fatty BM" models are never used following their initial description, and the castration model was abandoned after Figure 2f, where a comparison to healthy DNMT3A wildtype transplantation was never shown. Throughout the manuscript, the authors rely almost exclusively on their "NBM" vs. "FBM" model, which represents non-irradiated mice vs. mice irradiated with 225 Rad one week prior. It is not possible to claim that any differences between these two models are driven by adipocytes.

Total body irradiation causes cytokine storm and dramatic remodeling of all components of the BM microenvironment including osteoblasts, megakaryocytes, and vasculature (Dominic et al., Blood 2009; Zhao et al., Nature Medicine 2014; Kenswil et al., Cell Reports 2018; Hooper et al., Cell Stem Cell 2009; Winkler et al., Nature Medicine 2012). Castration also causes systemic hormonal and physiological changes that are not limited to BM adipocytes. There are much more suitable models that could have been used to explore the independent role of BM adipocytes; e.g., rosiglitazone treatment (which the authors optimized), high fat diet, A-ZIP/F-1 "fatless" mice, or conditional adipocyte knockout models ("FAT-ATTAC" mice). It is unclear whether the BM even remains in a fatty state 9 weeks post-radiation at the time DNMT3A mutant vs DNMT3A wildtype engraftment was evaluated.

2. In addition to requiring more precise in vivo models, the most direct and controlled way to evaluate the effects of BM adipocytes would be to generate in vitro co-culture systems. It is not clear why the authors have not attempted these important experiments.

3. The authors themselves seem to lack confidence that their IL-6 findings have relevance to human samples. In their Discussion section, they explain that murine IL-6 does not cross-react with the human IL-6 receptor, which makes it difficult to interpret Figure 2. They suggest that TNF α levels may instead explain the interactions seen with human PreL-HSPCs but provide no data to support this suggestion. The central question of this paper revolves around the natural progression of clonal hematopoiesis to acute leukemia in human patients; however, the paper concludes with little clarity around the mechanism behind this process.

4. The authors should provide evidence that DNMT3A mutant models truly represent a pre-leukemic state. In their AML patient-derived xenograft model, the authors should show more comprehensive cell surface phenotyping to clarify whether the grafts have multi-lineage differentiation capacity vs. leukemic engraftment that would be exclusively myeloid. Furthermore, it is unclear whether the R882H Dnmt3a knock-in model has been functionally validated to accelerate disease progression upon the introduction of additional mutations, e.g., as has been shown with the R878H Dnmt3a mouse model (Loberg et al., Leukemia 2019).

Major experimental issues:

1. Given that FBM and NBM groups actually represent "irradiation" vs "no irradiation" groups, it is quite surprising that there is no difference in engraftment levels when healthy human CD34+ cells or DNMT3AWT murine cells are transplanted, even considering that transplantation was delayed by one week post-radiation. Peter Quesenberry's group has shown that low dose radiation increases the engraftment of healthy donor cells even when transplantation is delayed to 8 weeks post-radiation (Stewart et al., Blood, 2001). How do the authors explain this discrepancy? This is important to address, as the lack of increased DNMT3AWT cell engraftment in FBM mice forms the basis of the unique observations seen when DNMT3A^{mut} cells are transplanted.

2. The authors argue that technical barriers precluded their ability to test the functional relevance of IL-6 exposure in the context of human preL-HSPCs. They argue that secondary transplantation assays would not be feasible with their human PreL-HSPC samples, however they could still perform in vitro experiments and primary transplantation assays at a minimum. It is unclear why they could not treat with IL-6 and IL-6 neutralizing antibodies in vitro, followed by %DNMT3A VAF analysis and CFU assays. Similarly, intraperitoneal injections of human IL-6 could have been delivered in xenograft models.

3. In Figure 2a and 2d, why haven't the authors shown %DNMT3A VAF for both NBM and FBM conditions?

4. In Figure 3, the authors place a lot of emphasis on the age of the donor hematopoietic cells and rationalize that older PreL-HSPCs will have more hypomethylation. The Acknowledgments section indicates that methylation assays were performed, however this was never shown. Why wouldn't the authors include these data to contextualize their findings in Figure 3?

5. In Figure 3, the authors compared 1 year old preL-HPSCs to control conditions, but did not perform statistical analysis directly comparing 1 year old preL-HPSCs to 2 month old preL-HSCs. In order for the older preL-HPSCs to be considered more affected by FBM as hypothesized, a combined analysis needs to be performed.

5. In Figure 3, how do the authors explain the observation that wildtype cells have increased engraftment after BADGE treatment? Adipocyte levels are intermediate in BADGE condition compared to other experimental groups. This suggests effects of BADGE on hematopoietic cells could be independent from adipocyte content.

6. The connection between IL-6 secretions and BM adipocytes is lacking. Figure 4d-g is really an analysis of cytokine secretion in the presence vs. absence of total body irradiation. The analysis of the FBM + BADGE condition adds some value (Fig S8c,d), but this should have been evaluated in other fatty BM models that are independent of total body irradiation. It is not surprising that there

would be an increase in inflammatory cytokines after acute injury to the BM, and this may have nothing to do with adipocytes.

Minor issues:

1. The presentation of data related to the SRSF2 P95H model should be in the main figures given the amount of emphasis in the text.
2. The authors should explain their rationale for performing scRNAseq experiments at day 3 post-transplant, as this was not explained in the text.
2. The Discussion section is unnecessarily long, and some points are only tangentially related to the study (e.g., discussion of cardiovascular disease and heart failure, etc).

Reviewer #3 (Remarks to the Author):

Zioni et al report a compelling story about the expansion of DNMT3A-mutated cells in fatty bone marrow (FBM). First, the authors develop two immunodeficient (NSG) mouse models of FBM, which could be quite impactful. Then, xenograft assays and two DNMT3A loss-of-function mouse models indicate that DNMT3A-mutated cells expand in FBM conditions. The authors find increased inflammatory signaling and specifically a role for IL-6 in the expansion of DNMT3A-mutated cells. The findings are novel and important.

There is a general lack of precision. These are examples from most to least concerning:

- 1- The abstract mentions a 20-50 fold increase in DNMT3A^{mut}-preL-HSCs, and on page 8 they state "A 50 fold increase in HSCs was noted when comparing DNMT3A^{mut} cells exposed to FBM and DNMT3A^{WT} injected to NBM (Figure 4c, Table S4)." Figure 4c shows something unrelated. In Table S4, the values for DNMT3A^{mut} FBM HSCs and DNMT3A^{WT} NBM HSCs are 51/260=20% and 3/164=1.8%, respectively. This is an 11-fold difference, not 50.
- 2- Figure 4A is a white/red heatmap that is uninterpretable due to the absence of a legend for the color gradient. Furthermore, if the rows are clustered, a dendrogram should be shown. It may be more clear if the rows were not clustered but rather shown in a logical order.
- 3- "The maintenance of HSCs among DNMT3A^{mut} cells exposed to FBM was followed by expansion of myeloid progenitors as opposed to enrichment of lymphoid progenitors in the naïve LSK cells (Figure 4a,b)." – this is not clear from Figure 4a due to the previous point and not clear from Figure 4b which does not show lymphoid progenitors.
- 4- Figure legend referrals are erroneous, for example, Figure 3e is not cited in the text, and Figure 4e-h on line 308 should presumably be Figure 4d-g.
- 5- The authors consistently refer to INF α and INF γ but likely mean IFN α and IFN γ .

From the human experiments, the authors suggest "a role for an adipocyte-rich environment in enhancing engraftment of human preL-HSPCs, but not for normal HSPCs". However, they appear to use mononuclear cells for the AML/lymphoma samples and CD34⁺ cells for the normal cord blood. This may not be a fair comparison. More importantly, the VAF of DNMT3A mutations does not increase in the human grafts (Figure 2a, d), which may be interpreted as an internal control arguing against a specific advantage for DNMT3A-mutated human cells.

The visualizations and data to support cell type classifications are uninformative (Figure S5 and Table S1). The authors should generate gene signatures for each cell type, include these signatures as a supplementary table, and include dimensionality reduction (UMAP or tSNE) plots colored by the signature score.

Minor comments

Flow cytometry is limited to a few markers. Example gating schemes should be included in the supplement. It would have been nice if the authors had included more markers beyond human CD45 and mouse CD45.1/CD45.2. Do the mice develop lympho-myeloid grafts? Is the stem cell compartment proportionally expanded in the FBM / DNMT3A-mutated mice?

To profile cytokine secretion, the authors appear to have used the irradiation model. Can similar results be obtained using the castration model?

Figure S1b should be quantified similarly to Figure 1e.

Please clarify if the GSEA analyses in the section on page 9 were performed on all cells or specifically on HSC clusters. If the analyses were done on all cells, please also include the HSC-specific comparisons.

What is "unselected" in Figure S6A?

Reviewer #4 (Remarks to the Author):

In their manuscript, Zioni et al characterise the role of fatty bone marrow (FBM) in promoting the engraftment of DNMT3A mutated HSCs post transplant. The interaction of pre-leukemic clones with an inflamed bone marrow niche is currently an extremely relevant topic in the field, and the paper is valuable in that it adds to a growing body of literature underscoring the importance of these interactions. Also, the idea of using IL-6 blockade is original, and FBM is a relevant condition to study.

A general concern about this manuscript is that the mouse model seems quite artificial not only in the way that FBM was induced (see my specific point 1), but also in how clonal hematopoiesis (CH) was modelled – really, the authors look at the ability of DNMT3A mutated cells to engraft in FBM, which does not seem to have a lot to do with human CH, where multiple clones co-exist in steady state and then some expand, possibly as a consequence of inflammation. I am not sure if the conclusions on clonal hematopoiesis in the abstract and title are really appropriate.

The manuscript does not provide a lot of mechanistic insights. How do IL-6, TNF-alpha and/or IFN- γ signalling drive the clonal expansion of HSCs carrying the DNMT3A mutation? Is this also due to methylation of AP-1 transcription factor genes (FOS, JUN, etc), as described e.g. by ref 42 from their manuscript?

Besides this general comment, I also have several specific comments and suggestions.

1. The mouse model they use (sub-lethal irradiation) seems a rather artificial way to induce FBM, and this treatment is likely to also affect healthy hematopoietic cells. In the absence of genetic models that specifically cause FBM, castration could be a better alternative, but was only used for a few experiments in the paper. It would be desirable to repeat some of the key experiments from figure 3+4 with this model. Additionally, they should characterize hematopoietic defects arising from the treatments (castration, irradiation) in the recipient mice.
2. Related to this, are the pro-inflammatory cytokines (in particular, IL-6) secreted from adipocytes, other mesenchymal cells, or myeloid cells? i.e. is this pro-inflammatory signaling a direct effect stemming from the expansion of adipocytes, or a more general sign of perturbed bone marrow?
3. Figure 2gh: I believe a more appropriate control would be bone marrow from age-matched healthy donors with no CH, and not cord blood HSCs
4. Figure 3: The question they really ask is if an interaction of FBM and DNMT3Amut HSCs leads to increased engraftment, or if these two factors merely impact engraftment additively. Hence, in addition to performing statistical tests between pairs of treatment groups, it would be appropriate to statistically test if engraftment is better explained by a linear null model containing the terms bone marrow status + DNMT3A status, or an alternative model containing additionally an interaction term between BM status and DNMT3A status. Visually, it looks like this might only be the case in 1 year old donor mice, but not in panel a, b.
5. Figure 4: There are no biological replicates for the FBM_mut condition. The increased fraction of HSCs is the key statement from this figure and should be shown in more replicates. They could also use conventional FACS for this, instead of scRNA-seq.
6. Differential expression analysis, gene score analysis: A more appropriate way to perform these

analysis would first be to identify cell types (e.g. using their metacell analysis, or by reference mapping), and then identify changes caused by FBM, DNMT3A etc within each cell type specifically (e.g. specifically in HSCs, MPPs, etc). This way, they can make sure that the differential expression results are not to some part just a consequence of the different cell type proportions.

7. I additionally have several smaller comments regarding the scRNA-seq experiment

a. Can they explain a bit more why LSK cells were already collected 3 days after transplantation? What exact biological question should this experiment address?

b. The conditions shown in figure 3a are a bit hard to parse, it would be better to explain more clearly in the main text what was compared

c. In line 269-270, make clear that this is one gene set per pathway/factor and how these gene sets were identified.

8. I don't agree with the statement "previous studies focused on external insults, while our study correlates ageing and micro-environmental changes" from the discussion. To me irradiation and castration seem like external insults.

Minor points:

- Figure 2a-f: Do the engrafting clones also carry other mutations (beyond DNMT3a)? Especially in AML it is not uncommon that the "pre-leukemic" clone carries several driver mutations. If exome or panel seq data is available it should be included.

- Y axis label of figure S3f is unclear

- Methods, the sorting strategy for the single cell experiment is not described well. I assume CD45.2+ LSK cells were sorted?

Blue reviewer comment

Black authors response

Red changes in manuscript text

RESPONSE TO REVIEWER'S COMMENTS

Reviewer #1

1. Add stem cell factor (SCF) to the list of cytokines produced by the adipocytes (Zhou B et al. 2017, reference 7), and the role of adipocytes to support myeloid-erythroid differentiation (Boyd A et al. reference 14) that is mentioned, but not as evidence of adipocytes as positive regulators or hematopoiesis.

We thank the reviewer for his comment. We added in the introduction “Gene expression analysis of BM adipocytes suggested that they have distinct immune regulatory properties and high expression of pro-inflammatory cytokines (IL1A, IL1B, IL-6, IL8, IL15, IL18 and stem cell factor (SCF)¹.” We also added the following text regarding Boyd A et.al “Adipocytes might provide a protective niche for leukemia cells during chemotherapy by decreasing Bcl-2 and Pim-2 mediated apoptosis of leukemic cells². On the other hand, AML cells can reduce FBM, resulting in imbalanced regulation of HSCs and in myelo-erythroid maturation³.”

I consider that the authors focus on the role of adipocytes as promoters of inflammation (line 68), and do not stress the fact that they also play a key role on the support normal hematopoietic stem and progenitor cells (HSPC).

We thank the reviewer for this comment. After reviewing the literature once again, we believe that the role of adipocytes in normal haematopoiesis is still not fully resolved. Depending on the model, used different results are obtained. For example, Naveiras O. et.al (Nature 2009)⁴ clearly demonstrated that lower numbers of HSPCs were derived from FBM. In this study different mice models were used including after irradiation and the use of PPAR γ i. Similar results were reported by other⁵. However, human studies based on in vitro models provided contradicting results⁶. Studies on fatless mice demonstrated that lack of BM adipocytes impair HSC function, suggesting that BMF plays as a positive regulator of HSPCs. Altogether, we suspect that there is some confusion in the field. It seems that adipocytes are important for normal haematopoiesis, however their age-related accumulation might be harmful. We believe that these issues should be further resolved in future studies and different FBM models combined with modern single cell assays should be used. Ideally a genetic model for FBM should be developed. We have now added this part to the discussion. “However, we believe that for future studies more FBM models should be established. We propose that an optimal humanized NSG mice that will include human IL-6 and other humanized cytokines together with a conditional activation of PPAR γ in MSCs.”

It should also be discussed if there is a possibility that in the mouse the adipocytes are more reactive than in humans, as their number is higher and it is increased with physiological aging, in contrast to the mice.

The reviewer raises an important point. The accumulation of FBM with age in mice is not well characterized. Some evidence suggest that mice accumulate FBM in their vertebra but not in long bones⁷. These data suggest that the aetiology for FBM accumulation in mice and human are different. Clearly hormonal changes are less important in the aging mice as does other factors contributing to FBM accumulation. In the current study we aimed at adding irradiation in order to mimic the accumulation of FBM in long bones. Quantitatively, our irradiation model does mimic the accumulation of FBM in long bones, however we did not report any qualitative studies on the adipocytes. While we did the experiments and performed single nuclear sequencing on adipocytes from long bones after irradiation, we do not report these results yet as it was technically difficult to obtain high quality single nuclear sequencing data. We believe that speculating on this matter at this point is not supported by any data yet, and prefer to wait for more solid results. Ideally, we will need to compare adipocytes from vertebra (age related) to adipocytes after irradiation (long bones). Based on the reviewer comment we have added this important distinction between vertebra and long bones in mice to the introduction: **“As mice do not accumulate FBM in long bones with age as human do⁷, we used different external stresses to induce FBM accumulation in long bones of NOD-SCID-Gamma (NSG) mice so we could study the interaction of FBM with both human and mice preL-HSPCs.”**

The paper of Aguilar Navarro (89) does not have a reference number, and it mentions it was done in leukemic samples, but the paper is on human normal cells in aging and do not include AML samples.

We agree with the reviewer. We added the reference number and changed the text to: **“With age, BM adiposity correlates with increased density of mature myeloid cells and CD34+ HSPCs that contributing to age related risk of myeloid malignancies”**

.2 a) The assessment of adipocytes was done at one week of irradiation, however there is no data on the content of adipocytes at end point of the experiment (8 weeks)....

We thank the reviewer for this point. We added H&E staining (new Figure s1b) showing high levels of BM adipocytes 8 weeks following irradiation. We also added the following text to the manuscript **“High FBM was maintained even two months after irradiation (Figure s1b).”**

b) Figure 1e represents the amount of lipidTOX stain but is not fully described in methods, the information is not enough to repeat the experiment. I suggest adding a reference of this method if it was used in another publication. It is not clear about how LipidTOX was quantified...

We now added in the Materials and Methods a more detailed of *LipidTOX* protocol, we also added that bones were crushed **“Furthermore, the bones were crushed in order to obtain the adipocytes attached to the bones”**. We added a reference⁸. We added in supplementary new Figure S1a, a representative imagestream figures analysis.

I strongly suggest adding another adipocyte maker (for example perilipin) or quantitate sinusoids (VEGFR3) to rule out their participation. Figure 1f need to be more detailed in figure legend and specify what are each panel. I was not able to assess the bottom panels.

We thank the reviewer for these comments. We have added details to the Fig 1f legend “**Representative stacked whole-mount immunofluorescence staining of epiphyseal-metaphyseal BM femur derived from NBM, FBM and FBM & PPAR γ i treated NSG mice. Adipocytes are depicted by FABP4 expression and by distinctive unilocular morphology in the DIC (differential interference contrast) channel. DAPI in blue. Adipocytes are additionally marked by yellow dots. Scale Bar: 200 μ m; n=4 775 μ m x 775 μ m x 50 μ m stacked images from 2 mice and 2 bones (femur, tibia) each group**”.

We also added a new Figure s2 and in the material and Methods a detailed explanation of the staining “**Z-stacked confocal images are generated from whole-mounts of bisected mouse bones in which the structural and cellular integrity is highly preserved, and they show epiphyseal/metaphyseal BM regions (see Image S2a, zoom ins). Additional markers for adipocytes, vasculature and bone are not necessary^{9,10}. FABP4 is also expressed by endothelial cells, however adipocytes have higher expression. The DIC (differential interference contrast) channel was additionally used for the detection of the typical unilocular morphology of the adipocytes (see Image S2a) and other structures. Indeed, in BM sections big sinusoidal vessels are often collapsing, however here we used protective whole-mounts and the “empty” spaces here are trabecular bone structures (without surrounding FABP4 cells, see Image S2c) which are very frequently present in epiphyseal/metaphyseal BM. The sinusoidal vessels, with specific morphology, are clearly visible by lower FABP4 expression, surrounded by FABP4^{low} endothelial cells and in the DIC channel bone structures and sinusoidal vessels are reflected differently (see Image S2b and S2c).**”

3. I suggest revising the panel a on figure 4. Label the track on top of the heat map as hematopoietic subpopulations, and add a reference of the heat map, what those colors represent? The supplemental figure on the single cell data is hard to read (Figure 5S). I advice adding a stain for MPO in situ by IHC to complement Figure 4.

We appreciate the reviewer's input. We added labels to the heat map, as well as a color gradient. We also increased the size of the figures and separated Figure S5 into figures S6-S9. We believe that the single cell RNAseq analysis is more comprehensive than a single marker. If the reviewer implies that a detailed 2D map of adipocyte and HSPCs interaction will be of interest he is correct, however the technology (10X Visium Spatial Gene Expression) and number of experiments for such studies are out of the scope of the current study. We believe that staining just for MPO will not be informative enough and is not justified in the era of spatial genomics.

4. The evidence of IL-6 is not direct, as it is a cytokine produced by many cell types. IL-6 is a key cytokine that supports the growth of progenitor cells and is included in methocult assays to grow human cells and is an essential cytokine to expand HSC, so it can not be consider to have a more harmful role (line 426). To design a specific experiment to rule out their specific participation, it will need an approach as the one taken by Zhou, which conditionally deleted SCF from adipocytes, or co-cultured DNMT3A mut and wt

cells with adipocytes or perform an ISH for IL-6 mRNA. I suggest avoiding any strong conclusions on the role of IL-6 secreted by adipocytes only.....

We thank the reviewer for this comment. The reviewer is correct that we cannot exclude that other cells contribute to the higher levels of IL-6 we observed in the FBM versus NBM. To answer this question, we have added now the following new data. To confirm that the cytokine production is limited to the FBM, we examined cytokine levels in the mouse serum in parallel. In the serum, we could not observe increase in IL-6 after irradiation (new Figure 4e). We believe that this new data supports our claim that increased IL-6 in the BM originates from cells in the BM and not from systemic effects of irradiation. With regard to the secretion of IL-6 from FBM we provide evidence that IL-6 levels are decreased after the administration of PPAR γ i (new Figure 4d).

The reviewer also comments on the claim that IL-6 is an essential cytokine. However, this is partially correct. Previous studies on the role of IL-6 in HSC biology suggested that low IL-6 levels cause a reduction in HSCs and change differentiation trajectories¹¹. Human xenograft experiments suggests that IL-6 is not required for human HSC self-renewal and differentiation as human HSCs do not cross-react with the mouse IL-6¹². Altogether while IL-6 clearly modifies HSC biology it's not clear whether it is essential for haematopoiesis. In our *in vitro* studies, we clearly demonstrate similar results (Figure 5b). IL-6 addition to methocult did not change significantly the colony forming capacity of WT cells however our most important result in this regard is that IL-6 gives a selective advantage to DNMT3A Mut, as they had improved colony-forming capacity with IL-6 (Figure 5b). In the current study we do not claim that IL-6 cannot modify WT HSPC biology but clearly demonstrate that DNMT3A Mut are more sensitive to its addition (Figure 5b).

As the reviewer can notice, most of our studies were performed *in vivo* as it is highly recommended to assess stem cell function *in vivo*. In this regard, our *in vivo* IL-6 neutralizing Ab is very similar to the *in vitro* experiment proposed by the reviewer. In Figure 5c,d we provide evidence that although mice were exposed to irradiation (which induces FBM IL-6) the administration of IL-6 neutralizing Ab reduced DNMT3A Mut advantage both in primary and secondary engraftment. We did not include a WT group here as can be observed in Figure 3e DNMT3A WT cells exposed to FBM could not create secondary engraftment. This fact prevented from us to demonstrate any effect of IL-6 blocking on self-renewal in WT cells.

Altogether, our claims are careful throughout. However, we to the advice of the reviewer and toned down our claims regarding IL-6 and made it clear that DNMT3A Mut cells have a better selective advantage under high IL-6 levels.

The authors may benefit to include reference of increased levels of TNF alpha and IL-6 on myelodysplastic syndromes, specially by stromal cells, as MDS cells also harbored similar pre-leukemic mutations.

We added this sentence in the text “Studies have shown that BM stromal cells are determinants of the fate of hematopoietic progenitors and have an important role in the pathogenesis of MDS in which TNF- α induces significant changes in gene expression, particularly in apoptosis-related genes and cytokines/chemokines such as IL-6 and IL-8”

5.- I did not understand the context for the term durability (328).

We changed the term durability to “self-renewal”

Reviewer #2

1. Although the authors dedicate the majority of their Introduction to BM adipocyte biology and develop four different models of “fatty BM”, there is no analysis of BM adipocytes shown beyond Figure 1. In fact, two of the “fatty BM” models are never used following their initial description, and the castration model was abandoned after Figure 2f, where a comparison to healthy DNMT3A wildtype transplantation was never shown

The reviewer correctly describes our work. However, it is not clear what is suggested. Should we repeat all of our experiments under the 4 different models we established? Is it suggested that we should find samples from healthy individuals with *DNMT3A* WT? We believe the choices we took in the current study while might not be perfect are supporting the claims of the study. Clearly, what we propose here is novel and future studies can validate it under other different models. As can be seen in the comments from reviewer 1 the description of the 4 different models is important to the field as this is the first report of castration as an inducer of FBM, and the same is true for old SGM3-mice. With regard to aged matched healthy *DNMT3A* WT. We not understand the comment as we used 2 such samples. The first from cord blood which we sequenced and proved to be *DNMT3A* WT. We understand that cord blood was not an aged matched control so we used another sample from an age matched *DNMT3A* WT samples.

Throughout the manuscript, the authors rely almost exclusively on their “NBM” vs. “FBM” model, which represents non-irradiated mice vs. mice irradiated with 225 Rad one week prior. It is not possible to claim that any differences between these two models are driven by adipocytes.

We thank the reviewer for this comment. The reviewer is partially correct. While we used the NBM as one of our controls our most important control was a group of mice which were also irradiated (like the FBM) but were also treated with a PPAR γ inhibitor (PPAR γ i). We would like to refer the reviewer to figure 1, figure 2b, figure 3, figure 4d-g, S1a, S2c and S3. In all these figures we used a crucial control, namely, a PPAR γ inhibitor (PPAR γ i) which selectively inhibit the most important transcription factor in adipogenesis and reduces FBM formation after irradiation. As was noted by reviewer 1 this control was crucial to make our claims. From the comments from reviewer #2 it seems that they do not fully appreciate the importance of having 2 types of controls for the experiments we performed. The fact PPAR γ i provides selective advantage to WT cells (as was demonstrated in the past, and in our studies) stress even more the significant differences we observed while comparing *DNMT3A* MUT cells injected to FBM versus PPAR γ i control. The fact that PPAR γ i might have some effects on WT HSPC encouraged us to include the non-irradiated mice control group (NBM) also.

Total body irradiation causes cytokine storm and dramatic remodeling of all components of the BM microenvironment including osteoblasts, megakaryocytes, and vasculature (Dominic et al., Blood 2009;

Zhao et al., Nature Medicine 2014; Kenswil et al., Cell Reports 2018; Hooper et al., Cell Stem Cell 2009; Winkler et al., Nature Medicine 2012).

We thank the reviewer for this comment. However, it is important to stress that we used sub-lethal irradiation, while most of the references the reviewer suggest discuss the side effects of lethal dose irradiation. However, based on the reviewer suggestion we added in the text “**Nevertheless, total body irradiation causes cytokine storm and dramatic remodelling of all components of the BM microenvironment including osteoblasts, megakaryocytes, and vasculature^{13–15}. To control all these off targets effects of irradiation and other external stresses we have used a control group of mice that were irradiated and treated with a PPAR γ , bisphenol ADiGlycidyl Ether (BADGE). Previous studies have shown that PPAR γ treatment inhibits adipogenic differentiation *in vitro*¹⁶”**

Castration also causes systemic hormonal and physiological changes that are not limited to BM adipocytes. There are much more suitable models that could have been used to explore the independent role of BM adipocytes; e.g., rosiglitazone treatment (which the authors optimized), high fat diet, A-ZIP/F-1 “fatless” mice, or conditional adipocyte knockout models (“FAT-ATTAC” mice)

We thank the reviewer for this comment. We have spent years of choosing the best model that will allow us both human and mice studies. Clearly, our model is not perfect but neither of the models suggested by the reviewer are. The rosiglitazone will have other systemic effects too, and the same will be true with high fat diet – which was very inconsistent in our hands data not shown. The genetic models suggested by the reviewer are not suitable for human studies neither they model FBM as they model the opposite. We do agree that better models can be created in the future as we wrote now in the revised manuscript. “**However, we believe that for future studies more FBM models should established. We propose that an optimal humanized NSG mice that will include human IL-6 and other humanized cytokines together with a conditional activation of PPAR γ in MSCs.**”

It is unclear whether the BM even remains in a fatty state 9 weeks post-radiation at the time DNMT3A^{mut} vs DNMT3A^{WT} engraftment was evaluated.

We thank the reviewer for this point. We added H&E staining (new Figure S1a) showing high levels of BM adipocytes 8 weeks following irradiation. On top of that the reason we choose to focus our single cell RNA-seq three days after injection is the fact, we estimate that the major influence of FBM on stem cells is at the time near engraftment. We are not sure if FBM accumulation is reversible, as suggested by the reviewer. To answer such question, we will need to develop a reversible FBM model and see what happens to preL-HSPCs after reversal of FBM to NBM. It remains unclear whether FBM has a continuous interaction with preL-HSPCs or is it mainly during the engraftment which is a stress by itself. Future studies will be needed to resolve these interesting questions.

2. In addition to requiring more precise in vivo models, the most direct and controlled way to evaluate the effects of BM adipocytes would be to generate in vitro co-culture systems. It is not clear why the authors have not attempted these important experiments.

We thank the reviewer for this comment. We explained in the text “The decision to use an *in vivo* model rather than an *in vitro* model stemmed from the fact that *in vivo* model allow both human and mice

stem cell self-renewal assays. The influence of the microenvironment is long-term, making *in vitro* experiments difficult to monitor". In general, as we answered reviewer #1 contradicting results arose when investigators compared the effect of FBM on normal HSCs *in vivo* versus *in vitro* (see response to comment #1 reviewer #1). As the reviewer can appreciate, we turned into *in vitro* studies when we had more specific questions.

3. The authors themselves seem to lack confidence that their IL-6 findings have relevance to human samples. In their Discussion section, they explain that murine IL-6 does not cross-react with the human IL-6 receptor, which makes it difficult to interpret Figure 2. They suggest that TNF α levels may instead explain the interactions seen with human PreL-HSPCs but provide no data to support this suggestion. The central question of this paper revolves around the natural progression of clonal hematopoiesis to acute leukemia in human patients; however, the paper concludes with little clarity around the mechanism behind this process.

We thank the reviewer for this comment. The reviewer is fully correct, we cannot conclude that IL-6 modulates human *DNMT3A* Mut cells. However, we did provide evidence that IL-6 is secreted by mice bone marrow (Figure 4). IL-6 enhances the colony forming capacity of mice *DNMT3A* Mut cells more than it effects WT cells (Figure 5b). Blocking IL-6 reduced the self-renewal of *DNMT3A* Mut. Such mechanistic studies are more complicated with human samples. We are not claiming that such studies should not be conducted; however, we propose that they should be validated *in vivo* and such model is not available to us. We believe that we can claim that FBM provides selective advantage to human *DNMT3A* and the mechanisms in humans remains to be resolved. The mice study suggest that the mechanism might modulated partially by IL-6. We did not claim more than that. Such extrapolation has been done in many other studies in the past. In fact, many of the studies on the interaction between CH and the environment were performed in mice models^{17,18,19}.

4. The authors should provide evidence that DMNT3A mutant models truly represent a pre-leukemic state. In their AML patient-derived xenograft model, the authors should show more comprehensive cell surface phenotyping to clarify whether the grafts have multi-lineage differentiation capacity vs. leukemic engraftment that would be exclusively myeloid.

We thank the reviewer and added new data in figure S2b in which we demonstrate that the graft has a multi-lineage differentiation marker, suggesting a pre leukemic state.

Furthermore, it is unclear whether the R882H Dnmt3a knock-in model has been functionally validated to accelerate disease progression upon the introduction of additional mutations, e.g., as has been shown with the R878H Dnmt3a mouse model (Loberg et al., Leukemia 2019).

Loberg et al, developed model in which both *DNMT3A* and *NPM1* were mutated. Indeed, MPD was developed in *DNMT3A*-mutant CH mice after induction of *NPM1* mutation. In our case, we were interested in *DNMT3A* pre-leukemic mutation and therefor used the R882H Knock in model. We did not validate the acceleration of disease progression. As we refer to the manuscript with more data on the mice, we have used, we did not include all the information from that study. However, its should be made clear that our mice develop all the phenotypes of a *DNMT3A* mutated mice including HSPC expansion and specific pattern of hypomethylation¹⁸.

Major experimental issues:

1. Given that FBM and NBM groups actually represent “irradiation” vs “no irradiation” groups, it is quite surprising that there is no difference in engraftment levels when healthy human CD34+ cells or DNMT3A^{WT} murine cells are transplanted, even considering that transplantation was delayed by one week post-radiation. Peter Quesenberry’s group has shown that low dose radiation increases the engraftment of healthy donor cells even when transplantation is delayed to 8 weeks post-radiation (Stewart et al., Blood, 2001). How do the authors explain this discrepancy? This is important to address, as the lack of increased DNMT3A^{WT} cell engraftment in FBM mice forms the basis of the unique observations seen when DNMT3A^{mut} cells are transplanted.

We thank the reviewer for this comment. We carefully looked at the reference provided by the reviewer (<https://doi.org/10.1182/blood.V97.2.557>). We do not think one can take the results obtained from Balb/c mice and extrapolate them into NSG mice. Furthermore, as we discussed before, normal human engraftment to NSG mice is done after 24-48 hours and to our knowledge there is no data on engraftment after 1 week. As we have discussed in the response to comment #1 to reviewer #1, FBM might have harmful effects on normal HSCs and thus some of the consequences of irradiation might be mitigated by the negative consequences of FBM which can explain the similar engraftment we observed. In a similar way in mice, the only time we observed increased engraftment a week after irradiation is when we used the irradiation in combination with a PPAR γ i which mitigates the accumulation of FBM.

2. The authors argue that technical barriers precluded their ability to test the functional relevance of IL-6 exposure in the context of human preL-HSPCs. They argue that secondary transplantation assays would not be feasible with their human PreL-HSPC samples, however they could still perform in vitro experiments and primary transplantation assays at a minimum.

It is unclear why they could not treat with IL-6 and IL-6 neutralizing antibodies in vitro, followed by %DNMT3A VAF analysis and CFU assays. Similarly, intraperitoneal injections of human IL-6 could have been delivered in xenograft models

We thank the reviewer for these comments. These are all good suggestions but they are all technically challenging. The *in vivo* experiment is not clearly feasible – how long should IL-6 be administered to mice? at what dose? Does IL-6 even reach the BM? this is all unknown and as far as we know was not studied in the context of human sample xenografts. We agree that such experiments are interesting but as their feasibility is questionable and we do not claim about the role of IL-6 in humans, we are currently inclined from doing them (unless the reviewer can provide references where human samples were injected with IL-6, which can guide us). With regard to the human *in vitro* studies, again a set of experiments with many complications and no guarantee for clear results. Our results from figure 4 support the role of HSCs in the response to IL-6. Obtaining enough HSCs and not progenitors from samples with DNMT3A mutations is of great difficulty. We do not know how long HSCs need to be exposed to IL-6 and as they differentiate rapidly *in vitro*, it remains unclear if the experiment is feasible. On top of that, our main phenotype is an increase in self-renewal, which cannot be assessed in human samples *in vitro*. We will appreciate if the reviewer can refer us to similar studies from primary human preleukemic samples with *in vitro* work as suggested. Most mechanistic work like this is generally done

in mice. We do agree with the reviewer that work like this is more feasible in leukemic samples. Altogether, these are great ideas that will need a lot of new technology and assays that would be our next set of experiments in the next few years.

3. In Figure 2a and 2d, why haven't the authors shown %DNMT3A VAF for both NBM and FBM conditions?

We thank the reviewer for noticing this missing data. The results of %DNMT3A VAF following engraftment in FBM is shown in Figure 2a and 2d. We also added in the text this explanation: **“Furthermore, we sequenced NBM samples. However, due to the limited engraftment, we were unable to obtain any human cells after sorting and no human reads were available after sequencing.”**

4. In Figure 3, the authors place a lot of emphasis on the age of the donor hematopoietic cells and rationalize that older PreL-HSPCs will have more hypomethylation. The Acknowledgments section indicates that methylation assays were performed, however this was never shown. Why wouldn't the authors include these data to contextualize their findings in Figure 3?

We thank the reviewer for this comment. All the methylation data is published and available in the Scheller et.al. manuscript¹⁸. We reanalyzed that data and did not find any informative information. With regard to the older mice. We clearly can observe they have increased self-renewal (Figure 3) as we did not perform the methylation analysis on older mice, we do not make any claims about it. We agree it is an important experiment to understand the evolution of methylation in the mice, however it was already done by others as we cite in the manuscript.

5. In Figure 3, the authors compared 1 year old preL-HPSCs to control conditions, but did not perform statistical analysis directly comparing 1 year old preL-HPSCs to 2 month old preL-HSCs. In order for the older preL-HPSCs to be considered more affected by FBM as hypothesized, a combined analysis needs to be performed.

We thank the reviewer for noticing this and we now added this analysis (new Figure S2c). In the text we added the following sentence: **“Indeed, significant increase in engraftment was detected following transplantation of old DNMT3A^{Mut} cells derived from one-year-old mice injected into NBM, FBM and PPAR γ i controls compare to two months DNMT3A^{Mut} derived BM (new Figure 2Sc)”**

5. In Figure 3, how do the authors explain the observation that wildtype cells have increased engraftment after BADGE treatment? Adipocyte levels are intermediate in BADGE condition compared to other experimental groups. This suggests effects of BADGE on hematopoietic cells could be independent from adipocyte content.

We thank the reviewer for this comment. We addressed this issue in the text **“Interestingly, the administration of PPAR γ i to FBM mice transplanted with one-year-old DNMT3A^{WT} cells resulted in a significant increase of engraftment (Figure. 3c, Figure S2c). Similar results on the effect of PPAR γ inhibition on HSCs have been reported in the past²⁰”**.

More importantly, the fact PPAR γ i provides selective advantage to WT cells (as was demonstrated in the past), stress even more the differences we observed while comparing DNMT3A MUT cells injected to

FBM versus PPAR γ i control. The fact that PPAR γ i might have some effects on WT HSPC encouraged us to include the non-irradiated mice control group also.

6. The connection between IL-6 secretions and BM adipocytes is lacking. Figure 4d-g is really an analysis of cytokine secretion in the presence vs. absence of total body irradiation. The analysis of the FBM + BADGE condition adds some value (Fig S8c,d), but this should have been evaluated in other fatty BM models that are independent of total body irradiation. It is not surprising that there would be an increase in inflammatory cytokines after acute injury to the BM, and this may have nothing to do with adipocytes.

We agree with the reviewer that total body irradiation can influence other cell types and other tissues. To control all these off targets effects of irradiation and other external stresses we have used a control group of mice that were irradiated and treated with a PPAR γ i, bisphenol ADiGlycidyl Ether (BADGE). More importantly we have now added new data to support our claims. First, we added cytokine levels in the serum (New figure 4e). In the serum IL-6 levels do not change suggesting the IL-6 is local to the BM. We also added new figure 4d. In this figure we took the advice of the reviewer and analyzed cytokines 2 month after irradiation. At this stage no acute irradiation side effects should be noticed. Most importantly BADGE which was administrated with the irradiation significantly reduced IL-6. All of this provide evidence that it is the FBM contributing to the high IL-6. On top of that, we are not the first to propose this (IL-6 was increased in both human and mice FBM) as we extensively discuss in the older version of the manuscript.

Minor issues:

1. The presentation of data related to the SRSF2 P95H model should be in the main figures given the amount of emphasis in the text.

We prefer to include these results in the supplementary data as they are negative results.

2. The authors should explain their rationale for performing scRNAseq experiments at day 3 post-transplant, as this was not explained in the text.

We thank the reviewer for this comment. In the revised manuscript we now explain this in material and Methods: **“We calibrated the quantity of LSK at three time points. a day, three and five days post injection. Our results demonstrated that following three days of injection, the optimal number of LSK for scRNA seq (data not shown).”**

2. The Discussion section is unnecessarily long, and some points are only tangentially related to the study (e.g., discussion of cardiovascular disease and heart failure, etc).

We edit the discussion section and shortened it from 3.5 to 2 pages.

Reviewer #3

1- The abstract mentions a 20-50 fold increase in DNMT3A^{mut}-preL-HSCs, and on page 8 they state “A 50 fold increase in HSCs was noted when comparing DNMT3A^{mut} cells exposed to FBM and DNMT3A^{WT} injected to NBM (Figure 4c, Table S4).” Figure 4c shows something unrelated. In Table S4, the values for

DNMT3A^{mut} FBM HSCs and DNMT3A^{WT} NBM HSCs are 51/260=20% and 3/164=1.8%, respectively. This is an 11-fold difference, not 50

We would like to thank the reviewer for this comment. We agree with this and changed it in the abstract. **“A ten fold increase in DNMT3A^{Mut}-preL-HSCs was observed under FBM conditions in comparison to other conditions in which myeloid differentiation occurred”**

2. Figure 4A is a white/red heatmap that is uninterpretable due to the absence of a legend for the color gradient. Furthermore, if the rows are clustered, a dendrogram should be shown. It may be more clear if the rows were not clustered but rather shown in a logical order.

We appreciate the reviewer's input. We added labels to the heat map, as well as a colour gradient.

3- “The maintenance of HSCs among DNMT3A^{Mut} cells exposed to FBM was followed by expansion of myeloid progenitors as opposed to enrichment of lymphoid progenitors in the naïve LSK cells (Figure 4a,b).” – this is not clear from Figure 4a due to the previous point and not clear from Figure 4b which does not show lymphoid progenitors.

We added labels to the heat map, as well as a colour gradient in new figure 4.

4- Figure legend referrals are erroneous, for example, Figure 3e is not cited in the text, and Figure 4e-h on line 308 should presumably be Figure 4d-g.

We performed all the suggested comments.

5- The authors consistently refer to INF α and INF γ but likely mean IFN α and IFN γ .

We corrected this.

From the human experiments, the authors suggest “a role for an adipocyte-rich environment in enhancing engraftment of human preL-HSPCs, but not for normal HSPCs”. However, they appear to use mononuclear cells for the AML/lymphoma samples and CD34+ cells for the normal cord blood. is may not be a fair comparison.

We thank the reviewer for this comment. In order to be able to study preleukemic HSPCs one of the options is to engraft AML samples and identify samples with multi lineage engraftment as we demonstrated in the past²¹. Actually, this is one of the most useful resources to isolate human naïve preL-HSPCs which were not exposed to chemotherapy as we collect the samples from AML diagnosis. We also studied the auto-BMT CD34 cells as another source for human preL-HSPCs. Cord blood are considered as the best source for human healthy HSCs as they have the best engraftment capacity. We believe we chose the right samples.

More importantly, the VAF of DNMT3A mutations does not increase in the human grafts (Figure 2a, d), which may be interpreted as an internal control arguing against a specific advantage for DNMT3A-mutated human cells.

We thank the reviewer for this point. In order to fully understand this, one should separate the VAF in figure 2a and 2d. In Figure 2a the original sample is fully clonal for both DNMT3A and NPM1 as we

sequenced the leukemic cells. The preleukemic cells are extremely rare in the diagnosis sample. For reasons we do not fully understand the leukemic cells do not engraft (as *NPM1* is absent in the mice). To fully appreciate whether the VAF in the mice is higher than the VAF in preL-HSPCs, one would need to isolate preL-HSPCs and sequence them. In our manuscript²¹, we demonstrated that after injecting AML cells to NSG mice, VAF is increasing significantly after sixteen weeks following transplantation. However, in figure 2a one can appreciate the high VAF of the preL-HSPCs as compared to an average VAF of preL-HSPCs in our Nature manuscript of 25.6%. The reviewer should be aware that any VAF value more than 50% may indicate a sequencing mistake or loss of heterozygosity in some of the clones. With regard to figure 2d, this is a different case as the cells injected and analyzed are a combination of preL-HSPCs and mature cells. Again, the same concepts apply. However, the most important comparison would have been the VAF on NBM, however as almost no engraftment was present, we could not assess it. See our response to reviewer #2 comment #3.

The visualizations and data to support cell type classifications are uninformative (Figure S5 and Table S1).

We now increased the figures so that the axis is visible in all these figures

The authors should generate gene signatures for each cell type, include these signatures as a supplementary table, and include dimensionality reduction (UMAP or tSNE) plots colored by the signature score.

We thank the reviewer for this comment. We have used a different methodology to cluster cell types. This approach was developed at the Weizmann institute of science and implemented in our lab²². This approach provides similar outputs to what the reviewer suggest, as can be seen in figure S9b and in Supplementary table 1. We highly encourage the reviewer to actively interrogate our data as we made special effort to make it accessible to all in a unique way. It is important to realize that each dot in our 2-dimensional projection is not a single cell but rather a meta-cell which includes cells with similar transcriptional programs. Supplementary table 1 give a direct distribution of the number of single cells in each cell type and the meta-cell numbers in the table correspond with the number in the Shiny application we made available. All this analysis was done with the guidance and help of a student from Amos Tanay group (Bercovich A) who wrote these algorithms and a bioinformatician from our lab (Chapal-Ilani N).

Minor comments

Flow cytometry is limited to a few markers. Example gating schemes should be included in the supplement. It would have been nice if the authors had included more markers beyond human CD45 and mouse CD45.1/CD45.2. Do the mice develop lympho-myeloid grafts? Is the stem cell compartment proportionally expanded in the FBM / DNMT3A-mutated mice?

We thank the reviewer for this comment. With regard to the human FACS we have now added new supplementary figure S3b with FACS examples. With regard to the mice, we did not perform detailed FACS analysis of the grafts but rather performed scRNA which provide more detailed information.

To profile cytokine secretion, the authors appear to have used the irradiation model. Can similar results be obtained using the castration model?

We thank the reviewer for this comment and added new data in figure S11b, cytokines secretion in NBM without CAS and FBM following castration.

Figure S1b should be quantified similarly to Figure 1e.

We agree with the reviewer this will improve the current publication but will take more time as we need to wait at least a month after castration. We now appreciate that we might need to wait even longer and we might get more FBM. If the reviewer insists on this, we can do it for the next revision.

Please clarify if the GSEA analyses in the section on page 9 were performed on all cells or specifically on HSC clusters. If the analyses were done on all cells, please also include the HSC-specific comparisons.

We thank the reviewer for this comment; we have now clarified it in the revised manuscript. The comparison was made on all cells. We could not do the HSC alone comparison as the reviewer can appreciate the number of HSCs in the NBM and control groups were very low.

What is “unselected” in Figure S6A?

The unselected population was not analysed.

Reviewer #4

A general concern about this manuscript is that the mouse model seems quite artificial not only in the way that FBM was induced (see my specific point 1), but also in how clonal hematopoiesis (CH) was modelled – really, the authors look at the ability of DNMT3A mutated cells to engraft in FBM, which does not seem to have a lot to do with human CH, where multiple clones co-exist in steady state and then some expand, possibly as a consequence of inflammation. I am not sure if the conclusions on clonal hematopoiesis in the abstract and title are really appropriate.

We thank the reviewer for his comment. With regard to our “artificial model” we refer the reviewer to our response to all the sub remarks made by reviewer #2 in comment #1. To conclude this, because this topic is new and there is no gold standard model as mentioned by reviewer #1, we choose to focus on two models. No model is perfect; for example, the aging of mice does not replicate the accumulation of fat in long bones (the major site of haematopoiesis and the place we make our interfemoral transplantations).

With regard to the CH model, we used 4 models: 3 of the models are mice (*DNMT3A* R882 *DNMT3A* haplo-insufficient and *SRSF2*) and several human samples. We focused on a single mutation in the first step as adding competition (which is clearly important as was recently published <https://doi.org/10.1038/s41586-021-04206-7>) will add to the complexity, of this already novel and complex interactions. We agree that our CH models are far from being perfect and discuss it in the discussion.

The manuscript does not provide a lot of mechanistic insights. How do IL-6, TNF-alpha and/or IFN-γ signalling drive the clonal expansion of HSCs carrying the DNMT3A mutation? Is this also due to methylation of AP-1 transcription factor genes (FOS, JUN, etc), as described e.g. by ref 42 from their manuscript?

We thank the reviewer for his comment. According to our scRNAseq data analysis, an upregulation of IL-6 pathway under FBM is shown following transplantation of *DNMT3A^{Mut}* cells. Our *in vitro* colony assays prove that high IL-6 levels secreted by FBM provide selective advantage to *DNMT3A^{Mut}*. We also show that IL-6 inhibition results in reduced engraftment of *DNMT3A^{Mut}* cells. The end product of these changes is increased self-renewal. We cannot shed more light on the regulation of self-renewal versus differentiation in the current project. More scRNA data preferentially on human samples will be needed to better study this as was suggested by reviewer #2, however as we explained in our response to reviewer #2 comment #3 such experiments are beyond the scope of the current study.

Besides this general comment, I also have several specific comments and suggestions.

1. The mouse model they use (sub-lethal irradiation) seems a rather artificial way to induce FBM, and this treatment is likely to also affect healthy hematopoietic cells. In the absence of genetic models that specifically cause FBM, castration could be a better alternative, but was only used for a few experiments in the paper. It would be desirable to repeat some of the key experiments from figure 3+4 with this model.

We thank the reviewer for this comment. The reviewer is partially correct. While we used the NBM as one of our controls our most important control was a group of mice which were also irradiated (like the FBM) but were also treated with a PPAR γ inhibitor (PPAR γ i). We would like to refer the reviewer to figure 1, figure 2b, figure 3, figure 4d-g, S1a, S2c and S3. In all these figures we used a crucial control, namely, a PPAR γ inhibitor (PPAR γ i) which selectively inhibit the most important transcription factor in adipogenesis and reduces FBM formation after irradiation. As was noted by reviewer #1 this control was crucial to make our claims. From the comments from reviewer #4 it seems that they do not fully appreciate the importance of having two types of controls for the experiments we performed. As the irradiation model is much simpler and does not involve an operation, and was more reproducible we chose to focus on it. We do provide some of the key experiments mainly in humans with another model but we disagree that all experiments should be repeated with the castration model, especially since it's clear that we use the PPAR γ i control. We hope that once the reviewer will realise the importance of the PPAR γ i control they will agree with our opinion and the opinion of other reviewers.

We took the advice of the reviewer and added new data in new figure S11b, dealing with cytokines levels in the BM two month after castration.

Additionally, they should characterize hematopoietic defects arising from the treatments (castration, irradiation) in the recipient mice.

We agree with the reviewer that total body irradiation can influence other cell types and other tissues. To control all these off targets effects of irradiation and other external stresses we have used a control

group of mice that were irradiated and treated with a PPAR γ i, bisphenol ADiGlycidyl Ether (BADGE). Previous studies have shown that PPAR γ i treatment inhibits adipogenic differentiation *in vitro*.

2. Related to this, are the pro-inflammatory cytokines (in particular, IL-6) secreted from adipocytes, other mesenchymal cells, or myeloid cells? i.e. is this pro-inflammatory signalling a direct effect stemming from the expansion of adipocytes, or a more general sign of perturbed bone marrow?

We would like to direct the reviewer to figure 4d, which depicts the level of IL-6 in NBM, FBM NSG mice, and after PPAR γ i treatment without cell transplantation. The level of IL-6 increases after irradiation but decreases dramatically after PPAR γ i treatment, indicating that IL-6 is produced by adipocytes. Furthermore, to confirm that the cytokine production is limited to the FBM, we examined cytokine levels in the mouse serum in parallel. Only MCP1 showed a significant difference between FBM and NBM (Figure 4e). We believe that these controls may convince reviewer questions regarding IL-6 secretion by adipocytes.

3. Figure 2gh: I believe a more appropriate control would be bone marrow from age-matched healthy donors with no CH, and not cord blood HSCs

We thank the reviewer for this comment. The sample we used was from an aged matched *DNMT3A* WT sample. The terminology of healthy is problematic (the patient was healthy at the time of BM collection). To control for that, we have used cord blood which is clearly healthy. We believe these two controls are adequate as was used in many other studies in the field. We added in the text **“To study the interaction between FBM and normal haematopoiesis we also transplanted wild type (WT) CD34+ cells from pooled cord blood samples and from aged matched healthy donor without clonal haematopoiesis”**

4. Figure 3: The question they really ask is if an interaction of FBM and DNMT3A mut HSCs leads to increased engraftment, or if these two factors merely impact engraftment additively. Hence, in addition to performing statistical tests between pairs of treatment groups, it would be appropriate to statistically test if engraftment is better explained by a linear null model containing the terms bone marrow status + DNMT3A status, or an alternative model containing additionally an interaction term between BM status and DNMT3A status. Visually, it looks like this might only be the case in 1 year old donor mice, but not in panel a, b.

We thank the reviewer for this comment. We performed a 3-way ANOVA with interaction using the following variables: BM (FBM/NBM), *DNMT3A* (Mut/WT), and Age (2/12 month-old). We found a statistically-significant difference in the engraftment levels yield by all three variables independently. Moreover, the interaction between these terms was also significant including all combinations as shown in the table 1 below.

Table 1

Analysis of Variance					
Source	Sum Sq.	d.f.	Mean Sq.	F	Prob>F
BM	473.17	1	473.167	13.98	0.0003
DNMT3A	708.38	1	708.38	20.92	0
AGE	736.38	1	736.384	21.75	0
BM*DNMT3A	307.52	1	307.516	9.08	0.0031
BM*AGE	258.95	1	258.946	7.65	0.0064
DNMT3A*AGE	682.36	1	682.362	20.16	0
BM*DNMT3A*AGE	301.97	1	301.969	8.92	0.0033
Error	4739.7	140	33.855		
Total	8032.91	147			

5. Figure 4: There are no biological replicates for the FBM_mut condition. The increased fraction of HSCs is the key statement from this figure and should be shown in more replicates. They could also use conventional FACS for this, instead of scRNA-seq.

We thank the reviewer for this comment. We actually repeated the experiment in Figure 4 several times as part of our calibration experiments for the scRNA-seq. So, we already have the data analysed by FACS for two more biological replicates. We have now added this new data as new figure S13b. While we agree with the reviewer that this is of importance, the reviewer should be aware that phenotypic stem cells are not the gold standard in the field. By demonstrating higher number of LSK it does not mean they are more functional as was shown by the group of Goodll Plos Gene 10.1371/journal.pbio.0050201. In this regard, the gold standard stem cell assays are engraftment and secondary engraftment. For these experiments, we performed several biological replicates. Figure 4's aim is mechanistic rather than to demonstrate what we have proven in figure 3, though it clearly supports it.

6. Differential expression analysis, gene score analysis: A more appropriate way to perform these analysis would first be to identify cell types (e.g. using their metacell analysis, or by reference mapping), and then identify changes caused by FBM, DNMT3A etc within each cell type specifically (e.g. specifically in HSCs, MPPs, etc). This way, they can make sure that the differential expression results are not to some part just a consequence of the different cell type proportions.

We thank the reviewer for this comment, and fully agree with it and indeed tried it. However, due to low cell numbers and multiple hypothesis comparisons we were not powered to identify significant changes. Accordingly, we used all cells for the differential expression. To make this point clear we have now added the following text: "To better characterize the DNMT3AMut FBM cluster, we calculated differential gene expression between the different clusters in the UMAP (Table S2) and used all cells in each cluster."

7. I additionally have several smaller comments regarding the scRNA-seq experiment

a. Can they explain a bit more why LSK cells were already collected 3 days after transplantation? What exact biological question should this experiment address?

We added this information in the new Materials and Methods section. "We calibrated the quantity of LSK at three time points: a day, three and five days post injection. Our results demonstrated that

following three days of injection, the highest number of LSK cells from all conditions (data not shown).”

b. The conditions shown in figure 3a are a bit hard to parse, it would be better to explain more clearly in the main text what was compared to c

We changed the text according to the reviewer comment: “In contrast to control mice with NBM or mice irradiated and treated with the PPAR γ inhibitor (PPAR γ i), the injection of 2 months BM derived *DNMT3A*^{Mut} cells (CD45.2) intra femorally (IF) to NSG (CD45.1) mice with FBM resulted in considerably higher engraftment of *DNMT3A*^{Mut} cells (Figure. 3a). When *DNMT3A*^{WT} cells were transplanted, this enhanced engraftment was not detected under FBM conditions (Figure. 3a)”. We also added Figure S2c in which we compare *DNMT3A*^{Mut} two months and one-year transplanted mice. We detailed it in the text: “Indeed, significant increase in engraftment was detected following transplantation of old *DNMT3A*^{Mut} cells derived from one-year-old mice injected into NBM, FBM and PPAR γ i controls compare to two months *DNMT3A*^{Mut} derived BM (new Figure S2c). *DNMT3A*^{Mut} cells derived from one-year-old mice injected into FBM had the most significant growth advantage in comparison to NBM and PPAR γ i controls. When *DNMT3A*^{WT} cells were injected, this effect of FBM could not be observed (new Figure.3c, Figure S2c). Interestingly, the administration of PPAR γ i to FBM mice transplanted with one-year-old *DNMT3A*^{WT} cells resulted in a significant increase of engraftment (new Figure. 3c, Figure S2c)”

c. In line 269-270, make clear that this is one gene set per pathway/factor and how these gene sets were identified.

We thank the reviewer for this comment the methods for the GSEA analysis are described in the methods section and we have edited them to add more information. “The DE genes of cluster 1 (Table S2) were ranked based on fold change and analyzed using the GSEA software version 4.1.06868686868. Significant genes set had FDR q-val<0.2. The Hallmark genes sets were used for the analysis.”

8. I don't agree with the statement “previous studies focused on external insults, while our study correlates ageing and micro-environmental changes” from the discussion. To me irradiation and castration seem like external insults.

We agree with this reviewer comment. We removed this sentence from the discussion.

Minor points:

- Figure 2a-f: Do the engrafting clones also carry other mutations (beyond DNMT3a)? Especially in AML it is not uncommon that the “pre-leukemic” clone carries several driver mutations. If exome or panel seq data is available it should be included.

We thank the reviewer for this comment. Indeed, all human samples were sequenced with our in house clonal haematopoiesis panel. The AML sample #160005: had only these two mutations: *NPM1*+*DNMT3A* R882H and no other preleukemic mutations. Sample #141464 had only *DNMT3A* R882H. We have now added a sentence to the manuscript describing this: “All samples were sequenced with our in-house clonal haematopoiesis panel (Beizuner T et.al. in press).”

- Y axis label of figure S3f is unclear

We changed to engraftment differences

- *Methods, the sorting strategy for the single cell experiment is not described well. I assume CD45.2+ LSK cells were sorted?*

We added in the text “Our results demonstrated that following three days of injection, the optimal number of LSK for scRNA seq (data not shown). Then, three days after injection LSK gated on CD45.2 population cells were isolated. We also isolated cells from the same donor mice before they were injected and termed them naïve cells”. We also added FACS data (new Figure 12S) illustrated how we sorted the cells

References

1. Zhou, B. O. *et al.* Bone marrow adipocytes promote the regeneration of stem cells and haematopoiesis by secreting SCF. *Nature Cell Biology* **19**, 891–903 (2017).
2. Behan, J. W. *et al.* Adipocytes impair leukemia treatment in mice. *Cancer Research* **69**, 7867–7874 (2009).
3. Boyd, A. L. *et al.* Acute myeloid leukaemia disrupts endogenous myelo-erythropoiesis by compromising the adipocyte bone marrow niche. *Nature Cell Biology* **19**, 1336–1347 (2017).
4. Naveiras, O. *et al.* Bone-marrow adipocytes as negative regulators of the haematopoietic microenvironment. *Nature* **460**, 259–263 (2009).
5. Ambrosi, T. H. *et al.* Adipocyte Accumulation in the Bone Marrow during Obesity and Aging Impairs Stem Cell-Based Hematopoietic and Bone Regeneration. *Cell Stem Cell* **20**, 771-784.e6 (2017).
6. Mattiucci, D. *et al.* Bone marrow adipocytes support hematopoietic stem cell survival. *Journal of Cellular Physiology* **233**, 1500–1511 (2018).
7. Naveiras, O. *et al.* Bone-marrow adipocytes as negative regulators of the haematopoietic microenvironment. *Nature* **460**, 259–263 (2009).
8. Majka, S. M. *et al.* Analysis and Isolation of Adipocytes by Flow Cytometry. (2014) doi:10.1016/B978-0-12-411619-1.00015-X.
9. Theresa Landspersky, Mehmet Saçma, Jennifer Rivière, Judith S. Hecker, Franziska Hettler, Erik Hameister, Katharina Brandstetter, Rouzanna Istvanffy, Sandra Romero Marquez, Romina Ludwig, Marilena Götz, Michèle Constanze Buck, Martin Wolf, Matthias Schiem, R. A. J. O. Autophagy In Mesenchymal Progenitors Protects Mice Against Bone Marrow Failure After Severe Intermittent Stress. *blood* (2021).
10. Saçma, M. *et al.* Haematopoietic stem cells in perisinusoidal niches are protected from ageing. *Nature Cell Biology* **21**, (2019).
11. Bernad, A. *et al.* Interleukin-6 is required in vivo for the regulation of stem cells and committed progenitors of the hematopoietic system. *Immunity* **1**, 725–731 (1994).
12. van Dam, M. *et al.* Structure-function analysis of interleukin-6 utilizing human/murine chimeric molecules. Involvement of two separate domains in receptor binding. *The Journal of biological chemistry* **268**, 15285–90 (1993).
13. Jared, K. *et al.* Characterization of Endothelial Cells Associated with Hematopoietic Niche Formation in Humans Identifies IL-33 As an Anabolic Factor Article Characterization of Endothelial Cells Associated with Hematopoietic Niche Formation in Humans Identifies IL-33 As. *Kenswil et al., Cell Reports 2018* 666–678 (2018) doi:10.1016/j.celrep.2017.12.070.
14. Hooper, A. T. *et al.* Article Engraftment and Reconstitution of Hematopoiesis Is Dependent on VEGFR2-Mediated Regeneration of Sinusoidal Endothelial Cells. *Stem Cell* **4**, 263–274 (2009).

15. Winkler, I. G. *et al.* Vascular niche E-selectin regulates hematopoietic stem cell dormancy, self renewal and chemoresistance. *Nature Medicine* (2012) doi:10.1038/nm.2969.
16. Wright, H. M. *et al.* A Synthetic Antagonist for the Peroxisome Proliferator-activated Receptor γ Inhibits Adipocyte Differentiation. *Journal of Biological Chemistry* **275**, 1873–1877 (2000).
17. Meisel, M. *et al.* Microbial signals drive pre-leukaemic myeloproliferation in a Tet2-deficient host. (2018) doi:10.1038/s41586-018-0125-z.
18. Scheller, M. *et al.* Hotspot DNMT3A mutations in clonal hematopoiesis and acute myeloid leukemia sensitize cells to azacytidine via viral mimicry response. *Nature Cancer* **2**, 527–544 (2021).
19. Hormaechea-Agulla, D. *et al.* Chronic infection drives Dnmt3a-loss-of-function clonal hematopoiesis via IFN γ signaling. *Cell Stem Cell* **28**, 1428-1442.e6 (2021).
20. Li, R. Z. M. W. Z. Hematopoietic recovery following chemotherapy is improved by BADGE-induced inhibition of adipogenesis. 58–72 (2013) doi:10.1007/s12185-012-1233-4.
21. Shlush, L. I. *et al.* Identification of pre-leukaemic haematopoietic stem cells in acute leukaemia. *Nature* **506**, 328–333 (2014).
22. Baran, Y. *et al.* MetaCell: analysis of single-cell RNA-seq data using K-nn graph partitions. doi:10.1186/s13059-019-1812-2.
23. Shlush, L. I. *et al.* Identification of pre-leukaemic haematopoietic stem cells in acute leukaemia. *Nature* (2014) doi:10.1038/nature13038.
24. Tuval, A. & Shlush, L. I. Evolutionary trajectory of leukemic clones and its clinical implications. *Haematologica* **104**, 872–880 (2019).

REVIEWERS' COMMENTS:

Reviewer #1 (Remarks to the Author):

According to the concerns raised on the previous version of the paper I consider that the authors addressed all my comments and resolved all of them, but one. Please find my comments to the addressed points below.

1.- Introduction:

Add stem cell factor (SCF) to the list of cytokines produced by the adipocytes (Zhou B et al. 2017, reference 7), and the role of adipocytes to support myeloid-erythroid differentiation (Boyd A et al. reference 14) that is mentioned, but not as evidence of adipocytes as positive regulators or hematopoiesis. I consider that the authors focus on the role of adipocytes as promoters of inflammation (line 68), and do not stress the fact that they also play a key role on the support normal hematopoietic stem and progenitor cells (HSPC). It should also be discussed if there is a possibility that in the mouse the adipocytes are more reactive than in humans, as their number is higher and it is increased with physiological aging, in contrast to the mice. The paper of Aguilar Navarro (89) does not have a reference number, and it mentions it was done in leukemic samples, but the paper is on human normal cells in aging and do not include AML samples.

- Resolved: the authors addressed all concerns and included the necessary information into the revised version.

2.- The assessment of adipocytes needs to be more detailed and include more information:

a) The assessment of adipocytes was done at one week of irradiation, however there is no data on the content of adipocytes at end point of the experiment (8 weeks). On my experience (data not published) the increased-on adipocytes and sinusoids is transitory, so I consider it will be very helpful to know the content at end point. If it is transitory, the model is still valid and important, but need to be taken into consideration.

- Resolved: the authors added an additional figure showing that in their hands, they found that the increased in adipocyte content can last up to 2 months after irradiation.

b) Figure 1e represents the amount of lipidTOX stain but is not fully described in methods, the information is not enough to repeat the experiment. I suggest adding a reference of this method if it was used in another publication. It is not clear about how LipidTOX was quantified. If it was in situ, they need to include representative images. They have included in methods, the Image Stream analysis to quantify adipocytes. The method quantifies adipocytes on flushed samples, but the adipocytes will unlikely be at this fraction. To obtained stromal cells, the bones need to be crushed. So, it is important that the authors show the method in detailed. This is relevant because the use of FABP4 as a marker for adipocytes is not specific. FABP4 also stains some sinusoids (which are evident in the right panel of non irradiated mice on figure 1f). It is very important to validate the model with at least two markers. Mouse sinusoids are larger and more open that their human counterpart, that usually look collapsed in bone marrow biopsies, so it is hard to assessed them by histology only in both species for different reasons. The use of LpidTOX is adequate, however it is not clear how it was used. I strongly suggest adding another adipocyte maker (for example perilipin) or quantitate sinusoids (VEGFR3) to rule out their participation. Figure 1f need to be more detailed in figure legend and specify what are each panel. I was not able to assess the bottom panels.

- Unresolved: The authors included the necessary information to understand the method they used to assess adipocytes by lipidTOX and they also added additional figures that explained the evaluation of the FABP4 marker. However, the authors did not acknowledge that FABP4 can also stained endothelial cells. Line 128 states: "These results were validated by staining irradiated BM with the adipocyte marker fatty acid binding protein 4 (FABP4) (Figure 1f)". The authors relied on a differential expression of the intensity of FABP4 and the segmentation by image analysis. I strongly believe that is necessary to acknowledge that FABP4 is not a specific marker (that it also stained endothelial cells), but that the use of image analysis restricted the identification to adipocytes according to the intensity of the marker, as discussed in the letter to the reviewers. The

authors did not take into consideration the advice to use a specific marker for adipocytes. Due to the relevance of this point to the paper, I do believe that the authors need to address this point.

3.- I suggest revising the panel a on figure 4. Label the track on top of the heat map as hematopoietic subpopulations, and add a reference of the heat map, what those colors represent? The supplemental figure on the single cell data is hard to read (Figure 5S). I advice adding a stain for MPO in situ by IHC to complement Figure 4.

- Resolved: The authors labeled the figures and provided a logic answer to the MPO point. However, it is important to consider that the number of cells analyzed by scRNAseq is very low. As an additional suggestion, the authors may add the confidence on the call of hematopoietic hierarchy.

4.- The evidence of IL-6 is not direct, as it is a cytokine produced by many cell types. IL-6 is a key cytokine that supports the growth of progenitor cells and is included in methocult assays to grow human cells and is an essential cytokine to expand HSC, so it can not be consider to have a more harmful role (line 426). To design a specific experiment to rule out their specific participation, it will need an approach as the one taken by Zhou, which conditionally deleted SCF from adipocytes, or co-cultured DNMT3A mut and wt cells with adipocytes or perform an ISH for IL-6 mRNA. I suggest avoiding any strong conclusions on the role of IL-6 secreted by adipocytes only. The experiments with the antibody are very informative, however it does not rule out the source of IL-6. The authors may benefit to include reference of increased levels of TNF alpha and IL-6 on myelodysplastic syndromes, specially by stromal cells, as MDS cells also harbored similar pre-leukemic mutations.

-Resolved: The author response to the comments and changes to the manuscript addressed this comment.

5.- I did not understand the context for the term durability (328).

-Resolved: The author changed the term.

Additional comments to the revised version:

I.- "Line 353- 355 We recognized that to date, there is no gold standard model that is specific on increasing the number of adipocytes. As we used external stress to increase FB it was crucial to us the PPAR γ i group to control for off targets effects of the sub-lethal irradiation on other cell types in the BM (stroma cells endothelial cells etc.)"

The authors relied on the use of PPAR gamma inhibitor to account for the unspecific nature of their model. However, it is important to consider that PPARgamma inhibitor, reduced, but not ablated the adipocytes after irradiation. This point is of relevance as some of the comments on the paper may be misleading. For example: Line 160,161: "Engraftment of sample #160005 cells was much higher under FBM conditions compared to normal BM (NBM) mice and the PPAR γ i-treated control, in which no adipocytes were accumulated (Figure 2b)." Figure 1f shows a significant number of adipocytes in the PPAR gamma inhibitor condition, the percentage is higher than the percentage detected by lipidTOX. However, the assessment by lipidTOX may be underestimated, as mature adipocytes are harder to isolate. I consider that as raised by other reviewer, the use of an in vitro co-culture system can make the paper stronger.

II.- In the revised version, the authors added a new paragraph
Line 147-148 "The decision to use an in vivo model rather than an in vitro model stemmed from the fact that in vivo model allow both human and mice stem cell self-renewal assays. The influence of the microenvironment is long-term, making in vitro experiments difficult to monitor" . I do not agree with the statement. I consider that the selection of an in vivo model does not prevent the inclusion of validation using in vitro models. I agreed on the use of in vivo models to evaluate HSC self renewal. However, they would benefit from co-cultures with adipocytes (with and without IL-6

inhibitors/ or CRISPR knock outs for IL-6) in order to validate their finding.

Minor comments:

Reference 2 and 3 are not in upper case.

Line 62 and 63, they both mentioned that adipocytes produced IL-6.

Revised the following sentence for concordance. Line78 "Positive effects of BM adipocytes include the ability to increase the capacity of adipocytes to sustain primitive hematopoietic cells in-vitro^{10,1}

The definition of S1e figures is low

Reviewer #2 (Remarks to the Author):

Zioni et al. have made minimal changes to their manuscript and the authors seem committed to their views over those of the reviewers and collective feedback.

The somewhat revised study continues to suffer from two central flaws that have also been recognized by other reviewers. First, there is no convincing demonstration that in vivo observations specifically contribute to BM adipocytes. Second, they have not developed a strong foundation to study Pre-LSCs that this reviewer or other reviewers can see, despite their opinion.

The mouse model used is for clonal hematopoiesis which is not necessarily a model for Pre-LSCs. Minimal experiments have been performed using human samples that could more convincingly represent Pre-LSCs, and these few human experiments also lacked important control groups.

These two limitations are a major concern when the central claims of their work revolve around the interaction of adipocytes with Pre-LSCs, and it is unclear why the authors have taken a unique hard stance of opinion vs taking a scientific sound viewpoint on the facts and data presented. I believe the reviewers have provided generous advice to improve the work and impact of their preliminary results.

Specific comments:

1. The story remains unfocused. Their abstract and introduction describe the study in the context of human hematopoiesis and understanding the etiology of leukemia by examining how phenotypes are selected by an aging microenvironment. However, they have not used aging models and in their rebuttal they suggest that they do not wish to perform human experiments with IL-6 because they do not make claims about the role of IL-6 in humans. If the conclusion of the study is not relevant to humans, then why would this have bearing on human leukemia?

2. In their rebuttal, the authors also defend that their choice of fatty BM models is appropriate because theirs is the first report to describe that castration or SGM-3 models act as inducers of fatty BM. Again, these observations are not germane to the goals they set out to investigate. The observation of fatty marrow in SGM3 mice is irrelevant to the central points around pre-leukemic HSPCs, as this model was never used for any further experimentation.

3. In their revised introduction, the authors incorrectly claim that "mice do not accumulate FBM in long bones with age" when they describe their choice of models. They cite Naveiras et al (Nature 2009) to support this statement but this paper never quantified adipocytes in long bones of mice at different ages. This issue has since been addressed by other studies, e.g., Figure 1 of Scheller et al. clearly shows a progressive accumulation of marrow adipocytes with progressing age in the tibia of several murine strains (Nature Communications 2015, 6:7808). It is concerning that the authors are unaware of the literature in this area, given the central focus on BM adipocytes as the foundation of their study. Or, perhaps they choose to ignore the literature for other reasons that are unclear to this reviewer.

4. Clonal hematopoiesis does not necessarily equate with a pre-LSC state (as discussed by numerous reviews, e.g., Sato et al., *Frontiers in Oncology* 2016, 6:187). As the authors admit, they “did not validate the acceleration of disease progression” in their chosen model. They simply describe that their mouse model has features of HSPC expansion and a pattern of hypomethylation. These are features of clonal hematopoiesis, which are not sufficient to claim that a cell is a pre-LSC. The cited paper for their mouse model does not contain the term pre-LSC anywhere.

5. The authors have addressed multiple comments by simply emphasizing the importance of their PPAR γ i control. The PPAR γ i control was only applied for one human patient sample and was omitted from the experiment with Patient #160005, as well as both of the DNMT3A wildtype controls. If it is such an essential control as the authors suggest in their rebuttal, it should be applied consistently throughout the paper. In the patient one case where the PPAR γ i control was indeed included, the DNMT3A^{mut} engraftment level was equivalent in NBM vs. FBM+ PPAR γ conditions despite the fact that FBM+ PPAR γ i has 100x more fatty marrow than their NBM condition (based on Fig 1e). If their hypothesis was correct, DNMT3A^{mut} engraftment should therefore engraft much less in the NBM condition as it has orders of magnitude fewer adipocytes than any other group. This discrepancy would suggest that the level BM adipocytes alone do not explain the engraftment ability of DNMT3A^{mut} HSPCs.

6. In their rebuttal, the authors address the criticism that no human DNMT3A wildtype control was used to validate their castration model. They then simply describe the two DNMT3A wildtype controls used in Figure 2g and 2h, but these were experiments with the irradiation model, not the castration model. This response seems disingenuous, as the criticism was specific to the castration model and the data for this specific model remain missing.

7. The authors have dismissed suggestions from multiple reviewers that genetic murine models would offer a more precise means of examining the role of BM adipocytes, and simply suggest that this should be tested in the future. The established precedent in the field is to either use genetic mouse models or at least in vitro co-culture assays to make claims about individual cell types in the microenvironment.

8. The authors are unwilling to perform simple in vitro experiments and continue to provide no direct evidence that IL-6 secretion is coming from adipocytes. The lack of IL-6 in the serum is insufficient to attribute local IL-6 secretion in BM to adipocytes following an insult as intense as total body irradiation. The authors suggest that because the radiation dose was sub-lethal, this is somehow rather a benign intervention and they can control for “all off target effects” with the addition of their PPAR γ i control. Sub-lethal irradiation is still a very substantial insult to the body that should not be disregarded.

9. The authors have declined to perform IL-6 experiments in xenograft models because they would like a referenced protocol to follow and unsure if IL-6 even reaches the BM when injected intraperitoneally. This is curious, as they have already used intraperitoneal injections of IL-6 for their Dnmt3a^{mut} mouse model of clonal hematopoiesis. Why would IL-6 reach the BM any differently if human cells are engrafted instead of mouse cells? The dose and administration schedule could be identical to that used for their mouse experiments. The fact that the authors have already performed such an experiment demonstrates that it is feasible.

10. The authors also argue that there would be some requirement to obtain human HSCs and not progenitors in order to perform CFU assays? They had no such concerns when performing CFU assays with mouse Dnmt3a^{mut} cells. If genotyping the colonies would be too much labor, then it seems the xenograft model would be a better choice for human experiments. If the authors instead prefer not to make claims that are relevant to humans, then the scope of the entire study needs to be reconsidered.

Reviewer #3 (Remarks to the Author):

The revised version of the manuscript by Zioni is an improvement.

The single-cell visualization (<https://tanaylab.weizmann.ac.il/FattyBM/>), as far as I can tell, does not contain sample annotations. The authors should add metadata to the web interface to allow users to compare the different groups (NBM, FBM, wt, mut, etc.).

Figure 4a: Has only one mouse been analyzed for NBM_mut and FBM_mut? It would be advisable to include more biological replicates.

Figure s10a: Please specify in the legend what cells were analyzed and what the grey (unselected) cells are.

Figure S10b: Please add P-values (and normalized enrichment scores).

Reviewer #4 (Remarks to the Author):

Point 1: I agree that PPAR γ inhibition is an important control to show that adipocytes play a role in promoting engraftment of DNMT3A mutant cells post irradiation. However, the adipocyte-mediated mechanism might act downstream or together with other aberrations caused by irradiation. In a model where adipocytes are specifically increased, different phenotypes might be observed. This is very related to the first point of reviewer 2, who elaborates on these issues in more detail. I was therefore asking for data from more specific models (or at least, from different models) and, at least, for a characterization of hematopoietic or "systemic" defects induced by their irradiation protocol. No further data is provided on that question. While it is clear that the model used is imperfect and artificial, it is therefore not clear if the findings generalise beyond this specific imperfect model. This remains a major limitation of the manuscript.

Point 2: The use of a PPAR γ inhibitor control does not prove that IL6 is produced by adipocytes, since it might also be produced by another cell type that gets stimulated by adipocytes.

Point 3: Convincingly addressed

Point 4: Convincingly addressed

Point 5: Explanation makes sense

Point 6: Not ideal. As I remarked earlier these analyses are confounded by changes in cell type abundance. This should at least explicitly be mentioned, if the cell numbers are insufficient to identify the consequences of the experimental treatments at the level of each cell type

Point 7: OK

Point 8: OK

RESPONSE TO REVIEWER'S COMMENTS

Reviewer #1

According to the concerns raised on the previous version of the paper I consider that the authors addressed all my comments and resolved all of them, but one. Please find my comments to the addressed points below.

1- Resolved: *the authors addressed all concerns and included the necessary information into the revised version.*

2- Resolved: *the authors added an additional figure showing that in their hands, they found that the increased in adipocyte content can last up to 2 months after irradiation.*

3 - Unresolved: *The authors included the necessary information to understand the method they used to assess adipocytes by lipidTOX and they also added additional figures that explained the evaluation of the FABP4 marker. However, the authors did not acknowledge that FABP4 can also stained endothelial cells. Line 128 states: "These results were validated by staining irradiated BM with the adipocyte marker fatty acid binding protein 4 (FABP4) (Figure 1f)". The authors relied on a differential expression of the intensity of FABP4 and the segmentation by image analysis. I strongly believe that is necessary to acknowledge that FABP4 is not a specific marker (that it also stained endothelial cells), but that the use of image analysis restricted the identification to adipocytes according to the intensity of the marker, as discussed in the letter to the reviewers. The authors did not take into consideration the advice to use a specific marker for adipocytes. Due to the relevance of this point to the paper, I do believe that the authors need to address this point.*

We thank the reviewer for this comment. We added a new data and figures with the adipocyte-specific marker Perilipin (which was suggested by the reviewer). (**New Figure 1g, New Figure S1d**). In the manuscript main text we have added the following sentence: These results were validated by staining... **"and with perilipin, which coats lipid droplets in adipocytes (Figure 1g)."**

We also included figure S1d, in which we demonstrate using the Perilipin staining, adipocyte accumulation after castration, which was decreased after treatment with PPAR γ i. In the main text we added: **"These findings were validated by staining bones from castrated mice with Perilipin (Figure S1d), which revealed an increase in adipocytes following castration and a substantial decrease after PPAR γ i treatment. (Figure S1d). We now added in the Materials and Methods of the Perilipin staining: RabbitPerilipin-1 (D418) Antibody, CellSignaling Technology Cat# 3470, RRID:AB_2167268, AB_2340436 and Alexa Fluor[®]488 Donkey Anti-RabbitCat#711-545-152,RRID:AB_2313584, Jackson ImmunoResearch. We also updated Figure 1 legend: Representative 3D whole-mount immunofluorescence staining of epiphyseal-metaphyseal BM long bones derived from control, NBM, FBM and FBM & PPAR γ i treated NSG mice. Adipocytes are depicted by Perilipin expression and by distinctive unilocular morphology in the DIC (differential interference contrast) channel. DAPI in blue. Adipocytes are additionally marked by red dots.1 unit Scale Bar:70.99 μ m; n=12-14708 μ m x 708 μ m x30-50 μ m stacked images from 3 miceand 2 bones (femur, tibia) each group. And, a legend for Figure s1d: Representative 3D whole-mount immunofluorescence staining of epiphyseal-metaphyseal BM long bonesderived from Castrated NSG mice and castrated mice treated with PPAR γ i. Adipocytes are depicted by Perilipin expression and by distinctive unilocular morphology in the DIC (differential interference contrast) channel. DAPI in blue.**

Adipocytes are additionally marked by red dots. 1 unit Scale Bar: 70.99 μm; n=12-14708 μm x 708 μm x 30-50 μm stacked images from 3 mice and 2 bones (femur, tibia) each group

4.- Resolved: *The authors labeled the figures and provided a logic answer to the MPO point. However, it is important to consider that the number of cells analyzed by scRNAseq is very low. As an additional suggestion, the authors may add the confidence on the call of hematopoietic hierarchy.*

Based on the reviewer comment we repeated the single cell RNAseq experiment with the following conditions: NBM_mut and FBM_mut, however we have moved to a 10X platform for the scRNAseq which allowed us to study more cells ~1500 for each condition. We have now replicated the results of Figure 4 not just with a biological replicate and more cells but also with a different library prep platform. We have added all these results to Figure s14. Figure s14a,b demonstrates significantly different distribution of HSPCs subpopulations most of the signal arising from 6 fold increase in HSCs in the FBM_mut cells. Based on the result we have modified the sentence in the abstract regarding this result. **“A 6-10 fold increase in DNMT3A^{Mut}-HSCs was observed under FBM conditions in comparison to normal bone marrow”**

We further validated the significantly increased IL-6 pathway in the FBM_mut condition compared to NBM_mut (Figure s14c).

5-Resolved: *The author response to the comments and changes to the manuscript addressed this comment.*

6 -Resolved: *The author changed the term.*

Additional comments to the revised version:

1. - *“Line 353- 355 We recognized that to date, there is no gold standard model that is specific on increasing the number of adipocytes. As we used external stress to increase FB it was crucial to us the PPARγi group to control for off targets effects of the sub-lethal irradiation on other cell types in the BM (stroma cells endothelial cells etc.)”*

The authors relied on the use of PPAR gamma inhibitor to account for the unspecific nature of their model. However, it is important to consider that PPARgamma inhibitor, reduced, but not ablated the adipocytes after irradiation. This point is of relevance as some of the comments on the paper may be misleading. For example: Line 160,161: “Engraftment of sample #160005 cells was much higher under FBM conditions compared to normal BM (NBM) mice and the PPARγi-treated control, in which no adipocytes were accumulated (Figure 2b).” Figure 1f shows a significant number of adipocytes in the PPAR gamma inhibitor condition, the percentage is higher than the percentage detected by lipidTOX.

We appreciate the reviewer's input. We agree that adipocytes can be found in NBM. As demonstrated in Figure 1e, we repeated the LipidTOX test on NBM samples. Figures 1e-1g reveal that adipocytes are not completely eliminated in the PPARγi -treated control. As a result, we changed the text: “Engraftment of sample #160005 cells was much higher under FBM conditions compared to normal BM (NBM) mice and the PPARγi-treated control, in which **less** adipocytes were accumulated (Figure 2b)”

2. However, the assessment by lipidTOX may be underestimated, as mature adipocytes are harder to isolate. I consider that as raised by other reviewer, the use of an *in vitro* co-culture system can make the paper stronger.

We thank the reviewer for his comment. To support our findings in the best way (as was suggested by the reviewers) we decided to set up *in vitro* systems in which we co-culture either undifferentiated bone marrow derived mesenchymal cells (MSCs) or adipocytes with *DNMT3A*^{mut} cells and compared the colony forming ability of *DNMT3A*^{mut} HSCs under the different conditions. In the first step we provide evidence that we were able to establish an *in vitro* mouse and human adipocyte culture conditions which were differentiated from MSCs (**New Fig S12e, f**). Oil red staining was utilized to quantify adipocytes. Next, we collected the media after ten days of adipocyte/MSC culture and analyzed IL-6 levels. IL-6 levels were significantly higher in both mouse (**New Fig 5c**) and human (**New Fig 5e**) derived adipocyte media.

Finally, *DNMT3A*^{mut} BM-derived Lin⁻ cells from one year old mice were co cultured with adipocytes/MSCs. After ten days of co-culture, we transferred the cells to perform a Colony Forming Unit (CFU) assay in which cells were plated in methylcellulose with adipocytes or MSCs-derived media. *DNMT3A*^{mut} mice HSPCs co-cultured with adipocytes produces significantly more colonies than co-culturing with MSCs (**New Fig 5d**). This experiment was replicated with *DNMT3A* mutated CD34+ cells derived from four human AML samples: sample #141164 (**New Figure 5f**), sample #141467 (**New Figure S12g**), #150279 (**New Figure S12h**) and sample #141464 (**New Figure S12i**). Co-culturing human *DNMT3A* mutant cells with adipocytes yielded more colonies. We have genotype 13 colonies and all of them were positive for *DNMT3A* and negative for *NPM1c* suggesting that they originate from preL-HSPCs (**New Fig 5g**). Altogether these findings suggest that adipocytes secreting IL-6 (and possibly other factors), provide selective advantage to both human and mouse HSPCs carrying *DNMT3A* mutations *in vitro*. We added in the text the following paragraph: “Furthermore, we set up experiments in which we cultured human/mouse bone marrow derived mesenchymal stem cells (MSCs) and adipocytes *in vitro* (Figure S12e, f). Adipocytes were quantified using oil red staining. After ten days in culture (without HSPCs), we collected the media and found significantly elevated IL-6 levels in the adipocyte cultures compared to MSCs cultures in both mouse (Figure 5c) and human (Figure 5e). Next, one-year old mice *DNMT3A*^{mut} BM-derived Lin⁻ cells were co cultured with adipocytes/MSCs. After ten days, we used Colony Forming Unit (CFU) assay in which cells were seeded in methylcellulose with adipocytes or MSCs-derived media. *DNMT3A*^{mut} HSPCs co-cultured with adipocytes, produced significantly more colonies than co-culturing with MSCs (Figure 5d). This experiment was replicated with *DNMT3A* mutated CD34+ cells derived from four human AML samples: sample #141164 (Figure 5f), sample #141467 (Figure S12g), #150279 (Figure S12h) and sample #141464 (Figure S12i). Again, co-culturing human *DNMT3A* mutant CD34+ cells with adipocytes yielded significantly more colonies than co-culturing with MSCs (Figure 5f, 5g, S12g-i). We have genotype 13 of the colonies co-cultured with adipocytes and all of them were positive for *DNMT3A* and negative for *NPM1c* suggesting that they originate from preL-HSPCs (Figure 5g). Altogether these findings suggest that adipocytes secreting IL-6 (and possibly other factors), provide selective advantage to both human and mouse HSPCs carrying *DNMT3A* mutations *in vitro*. “

We also added in M&M:

Human MSCs: primary bone marrow derived human MSCs were generously receive from RAMBM hospital. The cells were thawed and suspended with a suitable medium which contains MEM α , 1% P/S, 1% L-glu, 10% FBS (MSCs media).

Human Adipocytes differentiation *in vitro*: The primary human MSCs were seeded on 96 well plates with MSCs medium at 10⁴ cells/well. After two days, when reached confluence, the medium was changed with MesenCult Adipogenic Diff Kit, Human (stem cell, cat#05412) for 10-14 days until the MSCs differentiated into mature adipocytes.

Mouse MSCs production: MSCs were generated from the tibia and femur of two-month-old C57BL/6 mice. The bones were flushed, and cells were seeded in MSC media in a 6-well plate. Every other day, the medium was replaced.

Mouse Adipocytes differentiation *in vitro*: The primary mouse MSCs were seeded on 96 well plates at 10⁴ cells per well with MSCs cell media. After two days, the cells reached full confluence, and the medium was replaced with MesenCult Adipogenic Diff Kit, mouse (stem cell, cat#05507) for 10-14 days, until the MSCs differentiated into mature adipocytes.

Oil Red Staining: Cells were washed twice with PBSX1, fixed with 4% PFA and washed twice with distilled water. Oil red was added for one hour.

We also added in the discussion

“We provide evidence that the addition of IL-6 to the CFU assay increase the clonogenic capacity of preL-HSPCs. Furthermore *DNMT3A* mutated mice and human HSPCs have a selective advantage if co-cultured *in vitro* with bone marrow derived adipocytes which also secret IL-6 (Figure 5).”

3.- In the revised version, the authors added a new paragraph

*Line 147-148 “The decision to use an *in vivo* model rather than an *in vitro* model stemmed from the fact that *in vivo* model allow both human and mice stem cell self-renewal assays. The influence of the microenvironment is long-term, making *in vitro* experiments difficult to monitor” . I do not agree with the statement. I consider that the selection of an *in vivo* model does not prevent the inclusion of validation using *in vitro* models. I agreed on the use of *in vivo* models to evaluate HSC self renewal. However, they would benefit from co-cultures with adipocytes (with and without IL-6 inhibitors/ or CRISPR knock outs for IL-6) in order to validate their finding.*

We thank the reviewer for this comment. We have now provided new evidence that *in vitro* bone marrow derived adipocytes secrete significantly more IL-6 in comparison to MSCs in both human and mice (**new Fig 5c,e**). We have further provided evidence that IL-6 increase the *in vitro* clonogenic capacity of *DNMT3A*^{Mut} mice HSPCs (**Figure 5a,b**). To further support these results in humans we have now performed an *in vitro* co-culture experiment (as described above). In three different human samples preL-HSPCs carrying *DNMT3A* mutations demonstrated significantly increased clonogenic capacity if grown on adipocytes. The combination of these experiments suggest that IL-6 secreted by FBM provide *in vitro* selective advantage to *DNMT3A* mutated HSPCs, which is what we claim.

Minor comments:

We thank the reviewer for his comments.

1. *Reference 2 and 3 are not in upper case.* – was corrected

2. *Line 62 and 63, they both mentioned that adipocytes produced IL-6.* - We changed the text to: Gene expression analysis of BM adipocytes suggested that they have distinct immune regulatory properties and high expression of pro-inflammatory cytokines (~~IL1A~~, IL1B, ~~IL-6, IL8~~, IL15, ~~IL18~~ and stem cell factor (SCF)^{5,6}. **“Furthermore, BM adipocytes secrete IL6, IL8 and TNF α ”**

3. *Revised the following sentence for concordance. Line78 “Positive effects of BM adipocytes include the ability to ncrease the capacity of adipocytes to sustain primitive hematopoietic cells in-vitro10,11”* - We changed this sentence to : **“The ability of BM adipocytes to support primitive hematopoietic cells *in-vitro* is one of its beneficial effects.”**

4. *The definition of S1e figures is low:* We added in the legend: **NSG-hSCF (NSG mice that express human membrane-bound stem cell factor) or NSG-SGM3 mice (expressing human IL3, GM-CSF (CSF2) and SCF (KITLG))**

Reviewer #2

Zioni et al. have made minimal changes to their manuscript and the authors seem committed to their views over those of the reviewers and collective feedback.

The somewhat revised study continues to suffer from ten central flaws that have also been recognized by other reviewers. First, there is no convincing demonstration that in vivo observations specifically contribute to BM adipocytes. Second, they have not developed a strong foundation to study Pre-LSCs that this reviewer or other reviewers can see, despite their opinion.

The mouse model used is for clonal hematopoiesis which is not necessarily a model for Pre-LSCs. Minimal experiments have been performed using human samples that could more convincingly represent Pre-LSCs, and these few human experiments also lacked important control groups.

These ten limitations are a major concern when the central claims of their work revolve around the interaction of adipocytes with Pre-LSCs, and it is unclear why the authors have taken a unique hard stance of opinion vs taking a scientific sound viewpoint on the facts and data presented. I believe the reviewers have provided generous advice to improve the work and impact of their preliminary results.

Specific comments:

1. *The story remains unfocused. Their abstract and introduction describe the study in the context of human hematopoiesis and understanding the etiology of leukemia by examining how phenotypes are selected by an aging microenvironment. However, they have not used aging models and in their rebuttal they suggest that they do not wish to perform human experiments with IL-6 because they do not make claims about the role of IL-6 in humans. If the conclusion of the study is not relevant to humans, then why would this have bearing on human leukemia?*

We thank the reviewer for this comment. We have modified the abstract so it is more focused on FBM and its interaction with CH. We omitted preL-HSPCs from the abstract as we understand the reviewer does not agree with our definitions to this term, and as we do not explain this in detail in the abstract other might not understand /agree with this. Although we insist that this is a matter of terminology and we were one of the first groups to suggest it and some people in the field do accept the definition that an HSC carrying a leukemia related mutation but still capable of differentiation is by definition preL-HSPC although it will not necessarily lead to leukemia. We will explain in detail in response to comment #3 from reviewer 2 why we did not use aged NSG mice. We did use aged *DNMT3A* mice and demonstrated a stronger selective advantage under FBM conditions. We have also modified our focus on the models as we now have data from both mice and human on the castration model as will be presented in response to comment #2 from reviewer 2 and updated this in the modified abstract. The reason we do not make claims on human IL-6 is due to technical limitations of the mice models and the fact that we cannot provide strong claims based on *in vivo* stem cell assays. We believe that once an IL-6 humanized NSG mice will be available to us we will perform such experiments. Anyhow as some of the reviewers asked we know provide evidence that BM derived adipocyte *in vitro* secrete significantly higher levels of IL-6 and support human preL-HSPCs carrying *DNMT3A* mutations *in vitro* (response to additional comments to the revised version, comment #2 from reviewer 1 pages 3-4 in this rebuttal). We took all of the above and modified the abstract accordingly.

“Accumulation of fatty bone marrow (FBM) is one of the key age related changes possibly influencing the blood system. While a link between obesity and cancer evolution has been reported it remains unknown whether FBM can modify the evolution of the early stages of leukemia and clonal hematopoiesis (CH). To address this question, we established different FBM mouse models in immunodeficient mice in whom we can study both mouse and human cells. We focused our studies on two FBM models 1) after sublethal irradiation; 2) after castration; and in both we used an adipogenesis inhibitor as a control (PPAR γ inhibitor). We transplanted both human and mice hematopoietic stem cells (HSCs) carrying *DNMT3A* mutations into immunodeficient mice with FBM. A significant increase in self-renewal was found when *DNMT3A*^{Mut}-HSCs were exposed to FBM. To better understand the mechanisms of the FBM-CH interaction, we performed single cell RNA-sequencing on HSPCs after FBM exposure *in vivo*. A 6-10 fold increase in *DNMT3A*^{Mut}-HSCs was observed under FBM conditions in comparison to normal bone marrow. Mutated HSCs from mice exposed to FBM exhibited an activated inflammatory signaling (IL-6 and IFN γ). Cytokine analysis of BM fluid and BM derived adipocytes grown *in vitro* demonstrated increased IL-6 levels under FBM conditions. Anti-IL-6 neutralizing antibodies significantly reduced the selective advantage of mice derived *DNMT3A*^{Mut}-HSCs exposed to FBM. Overall, paracrine FBM inflammatory signals promote *DNMT3A*-driven clonal hematopoiesis, which can be inhibited by blocking the IL-6 receptor.”

We have also modified the introduction to answer comment #3 from reviewer 2.

2. In their rebuttal, the authors also defend that their choice of fatty BM models is appropriate because theirs is the first report to describe that castration or SGM-3 models act as inducers of fatty BM. Again, these observations are not germane to the goals they set out to investigate. The observation of fatty marrow in SGM3 mice is irrelevant to the central points around pre-leukemic HSPCs, as this model was never used for any further experimentation.

We appreciate the reviewer's comment. We developed numerous FBM models in this study. As stated in the text, "while the one-year-old NSG-SGM3 model had high FBM levels, this model was linked to a

reduction in the self-renewal capacity of normal HSCs and thus we did not use it in our future analysis." We do not fully understand why the reviewer is not interested to include these data in the manuscript. We still believe that this information should be published as yet another possible model for FBM. For example we have some preliminary results in the lab that *SRSF2* mutated samples grow better in NSG-SGM3 mice. However, as we will explain later, working with one year old NSG mice (aged) is problematic.

Furthermore, in the current manuscript version, we introduced new results on the castration (CAS) model which leads to FBM and a new *in vitro* FBM model.

1. **New Fig 2c**, in which PPAR γ i was administered to (CAS) FBM mice, resulting in a significant decrease in the engraftment of human AML sample #160005 cells, we added in the text: "The administration of PPAR γ i to (CAS) FBM mice resulted in a significant decrease of sample #160005 cells engraftment (Figure 2c)".

2. **New Fig 3f**, we transplanted *DNMT3A*^{Mut} cells derived from one-year-old mice injected into NBM, FBM(CAS) and CAS+PPAR γ i control and compared *DNMT3A*^{WT} derived BM. We added in the text "We repeated this experiment on the castration (CAS) FBM model and again a significantly higher engraftment of *DNMT3A*^{Mut} cells derived from one-year-old mice injected into FBM-CAS in comparison to NBM and PPAR γ i control was detected. FBM-CAS had no effect on *DNMT3A*^{WT} cells (Figure 3f)".

3. We thank the reviewer for his comment. To support our findings in the best way (as was suggested by the reviewers) we decided to set up *in vitro* systems in which we co-culture either undifferentiated bone marrow derived mesenchymal cells (MSCs) or adipocytes with *DNMT3A*^{Mut} cells and compared the colony forming ability of *DNMT3A*^{Mut} HSCs under the different conditions. In the first step we provide evidence that we were able to establish an *in vitro* mouse and human adipocyte culture conditions which were differentiated from MSCs (**New Fig S12e, f**). Oil red staining was utilized to quantify adipocytes. Next, we collected the media after ten days of adipocyte/MSC culture and analyzed IL-6 levels. IL-6 levels were significantly higher in both mouse (**New Fig 5c**) and human (**New Fig 5e**) derived adipocyte media.

Finally, *DNMT3A*^{Mut} BM-derived Lin⁻ cells from one year old mice were co cultured with adipocytes/MSCs. After ten days of co-culture, we transferred the cells to perform a Colony Forming Unit (CFU) assay in which cells were plated in methylcellulose with adipocytes or MSCs-derived media. *DNMT3A*^{Mut} mice HSPCs co-cultured with adipocytes produces significantly more colonies than co-culturing with MSCs (**New Fig 5d**). This experiment was replicated with *DNMT3A* mutated CD34+ cells derived from four human AML samples: sample #141164 (**New Figure 5f**), sample #141467 (**New Figure S12g**), #150279 (**New Figure S12h**) and sample #141464 (**New Figure S12i**). Co-culturing human *DNMT3A* mutant cells with adipocytes yielded more colonies. We have genotype 13 colonies and all of them were positive for *DNMT3A* and negative for *NPM1c* suggesting that they originate from preL-HSPCs (**New Fig 5g**). Altogether these findings suggest that adipocytes secreting IL-6 (and possibly other factors), provide selective advantage to both human and mouse HSPCs carrying *DNMT3A* mutations *in vitro*. We added in the text the following paragraph: "Furthermore, we set up experiments in which we cultured human/mouse bone marrow derived mesenchymal stem cells (MSCs) and adipocytes *in vitro*

(Figure S12e, f). Adipocytes were quantified using oil red staining. After ten days in culture (without HSPCs), we collected the media and found significantly elevated IL-6 levels in the adipocyte cultures compared to MSCs cultures in both mouse (Figure 5c) and human (Figure 5e). Next, one-year old mice *DNMT3A*^{Mut} BM-derived Lin⁻ cells were co cultured with adipocytes/MSCs. After ten days, we used Colony Forming Unit (CFU) assay in which cells were seeded in methylcellulose with adipocytes or MSCs-derived media. *DNMT3A*^{Mut} HSPCs co-cultured with adipocytes, produced significantly more colonies than co-culturing with MSCs (Figure 5d). This experiment was replicated with *DNMT3A* mutated CD34+ cells derived from four human AML samples: sample #141164 (Figure 5f), sample #141467 (Figure S12g), #150279 (Figure S12h) and sample #141464 (Figure S12i). Again, co-culturing human *DNMT3A* mutant CD34+ cells with adipocytes yielded significantly more colonies than co-culturing with MSCs (Figure 5f, 5g, S12g-i). We have genotype 13 of the colonies co-cultured with adipocytes and all of them were positive for *DNMT3A* and negative for *NPM1c* suggesting that they originate from preL-HSPCs (Figure 5g). Altogether these findings suggest that adipocytes secreting IL-6 (and possibly other factors), provide selective advantage to both human and mouse HSPCs carrying *DNMT3A* mutations *in vitro*.”

We also added in M&M:

Human MSCs: primary bone marrow derived human MSCs were generously receive from RAMBM hospital. The cells were thawed and suspended with a suitable medium which contains MEM α , 1% P/S, 1% L-glu, 10% FBS (MSCs media).

Human Adipocytes differentiation *in vitro*: The primary human MSCs were seeded on 96 well plates with MSCs medium at 10⁴ cells/well. After two days, when reached confluence, the medium was changed with MesenCult Adipogenic Diff Kit, Human (stem cell, cat#05412) for 10-14 days until the MSCs differentiated into mature adipocytes.

Mouse MSCs production: MSCs were generated from the tibia and femur of two-month-old C57BL/6 mice. The bones were flushed, and cells were seeded in MSC media in a 6-well plate. Every other day, the medium was replaced.

Mouse Adipocytes differentiation *in vitro*: The primary mouse MSCs were seeded on 96 well plates at 10⁴ cells per well with MSCs cell media. After two days, the cells reached full confluence, and the medium was replaced with MesenCult Adipogenic Diff Kit, mouse (stem cell, cat#05507) for 10-14 days, until the MSCs differentiated into mature adipocytes.

Oil Red Staining: Cells were washed twice with PBSX1, fixed with 4% PFA and washed twice with distilled water. Oil red was added for one hour.

We also added in the discussion

“We provide evidence that the addition of IL-6 to the CFU assay increase the clonogenic capacity of preL-HSPCs. Furthermore *DNMT3A* mutated mice and human HSPCs have a selective advantage if co-cultured *in vitro* with bone marrow derived adipocytes which also secret IL-6 (Figure 5).”

3. In their revised introduction, the authors incorrectly claim that “mice do not accumulate FBM in long bones with age” when they describe their choice of models. They cite Naveiras et al (Nature 2009) to support this statement but this paper never quantified adipocytes in long bones of mice at different ages. This issue has since been addressed by other studies, e.g., Figure 1 of Scheller et al. clearly shows a progressive accumulation of marrow adipocytes with progressing age in the tibia of several murine strains (Nature Communications 2015, 6:7808). It is concerning that the authors are unaware of the literature in this area, given the central focus on BM adipocytes as the foundation of their study. Or, perhaps they choose to ignore the literature for other reasons that are unclear to this reviewer.

We thank the reviewer for his important comment. We agree with the reviewer that the Naveiras et al. study does not support the statement "mice do not accumulate FBM in long bones with age." As we were not aware of the study by Scheller et.al Nat Comm 2015 (doi: 10.1038/ncomms8808) we did not include it initially. We thank the reviewer for highlighting this important manuscript. As the reviewer correctly suggest in the 2 mice strains studied by Scheller et.al one can observe age related accumulation of fat however with different dynamics between the strains in different behavior with age in different regions of the tibia between strains. As we chose to study FBM in NSG to allow us to study both human and mice in a similar model, one of our first experiments was to age NSG, NSG-HCF and SGM3 mice (Figure s1f). For reasons we can explain and as can be demonstrated in Figure s1f the 1 year old NSG mice did not demonstrated tibia fat accumulation, and that was the one of the reasons we looked for methods to increase FBM in NSG mice with irradiation and castration. Based on the study by Scheller et.al we can now hypothesize that the NSG is yet another strain with different dynamics of age related FBM accumulation. Furthermore, aging of NSG mice is problematic. A recent comprehensive analysis of NSG mice demonstrated that their median survival is 52 weeks and at death 50% of them have malignant tumors (doi: [10.1177/0300985817698210](https://doi.org/10.1177/0300985817698210)). This fact makes it problematic to use aged NSGs as a model. Aged SGM3 mice could potentially be studied however we hypothesize that a similar age related malignancy would be present also, but have not been demonstrated yet. Altogether, if one wants to study both human and mice in a similar model our approach seems reasonable, especially now that we have added the new data on the castration of NSG mice.

Based on these we have now modified our introduction:

" In mice FBM accumulates in the tibia with age, however strain has a considerable influence on the number of adipocytes and their age related dynamics¹⁹. As in the current study we aimed at studying both human and mice preL-HSPCs we chose immunodeficient mice models, however such mice have a shorter life span and tend to develop malignant tumors at a median age of 52 weeks thus aging them would be biased²⁰. Therefore we searched for other options to increase FBM in immunodeficient mice."

4. Clonal hematopoiesis does not necessarily equate with a pre-LSC state (as discussed by numerous reviews, e.g., Sato et al., Frontiers in Oncology 2016, 6:187). As the authors admit, they “did not validate the acceleration of disease progression” in their chosen model. They simply describe that their mouse model has features of HSPC expansion and a pattern of hypomethylation. These are features of clonal hematopoiesis, which are not sufficient to claim that a cell is a pre-LSC. The cited paper for their mouse model does not contain the term pre-LSC anywhere.

We have modified the abstract so it is more focused on FBM and its interaction with CH. We omitted preL-HSPCs as we understand the reviewer does not agree with our definitions to this term, and as we do not explain this in detail in the abstract other might not understand /agree with this. Although we insist that this is a matter of terminology and we were one of the first groups to suggest it and some people in the field do accept the definition that an HSC carrying a leukemia related mutation but still capable of differentiation is by definition preL-HSPC although it will not necessarily lead to leukemia.

Here is a quote from Blood. 2018;131(5):496-504 from a review on the subject “The clear evidence that ARCH can be a preleukemic condition was provided by the identification of preleukemic stem cells (pre-L-HSPCs). These cells carried mutations that were also observed among healthy individuals and in mature, non-leukemic cells.” The distinction the reviewer makes between such cells in mice and human is plausible.

5. *The authors have addressed multiple comments by simply emphasizing the importance of their PPAR γ i control. The PPAR γ i control was only applied for one human patient sample and was omitted from the experiment with Patient #160005, as well as both of the DNMT3A wildtype controls.*

We appreciate the reviewer's input. We included a patient #160005 experiment in which we employed PPAR γ i control in both the irradiated (figure 2b) and castrated models (**New figure 2C**). Engraftment was reduced in both cases when PPAR γ i control was administered.

If it is such an essential control as the authors suggest in their rebuttal, it should be applied consistently throughout the paper. In the patient one case where the PPAR γ i control was indeed included, the DNMT3A^{mut} engraftment level was equivalent in NBM vs. FBM+ PPAR γ conditions despite the fact that FBM+ PPAR γ i has 100x more fatty marrow than their NBM condition (based on Fig 1e). If their hypothesis was correct, DNMT3A^{mut} engraftment should therefore engraft much less in the NBM condition as it has orders of magnitude fewer adipocytes than any other group. This discrepancy would suggest that the level BM adipocytes alone do not explain the engraftment ability of DNMT3A^{mut} HSPCs.

We thank the reviewer for bringing this to our attention. We conducted further NBM and FBM+ PPAR γ i image stream analysis to persuade the reviewer of our idea (**New Figure 1e**). Figure 1e demonstrates that after PPAR γ i treatment, the number of adipocytes is reduced almost to the level of NBM. Furthermore, we do not argue that the number of BM adipocytes alone explains the ability to engraft, but rather the inflammatory signals from the fatty marrow that might affect the early evolution of CH. And we do not claim this is the only mechanism there might be others.

6. *In their rebuttal, the authors address the criticism that no human DNMT3A wildtype control was used to validate their castration model. They then simply describe the ten DNMT3A wildtype controls used in Figure 2g and 2h, but these were experiments with the irradiation model, not the castration model. This response seems disingenuous, as the criticism was specific to the castration model and the data for this specific model remain missing.*

We thank the reviewer for this comment. In the current manuscript version, we introduced new results on the castration (CAS) model which leads to FBM:

1. New Fig 2c, in which PPAR γ i was administered to (CAS) FBM mice, resulting in a significant decrease in the engraftment of human AML sample #160005 cells, we added in the text: “**The administration of PPAR γ i to (CAS) FBM mice resulted in a significant decrease of sample #160005 cells engraftment (Figure 2c)**”.

2. New Fig 3f, we transplanted DNMT3A^{mut} cells derived from one-year-old mice injected into NBM, FBM(CAS) and CAS+PPAR γ i control and compared DNMT3A^{WT} derived BM. We added in the text “We repeated this experiment on the castration (CAS) FBM model and again a significantly higher engraftment of DNMT3A^{mut} cells derived from one-year-old mice injected into FBM-CAS in comparison to NBM and PPAR γ i control was detected. FBM-CAS had no effect on DNMT3A^{WT} cells (Figure 3f)”.

7. The authors have dismissed suggestions from multiple reviewers that genetic murine models would offer a more precise means of examining the role of BM adipocytes, and simply suggest that this should be tested in the future. The established precedent in the field is to either use genetic mouse models or at least in vitro co-culture assays to make claims about individual cell types in the microenvironment.

We thank the reviewer for this comment. As it was raised by reviewer 1 also we have answered this in detail in response to Additional comments to the revised version, comment #2 by reviewer #1 (pages 3-3).

8. The authors are unwilling to perform simple in vitro experiments and continue to provide no direct evidence that IL-6 secretion is coming from adipocytes. The lack of IL-6 in the serum is insufficient to attribute local IL-6 secretion in BM to adipocytes following an insult as intense as total body irradiation. The authors suggest that because the radiation dose was sub-lethal, this is somehow rather a benign intervention and they can control for “all off target effects” with the addition of their PPAR γ i control. Sub-lethal irradiation is still a very substantial insult to the body that should not be disregarded.

We thank the reviewer for this comment. As it was raised by reviewer 1 also we have answered this in detail in response to Additional comments to the revised version, comment #2 by reviewer #1 (pages 3-3).

9. The authors have declined to perform IL-6 experiments in xenograft models because they would like a referenced protocol to follow and unsure if IL-6 even reaches the BM when injected intraperitoneally. This is curious, as they have already used intraperitoneal injections of IL-6 for their Dnmt3amut mouse model of clonal hematopoiesis. Why would IL-6 reach the BM any differently if human cells are engrafted instead of mouse cells? The dose and administration schedule could be identical to that used for their mouse experiments. The fact that the authors have already performed such an experiment demonstrates that it is feasible.

We thank the reviewer for this comment but we do not fully understand it. We have never injected IL-6 but rather IL-6 neutralizing antibodies which are stable *in vivo* and known to be functional. We have only added IL-6 *in vitro* and calibrating IL-6 injection *in vivo* is not an easy task. While such an experiment might be of value it is out of the score of the current manuscript.

10. The authors also argue that there would be some requirement to obtain human HSCs and not progenitors in order to perform CFU assays? They had no such concerns when performing CFU assays with mouse Dnmt3amut cells. If genotyping the colonies would be too much labor, then it seems the xenograft model would be a better choice for human experiments. If the authors instead prefer not to make claims that are relevant to humans, then the scope of the entire study needs to be reconsidered.

We thank the reviewer for his comment. We established experiments in which we cultured *vitro* human MSCs and adipocytes and performed human CFU assay and genotyping, all the results are detailed in response to additional comments to the revised version, comment #2 by reviewer #1 (page 3-4).

Reviewer#3:

The revised version of the manuscript by Zioni is an improvement.

1. The single-cell visualization (<https://tanaylab.weizmann.ac.il/FattyBM/>), as far as I can tell, does not contain sample annotations. The authors should add metadata to the web interface to allow users to compare the different groups (NBM, FBM, wt, mut, etc.).

We thank the reviewer for this comment we have now added the metadata to the single-cell visualization tool. The original RNAseq data and the new data for this revision can be found in the following link <https://tanaylab.weizmann.ac.il/MCV/FBM/>.

2. Figure 4a: Has only one mouse been analyzed for NBM_mut and FBM_mut? It would be advisable to include more biological replicates.

We thank the reviewer for this comment. First more than one mice was used per condition however the mice were pooled together to a single sequencing experiment as the reviewer correctly points out. Based on the reviewer comment we repeated the single cell RNAseq experiment with the following conditions: NBM_mut and FBM_mut, however we have moved to a 10X platform for the scRNAseq which allowed us to study more cells ~1300 for each condition. We have now replicated all the results of figure 4 not just with a biological replicate and more cells but also with a different library prep platform. We have added all these results to Figure s14. Figure s14a,b demonstrates significantly different distribution of HSPCs subpopulations most of the signal arising from 6 fold increase in HSCs in the FBM_mut cells. Base on the result we have modified the sentence in the abstract regarding this result.

“A 6-10 fold increase in *DNMT3A*^{Mut}-HSCs was observed under FBM conditions in comparison to normal bone marrow”.

We have also edited the results section accordingly:

“We have repeated the scRNA-seq experiment with more cells and with the 10X genomics platform from *DNMT3A*^{Mut} cells exposed to FBM or NBM and validated the significantly increased HSCs population in the *DNMT3A*^{Mut} cells exposed to FBM (this time 6 fold more HSCs than under NBM conditions) (Figure s14a,b).”

We further validated the significantly increased response to IL-6 and and IFN γ pathways in the FBM_mut condition compared to NBM_mut (Figure s14c-f). We have added this text:

“We have repeated the scRNA-seq experiment with more cells and with the 10X genomics platform from *DNMT3A*^{Mut} cells exposed to FBM or NBM and validated that exposure to FBM activated the IFN γ , and IL-6 pathways in *DNMT3A*^{Mut} derived from one year old mice (Figure 14c-f).”

We have also updated the methods section:

“We have repeated the scRNA-seq experiment for validation of our results. One-year-old *DNMT3A^{Mut}* cells were injected to FBM and NBM mice. Then, three days after injection CD45.2 LSK cells were isolated. Cells were sorted to low binding Eppendorf tubes and library was prepared by the 10X genomics V3 3’ kit. Libraries were sequenced by Novaseq. For the *DNMT3A^{Mu}* FBM condition we analyzed 1228 cells with median coverage of 60,000 reads per cell and 9440 UMIs per cells. For the *DNMT3A^{Mu}* NBM condition we analyzed 1405 cells with median coverage of 195,000 reads per cell and 9469 UMIs per cells. We have performed the same bioinformatics analysis as in the first experiment.”

3. *Figure s10a: Please specify in the legend what cells were analyzed and what the grey (unselected) cells are.*

We thank the reviewer for this comment and added the grey cell definition into Figure s10a legend. “All other cells which include Mut and Wt cells on normal bone marrow (NBM); naïve cells- are cells extracted directly from BM of respective mice without transplantation and cre control are included in the Unselected group and are marked in grey.”

4. *Figure S10b: Please add P-values (and normalized enrichment scores).*

We thank the reviewer for this comment and added this information from Table a3 to Figure s10b, and to the legend also.

Reviewer #4

Point 1: I agree that PPAR γ inhibition is an important control to show that adipocytes play a role in promoting engraftment of DNMT3A mutant cells post irradiation. However, the adipocyte-mediated mechanism might act downstream or together with other aberrations caused by irradiation. In a model where adipocytes are specifically increased, different phenotypes might be observed. This is very related to the first point of reviewer 2, who elaborates on these issues in more detail. I was therefore asking for data from more specific models (or at least, from different models) and, at least, for a characterization of hematopoietic or “systemic” defects induced by their irradiation protocol. No further data is provided on that question. While it is clear that the model used is imperfect and artificial, it is therefore not clear if the findings generalise beyond this specific imperfect model. This remains a major limitation of the manuscript.

We appreciate the reviewer's comment. In the revised manuscript we introduce in more detail 2 models. The first is an in vitro co-culture model detailed in the response to additional comments to the revised version: comment #2 by reviewer #1 (page 3-4 in this rebuttal). Second more experiments with the castration model on both human and mice HSCs carrying *DNMT3A* mutations.

1. **New Fig 2c**, in which PPAR γ i was administered to (CAS) FBM mice, resulting in a significant decrease in the engraftment of human sample #160005 cells, we added in the text: “The administration of PPAR γ i to (CAS) FBM mice resulted in significant decrease of sample #160005 cells engraftment (Figure 2c)”.

2. **New Fig 3f**, we transplanted old *DNMT3A^{Mut}* cells derived from one-year-old mice injected into NBM, (CAS) FBM and CAS+PPAR γ i controls compare *DNMT3A^{WT}* derived BM. We added in the text “We

repeated this experiment on the castration (CAS) FBM model and again a significantly higher engraftment of *DNMT3A*^{Mut} cells derived from one-year-old mice injected into (CAS) FBM in comparison to NBM and PPAR γ i controls was detected. FBM had no effect on *DNMT3A*^{WT} cells (Figure 3f)".

Point 2: The use of a PPAR γ inhibitor control does not prove that IL6 is produced by adipocytes, since it might also be produced by another cell type that gets stimulated by adipocytes.

We thank the reviewer's for his comment. We set up *in vitro* experiments in which we cultured human/mouse MSCs and adipocytes *in vitro* (new FigS12e). Oil red staining was utilized to quantify adipocytes. After ten days in culture, we collected the media and evaluated the IL-6 levels. IL-6 secretion was shown to be higher in mouse/human adipocyte media (new Fig 5c, e) than in human and mice derived MSCs cell media

Point 3: Convincingly addressed

Point 4: Convincingly addressed

Point 5: Explanation makes sense

Point 6: Not ideal. As I remarked earlier these analyses are confounded by changes in cell type abundance. This should at least explicitly be mentioned, if the cell numbers are insufficient to identify the consequences of the experimental treatments at the level of each cell type

Based on the reviewer comment we repeated the single cell RNAseq experiment with the following conditions: NBM_mut and FBM_mut, however we have moved to a 10X platform for the scRNAseq which allowed us to study more cells ~1300 for each condition. We have now replicated all the results of figure 4 not just with a biological replicate and more cells but also with a different library prep platform. We have added all these results to Figure s14. Figure s14a,b demonstrates significantly different distribution of HSPCs subpopulations most of the signal arising from 6 fold increase in HSCs in the FBM_mut cells. Base on the result we have modified the sentence in the abstract regarding this result. **"A 6-10 fold increase in *DNMT3A*^{Mut}-HSCs was observed under FBM conditions in comparison to normal bone marrow".**

We have also edited the results section accordingly:

"We have repeated the scRNA-seq experiment with more cells and with the 10X genomics platform from *DNMT3A*^{Mut} cells exposed to FBM or NBM and validated the significantly increased HSCs population in the *DNMT3A*^{Mut} cells exposed to FBM (this time 6 fold more HSCs than under NBM conditions) (Figure s14a,b)."

We further validated the significantly increased response to IL-6 and and IFN γ pathways in the FBM_mut condition compared to NBM_mut (Figure s14c-f). We have added this text:

"We have repeated the scRNA-seq experiment with more cells and with the 10X genomics platform from *DNMT3A*^{Mut} cells exposed to FBM or NBM and validated that exposure to FBM activated the IFN γ , and IL-6 pathways in *DNMT3A*^{Mut} derived from one year old mice (Figure 14c-f)."

We have also updated the methods section:

“We have repeated the scRNA-seq experiment for validation of our results. One-year-old *DNMT3A^{Mut}* cells were injected to FBM and NBM mice. Then, three days after injection CD45.2 LSK cells were isolated. Cells were sorted to low binding Eppendorf tubes and library was prepared by the 10X genomics V3 3' kit. Libraries were sequenced by Novaseq. For the *DNMT3A^{Mu}* FBM condition we analyzed 1228 cells with median coverage of 60,000 reads per cell and 9440 UMIs per cells. For the *DNMT3A^{Mu}* NBM condition we analyzed 1405 cells with median coverage of 195,000 reads per cell and 9469 UMIs per cells. We have performed the same bioinformatics analysis as in the first experiment.”

Point 7: OK

Point 8: OK

REVIEWER COMMENTS

Reviewer #1 (Remarks to the Author):

The authors have presented a reviewed manuscript that has addressed all my concerns. One of the major concerns on the last version of the manuscript, was the use of a specific marker for adipocytes. As a result, the authors have performed new experiments using perilipin.

The manuscript also incorporates new figures and additional in vitro experiments and scRNAseq data. The in vitro experiments provide support for the role of IL-6. The use of scRNAseq is not integrated with their previous analysis.

There is no other major concern with the manuscript, however, this version has several minor details that need to be addressed.

Manuscript:

Line 56, the references 2 and 3 are not in the superscript.

Line 79, there is an additional dot before references 10,11

Line 130, revise the syntaxes after perilipin, "which is coats lipid droplets in adipocytes"

Lines 148-149. The authors mentioned that "to systematically study FBM interactions with preL-HSC they need to focus on a specific model". However, this is not a valid argument. It can be argued that they could have done it using the four models. The authors need to use their own data to mention a valid reason to choose only two. They mentioned that using an aged model has an impact on HSC, for example. They should also explain why they did not follow the model with PPARgamma activator, which seem to be the "cleaner" model.

Lines 150-152. As the authors are also adding in vitro experiments, I suggest removing this sentence.

Line 164, delete the no after which, as the sentence reads "in which no adipocytes were less accumulated".

Line 182. Include cord blood derived after normal HSPC. It is important to mentioned that the comparison is between CD34+ from different sources, and not only about the presence or absence of DMT3A mutations.

Lines 265-266. The statement is not clearly shown in the figures. The difference in cell type abundance within scRNAseq data might be caused by artifacts in the method (e.g dropout, sorting), considering the low number of cells analyzed and the number of experiments.

Line 273. The authors mentioned that the scRNAseq data cluster in one single cluster according to treatment. The results might reflect the biology of the data, but it can also be caused by batch effects. It is important that the authors rule out the last possibility using an algorithm to correct it (e.g. Harmony).

Line 465. The information about patient samples lacks the number of samples, the source, age and sex. The last information is important as raised by the authors (age and sex).

Figures:

Minor Fig1 g, the IF panels from Week post Irradiation and Week post Irradiation with PPAR(gamma) inhibitor, are missing the size bar and they seems to be on a different magnification from non-irradiated (which do have a bar).

Fig 2 , the format from tables a and d need to be revised, the bottom line from figure a is missing

and there are fine grey lines on the table.

Fig 4 panel b and c , re-format the outer layer of the panels, they look incomplete and crowded in the figure.

Fig 5b, the photographs are not aligned within each panel and between both panels (re-plate 1 and re plate 4. The legends have different sizes.

Supplemental Figures

Suppl Fig 1 panel d, is using the same photographs as the non-irradiated Figure 1. It need to be changed for a non-castrated mouse, instead of a non irradiated (which is the same control).

Suppl Fig 1 panel f, the pictures have very low resolution.

Supplementary Figure legends

Line 77, change UMPA for UMAP

Reviewer #2 (Remarks to the Author):

The manuscript, from its original submission is nearly completely different ranging from its claims, conclusions, interpretations. The authors should be commended for their resilience and efforts, and the study now is supported by science and related data (with some exceptions noted).

However, as the conclusions and scope, that now actually reflect the data generated, arrive at a questionable conceptual advance that needs to be re-evaluated, as the impact of the scientifically supported claims only resonate to a very unique and precise specialized audience.

Other points.

- the models eg. aged etc, are not validated, and do not simply relate to FBM alone. This claim by the authors is simply incorrect. Several other changes take place in these recipient mice. However, its seems the authors have decided to disregard these changes, and contest the models for their derived purposes of FBM. I would strongly suggest validation of these models and keep on those that reflect sound conclusions, as this will not meet standards in the field and likely generate controversy.
- No Quantitative analysis, despite change in markers for adipocytes is not correct. No Adipocyte function or maturity is provided which is now important given the revised title, abstract, and claims of this work. This specific analysis and data is superficial compared to the standards established, although it is applauded that the authors now have the correct techniques, and methods in place.
- comments regarding leukemia are softened and "focused" as the author indicate, but leukemia related to FBM should remain in the discussion alone for a specialized journal that may appreciate this level of speculation.
- although the perceived need and expense to perform scRNA-seq is understandable and admirable, the data derived does not provide any insight into the biology of HSCs, CHIP, or relation to FBM. Other techniques would have given the same, if not better answer
- it is clear what was done with IL-6 neutralization, but this was not the experiment suggested or question. Nonetheless, this is now a moot point, as the grand conclusion have been removed, as is now the impact of the study related to those claims.

Reviewer #3 (Remarks to the Author):

The authors have addressed all my concerns.

Reviewer #4 (Remarks to the Author):

I read the manuscript again with a fresh mind and I do think it adds to the field. The authors have now explained much better why these, definitely imperfect, fatty bone marrow transplantation models are useful. The technical concerns I had are resolved and the added data from the castration model, the expanded use of PPAR-gamma inhibitors, and the inclusion of replicates in single cell RNA-seq, make the results much more convincing. Overall, I recommend publication.

RESPONSE TO REVIEWERS' COMMENTS

Reviewer #1

1. There is no other major concern with the manuscript, however, this version has several minor details that need to be addressed.

Manuscript:

We thank the reviewer for all the help.

2. Line 56, the references 2 and 3 are not in the superscript. - Done

3. Line 79, there is an additional dot before references 10,11- Done

4. Line 130, revise the syntaxes after perilipin, “which is coats lipid droplets in adipocytes” – We changed the sentence to: “which, is located on the surface layer of intracellular lipid droplets²⁸”

We have added reference 28: Blanchette-Mackie, E. J. et al. Perilipin is located on the surface layer of intracellular lipid droplets in adipocytes. *J Lipid Res* 36, 1211–1226 (1995).

5. Lines 148-149. The authors mentioned that “to systematically study FBM interactions with preL-HSC they need to focus on a specific model”. However, this is not a valid argument. It can be argued that they could have done it using the four models. The authors need to use their own data to mention a valid reason to choose only two. They mentioned that using an aged model has an impact on HSC, for example. They should also explain why they did not follow the model with PPARgamma activator, which seem to be the “cleaner” model. –

We thank the reviewer for this comment. Although the *PPARgamma* agonist was effective, the amount of fat that was accumulated was sparse as can be seen in figure supp1f and took over one month.

Furthermore, we predicted that calibrating the BADGE control and the rosiglitazone maleate would be challenging (as there would have been a period of overlap between the two).

We have now added these considerations to the text (now lines 159-167)

*“While none of our models is perfect we chose to focus our efforts on the interaction between human preL-HSPCs and FBM in the post-irradiation and post-castration FBM models. We selected the irradiation model as it was the most robust (reproducible) and created the largest effect (accumulation of FBM). We decided to work with the castration model because it best reflects the aging process in males. We predicted that calibrating the BADGE control and the rosiglitazone maleate (both dealing with *PPARG*)*

would be challenging. Additionally, rosiglitazone maleate produced fewer adipocytes; despite these drawbacks, this model should be improved and tested in the future, perhaps with a different control. ~~The decision to use an *in vivo* model rather than an *in vitro* model stemmed from the fact that *in vivo* model allow both human and mice stem cell self-renewal assays. The influence of the microenvironment is long-term, making *in vitro* experiments difficult to monitor~~.
”

6. Lines 150-152. As the authors are also adding *in vitro* experiments, I suggest removing this sentence. - Done (now line 165-168).

7. Line 164, delete the *no* after which, as the sentence reads “in which no adipocytes were less accumulated”. – Done

8. Line 182. Include cord blood derived after normal HSPC. It is important to mentioned that the comparison is between CD34+ from different sources, and not only about the presence or absence of DNMT3A mutations. – We added in line 194: “we used normal CD34+ cells (no DNMT3A mutation) derived from different sources” and added in line 200: “but not for normal cord blood derived HSPCs.”

9. Lines 265-266. The statement is not clearly shown in the figures. The difference in cell type abundance within scRNAseq data might be caused by artifacts in the method (e.g dropout, sorting), considering the low number of cells analyzed and the number of experiments.

We thank the reviewer for this comment. The reviewer is referring to the following sentence (now lines 284-285): “Altogether the scRNA-seq data confirmed our hypothesis that *DNMT3A*^{Mut} cells exposed to FBM undergo self-renewal, while all other conditions undergo (mostly myeloid) differentiation. “
Indeed scRNAseq has its biases, and accordingly we have repeated the scRNAseq experiment with significantly higher number of cells and with the most important experiment. The increased HSC frequency in the FBM was replicated with very low P value (See here Figure s15, b also below here). Based on the reproducibility of the scRNA-seq data and the data from other experiments we believe our statement is correct. However the reviewer is correct that we mainly validated the NBM versus FBM conditions and that we can tone down the statement, so we have modified the sentence accordingly:

“Altogether, the scRNA-seq data like the secondary engraftment experiments also suggests ~~confirmed our hypothesis~~ that *DNMT3A*^{Mut} cells exposed to FBM undergo self-renewal, while under NBM conditions they undergo (mostly myeloid) differentiation.”

10. Line 273. The authors mentioned that the scRNAseq data cluster in one single cluster according to treatment. The results might reflect the biology of the data, but it can also be caused by batch effects. It is important that the authors rule out the last possibility using an algorithm to correct it (e.g. Harmony).

We thank the reviewer for this comment. The issue of batch effects and other lateral effects (cell cycle stress) has been dealt in our analysis and is part of the Metacell pipeline (Ben-Kiki, O., Bercovich, A., Lifshitz, A. & Tanay, A. Metacell-2: a divide-and-conquer Metacell algorithm for scalable scRNA-seq analysis. Genome Biol 23, 100 (2022).)

Here is the quotation from the manuscript now lines 646-648:

" We then used the Metacell library for noise reduction, clustering and cell type annotation^{42, 43}. Removal of lateral effects (cell-cycle, stress) and batch effects was performed using gene module analysis to filter genes used for Metacells grouping."

We have used the same methodology for all our scRNAseq and single nuclei. The analysis was performed by Bercovich, A from the Tanay A group who is the second author of the current manuscript and the Metacell2 manuscript.

11. Line 465. The information about patient samples lacks the number of samples, the source, age and sex. The last information is important as raised by the authors (age and sex).-

We thank the reviewer for this comment. We have added this as a supplementary table: "Information on all samples can be found in the Table S7."

Figures:

12. Minor Fig1 g, the IF panels from Week post Irradiation and Week post Irradiation with PPAR(gamma) inhibitor, are missing the size bar and they seems to be on a different magnification from non-irradiated

(which do have a bar). We added the size bar

13. Fig 2 , the format from tables a and d need to be revised, the bottom line from figure a is missing and there are fine grey lines on the table. - Done

14. Fig 4 panel b and c , re-format the outer layer of the panels, they look incomplete and crowded in the figure.- Done

15. Fig 5b, the photographs are not aligned within each panel and between both panels (re-plate 1 and re plate 4. The legends have different sizes.- Done

Supplemental Figures

16. Suppl Fig 1 panel d, is using the same photographs as the non-irradiated Figure 1. It need to be changed for a non-castrated mouse, instead of a non irradiated (which is the same control). – Done

17. Suppl Fig 1 panel f, the pictures have very low resolution – Resolution was improved

Supplementary Figure legends

18. Line 77, change UMPA for UMAP - Done

Reviewer #2

The manuscript, from its original submission is nearly completely different ranging from its claims, conclusions, interpretations. The authors should be commended for their resilience and efforts, and the study now is supported by science and related data (with some exceptions noted).

We thank the reviewer for their effort and scientific honesty.

However, as the conclusions and scope, that now actually reflect the data generated, arrive at a questionable conceptual advance that needs to be re-evaluated, as the impact of the scientifically supported claims only resonate to a very unique and precise specialized audience.

Other points.

- the models eg. aged etc, are not validated, and do not simply relate to FBM alone. This claim by the authors is simply incorrect. Several other changes take place in these recipient mice. However, its seems the authors have decided to disregard these changes, and contest the models for their derived purposes of FBM. I would strongly suggest validation of these models and keep on those that reflect sound conclusions, as this will not meet standards in the field and likely generate controversy.*

We thank the reviewer for their effort and scientific honesty. It seems we will not reach an agreement

on this point. No model is perfect. The choice of the right model depends on the question. In the current study we took a deep delve into modelling the interaction between human preL-HSPCs and FBM. We tried to control the side effects of our models with the BADGE controls and with a second model (Castration) which mimics more faithfully human ageing. Now we have added castration experiments for both human and mice and with the BADGE control. We have replicated some of results *in vitro*. We have added the *in vitro* results per the reviewer request although they too have many pitfalls and biases in their current form (mainly batch effects due to interindividual heterogeneity), but also the lack of other cell types which might be of importance (like monocytes and macrophages in the cases of *DNMT3A* mutations as they also can modulate the interaction between FBM and preL-HSPCs). In the future we will perform more sophisticated human *in vitro* assays especially once an advance in technology will allow human HSCs to undergo extensive selfrenewal *in vitro*. For now, the best model for human selfrenewal and other stem cell functions is still the xenograft model. As many mice and human cytokines do not cross react growing human cells on mice adipocytes is not a solution. So while the reviewer is correctly criticizing our models they do not provide with a better solution for the human studies. Our study is pioneering in this regard as the interaction between human cells and FBM was not studied almost at all. Indeed our models might ignite some controversy in the field which was sleepy to begin with. Each one of the experiments we did by itself can have many biases, however taken together they provide evidence to our claims.

• *No Quantitative analysis, despite change in markers for adipocytes is not correct. No Adipocyte function or maturity is provided which is now important given the revised title, abstract, and claims of this work. This specific analysis and data is superficial compared to the standards established, although it is applauded that the authors now have the correct techniques, and methods in place.*

We thank the reviewer for this comment. We agree with reviewer 2 that a better description of the adipocytes in our models might provide some more insights. For the post irradiation *in vivo* model we aimed at characterizing gene expression of single nuclei from BM cells from mice a week after irradiation and the same from the BADGE control mice. We compared our results to those of Baccin et.al Nature cell Biology 2020 which reported the gene expression patterns of BM derived adipocytes and other BM cells.

Initially, we thought to present this data in future work. We are currently trying to develop an *in vitro* system that will mimic FBM without using adipocytes as they are prone for batch effects, and they do not support human self renewal. To this aim we plan to compare scRNAseq data (adipocytes and other cells) between *in vivo* and *in vitro* in both mice and human, find common features and try to create a media that will mimic such conditions. The task is even more challenging among humans as we will have to take into account interindividual heterogeneity in the source of FBM especially taking into account age and gender. For all of these reasons we believe that what the reviewer is asking is out of the scope of the current study. However, as the current study relies on the post irradiation FBM model we can add these data now.

“To better characterize the adipocytes in our FBM model we have performed single nuclei RNA sequencing on BM derived cells, as was previously described²⁹. BM cells from both irradiated NSG mice and BADGE

control mice were analyzed. Gene expression in our cells was compared to a recently reported atlas of mice BM cells³⁰. While the majority of our cells originated from the hematopoietic lineage, we identified 25 adipocytes most of them (68%) were retrieved from the irradiated mice and the rest from the BADGE control. Our Adipocytes (n=25) clustered together with previously reported BM derived adipocytes (n=9) in the same Metacells, and with other adipocyte Metacells (Figure S3a Metacells#81) and expressed many known adipocyte markers (*Lepr*, *Adipoq*, *Cxcl12*, *Cxcl14*, *Kng1*, *Lpl* and more) (Figure S3b). The low number of adipocytes in our experiments reduces the ability to identify heterogeneity in adipocyte populations in our data, however it suggests that they share high degree of similarity with normal adipocytes. The integration of our data with the data by Baccin et.al can be observed here: (https://tanaylab.weizmann.ac.il/MCV/FBM/Single_nuclei_for_adipocytes/)."

We encourage the reviewer to explore the data.

All the raw data is located here:

https://www.dropbox.com/sh/aohmg1besu500j6/AADWFFaVeOsO_nq-5A1Dzxllra?dl=0

To gain a better understanding on the cytokine levels in our adipocytes we specifically looked for cytokines which we identified in our cytokine array analysis.

"Next, we tested cytokine expression in our single nuclei RNAseq data. The adipocytes derived from irradiated NSG mice expressed IL-6, IL-18 and IL-1b. Specifically, IL-6 was not expressed in all adipocyte Metacells (from the Baccin et.al data) and the same was true for other cytokines as well. These suggests that some of the heterogeneity between BM adipocytes is driven by cytokine expression (Figure S3c). To better characterize such heterogeneity in our adipocyte many more cells are needed which is not the scope of the current study."

We have also added a section in the methods:

"Enrichment for mouse adipocytes for single nuclei sequencing: Both femurs and tibias were collected and cleaned in sterile PBS. Both ends of femurs and tibias were snapped. Bones were placed in a 0.6-mL microcentrifuge tube that was cut open at the bottom and placed into a 1.5-mL microcentrifuge tube. Fresh bone marrow was spun out by quick centrifuge (from 0 to 10,000 rpm, 9 s, room temperature (RT)). Red blood cells (RBC) were lysed using RBC lysing buffer (Sigma). After centrifugation (3,000 rpm, 5 min, RT), floating cells were collected from the top layer and washed with PBS for three times. "

And also in the subsection of scRNAseq

"To gain more insights into the characteristics of adipocytes under our FBM irradiation model. we have used single-nuclei RNA sequencing (snRNA-seq) due to the challenging fragility of lipid-filled adipocytes. We have adopted a protocol developed by others²⁹. The protocol combines an adipocyte enrichment protocol (described above) followed by nuclei "cleanup" step by sorting. Nuclei were sorted to low binding Eppendorf tubes and library was prepared by the 10X genomics V3 3' kit. Libraries were sequenced by Novaseq. We have used the Metacell platform to analyze the cells. After filtering empty cells and doublets

we analyzed 4058 cells from the irradiation FBM model and 3428 cells from the BADGE control group. Removal of lateral effects (cell-cycle, stress) was performed using gene module analysis to filter genes used for Metacells grouping, and cells with high expression of mitochondrial genes were excluded. We have combined our dataset together with that of Baccin et.al and have annotated cells using key gene markers described in Baccin et.al³⁰. The Metacell analysis partitioned cells to 181 Metacells. Next, we compared the nuclei we sequenced to adipocytes from Baccin et.al. The majority of the cells in our cohort were monocyte/neutrophils and late erythroid cells, which had similar gene expression compared to the same populations in Baccin et.al. We identified one Metacell and 25 cells in our cohort which clustered together with 9 adipocytes from Baccin et al. these cells had similar gene expression.”

- *comments regarding leukemia are softened and “focused” as the author indicate, but leukemia related to FBM should remain in the discussion alone for a specialized journal that may appreciate this level of speculation.*

- *although the perceived need and expense to perform scRNA-seq is understandable and admirable, the data derived does not provide any insight into the biology of HSCs, CHIP, or relation to FBM. Other techniques would have given the same, if not better answer*

- *it is clear what was done with IL-6 neutralization, but this was not the experiment suggested or question. Nonetheless, this is now a moot point, as the grand conclusion have been removed, as is now the impact of the study related to those claims.*

Reviewer #3

The authors have addressed all my concerns.

We thank the reviewer for all the help.

Reviewer #4

I read the manuscript again with a fresh mind and I do think it adds to the field. The authors have now explained much better why these, definitely imperfect, fatty bone marrow transplantation models are useful. The technical concerns I had are resolved and the added data from the castration model, the expanded use of PPAR-gamma inhibitors, and the inclusion of replicates in single cell RNA-seq, make the results much more convincing. Overall, I recommend publication.

We thank the reviewer for all the help.

REVIEWERS' COMMENTS

Reviewer #1 (Remarks to the Author):

The authors have addressed all my suggestions.

Two minor suggestions:

Line 99. Study and studying words are very close to each other. I suggest using a synonym or change the sentence to avoid repetition.

Line 160. I suggest changing the sentence "while none of our models is perfect", with a sentence that acknowledges the fact that the models increased the number of adipocytes, but the effect may not be specific to only that cell type. Perfect is subjective, and is better to be specific to the fact that the models are not specific to increase the number of adipocytes.

RESPONSE TO REVIEWERS' COMMENTS

Reviewer #1

The authors have addressed all my suggestions.

We thank the reviewer for all the help.

Two minor suggestions:

Line 99. Study and studying words are very close to each other. I suggest using a synonym or change the sentence to avoid repetition.

We changed the text to: As in the current study we aimed at **investigating** both human and mice preL-HSPCs

Line 160. I suggest changing the sentence "while none of our models is perfect", with a sentence that acknowledges the fact that the models increased the number of adipocytes, but the effect may not be specific to only that cell type. Perfect is subjective, and is better to be specific to the fact that the models are not specific to increase the number of adipocytes.

We changed the text to: **While all of our models increase FBM, they most probably have other molecular and cellular consequences in the BM and other tissues. To mitigate these of target effects we chose to study** the interaction between human preL-HSPCs and FBM in **two different models**: post-irradiation and post-castration FBM models.